# Homeostatic membrane tension constrains cancer cell dissemination by counteracting BAR protein assembly

Kazuya Tsujita [1,2,3 ✉], Reiko Satow[4], Shinobu Asada[4], Yoshikazu Nakamura[4,5], Luis Arnes[6], Keisuke Sako [7], Yasuyuki Fujita[8], Kiyoko Fukami[4] & Toshiki Itoh[1,2]

Malignancy is associated with changes in cell mechanics that contribute to extensive cell deformation required for metastatic dissemination. We hypothesized that the cell-intrinsic physical factors that maintain epithelial cell mechanics could function as tumor suppressors. Here we show, using optical tweezers, genetic interference, mechanical perturbations, and in vivo studies, that epithelial cells maintain higher plasma membrane (PM) tension than their metastatic counterparts and that high PM tension potently inhibits cancer cell migration and invasion by counteracting membrane curvature sensing/generating BAR family proteins. This tensional homeostasis is achieved by membrane-to-cortex attachment (MCA) regulated by ERM proteins, whose disruption spontaneously transforms epithelial cells into a mesenchymal migratory phenotype powered by BAR proteins. Consistently, the forced expression of epithelial–mesenchymal transition (EMT)-inducing transcription factors results in decreased PM tension. In metastatic cells, increasing PM tension by manipulating MCA is sufficient to suppress both mesenchymal and amoeboid 3D migration, tumor invasion, and metastasis by compromising membrane-mediated mechanosignaling by BAR proteins, thereby uncovering a previously undescribed mechanical tumor suppressor mechanism.

[1] Biosignal Research Center, Kobe University, Kobe, Hyogo 657-8501, Japan. [2] Division of Membrane Biology, Department of Biochemistry and Molecular Biology, Kobe University Graduate School of Medicine, Kobe, Hyogo 650-0017, Japan. [3] AMED-PRIME, Japan Agency for Medical Research and Development, Tokyo 100-0004, Japan. [4] Laboratory of Genome and Biosignals, Tokyo University of Pharmacy and Life Sciences, Hachioji, Tokyo 192-0392, Japan. [5] Department of Applied Biological Science, Faculty of Science and Technology, Tokyo University of Science, Noda, Chiba 278-8510, Japan. [6] The Novo Nordisk Foundation Center for Stem Cell Biology (DanStem), Biotech Research & Innovation Centre, University of Copenhagen, Copenhagen, Denmark. [7] National Cerebral and Cardiovascular Center Research Institute, Osaka 565-8565, Japan. [8] Division of Molecular Oncology, Graduate School of Medicine, Kyoto University, Kyoto 606-8501, Japan. ✉email: tsujita@people.kobe-u.ac.jp

Although metastasis is the major cause of cancer-related deaths, our understanding of key determinants underlying tumor dissemination is limited[1]. From a biophysical perspective, it has long been proposed that changes in cell mechanics are integral to metastatic dissemination given that cancer cells must undergo extensive deformation to migrate through tissues and enter blood vessels[2–6]. Recent advances in nanotechnology have unveiled "mechanical signatures" of malignant cells, as showed by the strong correlation between decreasing cell stiffness and increasing invasive and metastatic efficiency[5–12]. This suggests that heritable changes in cell mechanics, which should be caused by malignant progression, are critical to the metastatic process. Conversely, epithelial cells may already have strategies to maintain mechanical homeostasis, which could function as an endogenous tumor suppressor. However, the key cell-intrinsic physical parameters for the transition to the malignant phenotype remain unknown, limiting our understanding of the link between oncogenic signaling and dysregulation of cell mechanics, and of how mechanical changes are transduced (mechanotransduction) to regulate cancer cell motility.

Cell motility is fundamental to metastatic dissemination[13]. Recent studies using three-dimensional (3D) environments have indicated that malignant cells exhibit two different invasive migration modes, termed mesenchymal and amoeboid migration[14–17]. Mesenchymal migration, which shows a spindle-like shape, depends on PM protrusions pushed forward by Arp2/3 complex-dependent branched actin polymerization. In contrast, amoeboid migration, which is characterized by a rounded morphology, is highly heterogeneous and displays both actin-based protrusions and contractility-driven membrane blebs. Importantly, these two migration modes are interconvertible: cancer cells can actively switch between these modes of movement and even exhibit mixed phenotypes of both[16,18,19]. Such plasticity and complexity are considered a major obstacle in developing therapeutic strategies to limit tumor invasion and dissemination[16,17,20].

Plasma membrane (PM) reversibly associates with the actin cortex via linker proteins, such as ezrin, radixin, and moesin (ERM) family proteins, whereby cell membrane mechanics are intrinsically dependent on the degree of membrane-cortex attachment (MCA)[21–24]. Indeed, it has emerged that PM tension, which is primarily determined by this composite structure[25–28], plays an essential role in cell-shape changes and motility[26,29–33]. Given that tense membranes resist cell membrane deformations, PM tension is assumed to disfavor the formation of any membrane protrusions[21,29,31,34]. This led us to hypothesize that tensional homeostasis in the PM might be an inherent mechanical feature that maintains epithelial cells in a non-motile state, with mechanisms that could function as a potent metastasis suppressor if restored in malignant cell settings.

Here, we use optical tweezers, genetic interference, cancer genome data, mechanical perturbations, and in vivo studies to identify homeostatic PM tension as a mechanical suppressor of cancer cell dissemination by counteracting mechanosensitive BAR proteins. We demonstrate that reduced PM tension is a mechanical hallmark of malignant cells, regardless of whether they display mesenchymal or amoeboid motility modes. Maintaining high PM tension is sufficient to suppress such 3D migration modes, tumor invasion, and metastasis while having, in principle, no effect on non-tumorigenic cells. This work paves the way for new precise therapeutic strategies for the treatment of metastatic cancers by targeting the cell membrane mechanics of cancer cells.

## Results

**Epithelial cells have higher PM tension than their malignant counterparts.** To determine whether there is a difference in PM tension between epithelial cells and their metastatic counterparts,

we used optical tweezers to analyze the membrane tether force, which is proportional to PM tension[21–24] (Supplementary Fig. 1a). We primarily used human non-invasive mammary epithelial cells (MCF10A) and metastatic breast cancer cells (MDA-MB-231), as they are a commonly used model of malignant transition. To avoid cell-extrinsic effects, such as cell–cell contact, we measured the tether force at the single-cell level. We found that the tether force of MCF10A cells was largely comparable to that of the low-invasive human breast cancer cells (AU565 and MCF7; Fig. 1a). Unlike these low-invasive cells, metastatic breast cancer cells, such as MDA-MB-231 and Hs578T cells, formed both prominent membrane ruffles and blebs when cultured on uncoated glass substrates (Supplementary Fig. 1b and Supplementary Movie 1). Interestingly, we found that there was no difference in the tether force between ruffling and blebbing cells, and their tether force was significantly lower than that of their low-motility counterparts (Fig. 1a). Similar results were obtained in aggressive prostate cancer cells (PC-3) and pancreatic cancer cells (PANC-1; Fig. 1a). PM tension was approximately twofold lower in metastatic cells than in low-motility cells (Supplementary Fig. 1c). The ability to maintain higher PM tension than malignant cells appears to be a common characteristic of epithelial cells across species and tissues, as the tether force of normal epithelial cells, such as canine kidney MDCK II cells and rat liver IAR-2 cells, was similar to that of MCF10A cells (Fig. 1a and Supplementary Fig. 1c).

PM tension has contributions from the in-plane tension of the lipid bilayer and MCA (Supplementary Fig. 1d). Given that PM tension is thought to be largely dependent on MCA[25–27], we focused on ERM proteins and F-actin beneath the PM. ERM proteins are activated by the phosphorylation of conserved threonine residues[35]. We observed that MCA10A and AU565 cells had intense phosphorylated ERM (pERM) signals globally decorating the PM, where they partially co-localized with F-actin (Fig. 1b, c and Supplementary Fig. 1e, f). In contrast, MDA-MB-231 and Hs578T cells had globally lower levels of membrane-associated pERM and F-actin, regardless of whether they exhibited membrane ruffling or blebbing, and some pERM levels were restricted to the cell rear and possibly shrinking blebs (Fig. 1b, c and Supplementary Fig. 1e, f). Consistent with these 2D observations, confocal and reconstituted 3D images showed that MCF10A and AU565 cells embedded in a collagen matrix (3D) always exhibited a rounded shape with uniform and intense pERM and F-actin levels beneath the PM (Fig. 1d, e and Supplementary Fig. 1g, h). MDA-MB-231 cells exhibited both actin-based elongated mesenchymal and actin- and bleb-based rounded amoeboid migration phenotypes in 3D[19] (Fig. 1d and Supplementary Fig. 1h). In both cases, membrane-proximal pERM and F-actin displayed overall decreased levels (Fig. 1d, e). Similarly, Hs578T cells, which predominantly exhibit an actin-based elongated phenotype in 3D (Supplementary Fig. 1g, h), gave a consistent phenotype (Fig. 1e). These results suggest that the observed differences in PM tension between non-motile and invasive cells are primarily due to ERM-mediated changes in MCA and that such mechanical disruptions correlate with an invasive phenotype irrespective of the type of protrusions and motile modes.

**Epithelial cells spontaneously convert to a mesenchymal migratory phenotype when PM tension is reduced.** The above findings prompted us to hypothesize that reducing PM tension may be sufficient for epithelial cells to acquire invasive behavior. To test this, we manipulated MCA by simultaneously knocking down three members of ERM (ezrin, radixin, and moesin) or their specific kinases (SLK and STK10 [also known as LOK])[36]

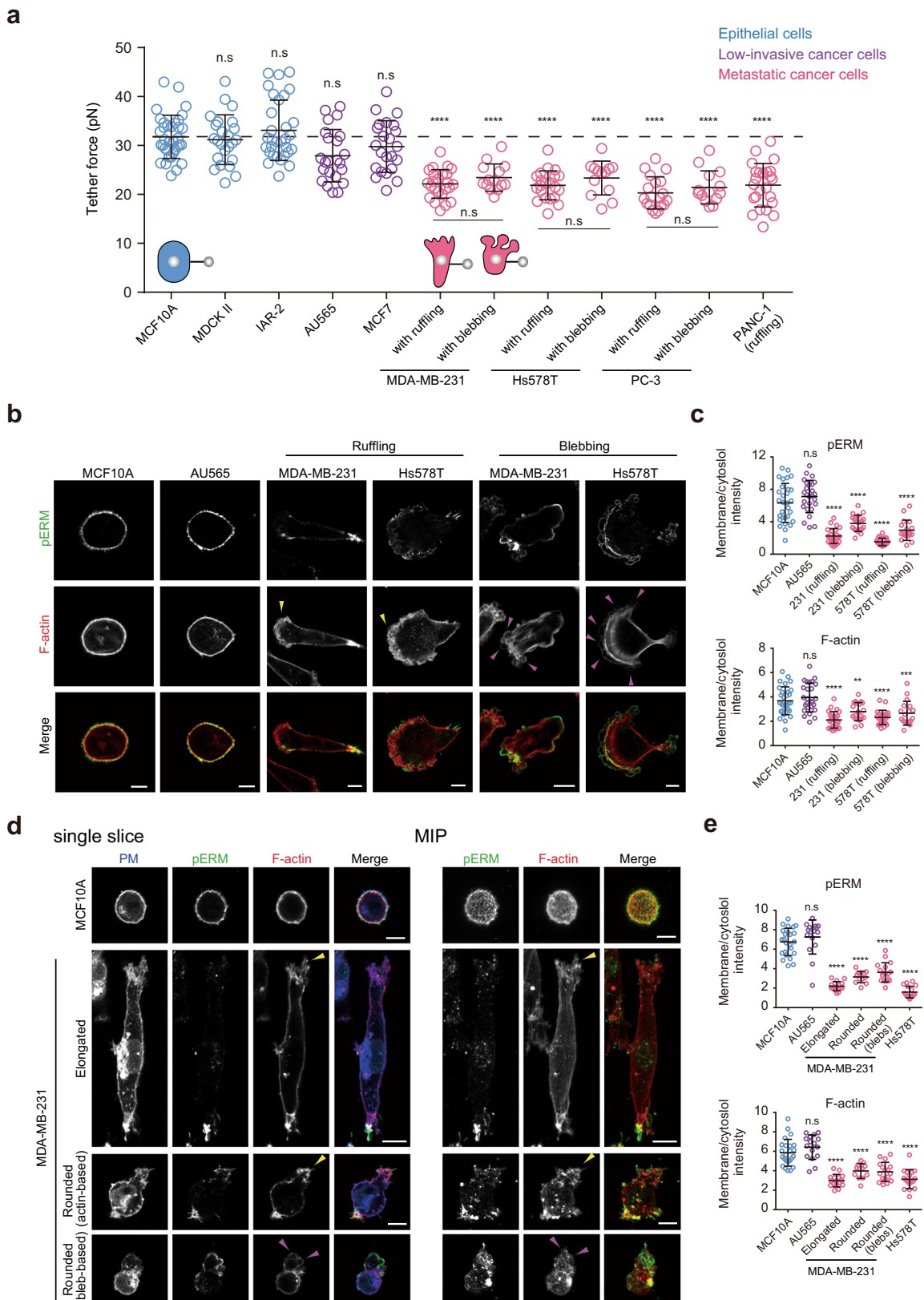

(Supplementary Fig. 2a). We also focused on RHOA, as it is known to act as an upstream regulator of these ERM kinases[37] (Supplementary Fig. 1d). Paradoxically, despite its key role in cell migration, RHOA has been reported to play an inhibitory role in invasion and metastasis[38,39]. Notably, knockdown of these proteins in MCF10A cells led to a significant decrease in tether force

(Supplementary Fig. 2b), which corresponded to a decrease in PM tension (Fig. 2a). Of note, phosphorylated myosin light chain (pS19MLC), a marker for myosin II activation, was not affected in ERM and SLK+STK10 depleted cells (Supplementary Fig. 2c), indicating that the observed differences in PM tension were specifically due to changes in MCA rather than those in

**Fig. 1 Plasma membrane (PM) tension is higher in non-invasive cells than in their metastatic counterparts. a** Scatter plot comparing the tether force of the indicated cells. $n = 35$ (MCF10A), $n = 24$ (MDCK II), $n = 31$ (IAR-2), $n = 23$ (AU565), $n = 23$ (MCF7), $n = 24$ (MDA-MB-231 with ruffling), $n = 14$ (MDA-MB-231 with blebbing), $n = 21$ (Hs578T with ruffling), $n = 13$ (Hs578T with blebbing), $n = 19$ (PC-3 with ruffling), $n = 14$ (PC-3 with blebbing), $n = 24$ (PANC-1) cells pooled from three independent experiments. Mean ± SD. **b** Confocal images of the indicated cells stained with anti-phosphorylated ERM (pERM) antibodies and phalloidin. Yellow and magenta arrowheads indicate actin- and bleb-based protrusions, respectively. See also Supplementary Fig. 1e. **c** Quantification of (**b**). Membrane/cytoplasm intensity ratio of pERM and F-actin of $n = 30$ (MCF10A), $n = 26$ (Au565), $n = 24$ (MDA-MB-231 with ruffling), $n = 20$ (MDA-MB-231 with blebbing), $n = 22$ (Hs578T with ruffling), and $n = 18$ (Hs578T with blebbing) cells pooled from three independent experiments. Mean ± SD. $**P = 0.0073$; $***P = 0.0014$. **d** Left, confocal images of MCF10A or MDA-MB-231 cells stained with anti-pERM antibodies, phalloidin, and wheat germ agglutinin (WGA) in a 3D collagen matrix (3D). Plasma membrane (PM) was labeled with WGA. Right, maximum intensity projections (MIP) of their confocal stacks. Yellow and magenta arrowheads indicate actin- and bleb-based protrusions, respectively. **e** Quantification of (**d**) and Supplementary Fig. 1g. Membrane/cytoplasm intensity ratio of pERM and F-actin of $n = 24$ (MCF10A), $n = 16$ (AU565), $n = 15$ (MDA-MB-231, elongated), $n = 12$ (MDA-MB-231, rounded with actin-based protrusion), $n = 17$ (MDA-MB-231, rounded with blebs), and $n = 17$ (Hs578T) cells pooled from three independent experiments. Mean ± SD. Significance tested using one-way ANOVA with Tukey's multiple comparisons test (**a**) and the two-tailed Mann–Whitney test (**c**, **e**). n.s. not significant; $****P < 0.0001$. All scale bars, 10 μm.

actomyosin contractility. These knocked-down cells exhibited overall decreased levels of membrane-associated pERM and F-actin, concomitant with cell spreading with prominent actin-based protrusions, such as lamellipodia and membrane ruffling in 2D (Fig. 2b, arrowheads, c, Supplementary Fig. 2d and Supplementary Movie 2). We next evaluated the effects of decreased PM tension in a 3D on-top culture system with a mixture of collagen and Matrigel commonly used in cancer research[40]. As expected, control cells formed perfectly rounded spheroids[40] (Fig. 2d). Strikingly, over 60% of the low-tension cells displayed an elongated invasive phenotype, exhibiting cell dissemination into the surrounding matrix (Fig. 2d). These cells consistently displayed a marked increase in motility behaviors, including invasion and confined migration through narrow gaps (Fig. 2e, f). The observed invasive phenotype seemed to be independent of the classical EMT program, as these low-tension cells retained epithelial characteristics (Supplementary Fig. 2c), and E-cadherin depletion resulted in a slight but significant inhibition of invasion and migration (Fig. 2e, f). The deletion of ERM proteins in low-invasive breast cancer cells also led to a drastic increase in invasion (Supplementary Fig. 2e). Decreased MCA did not affect the proliferation of MCF10A cells (Supplementary Fig. 2f).

We further investigated the effect of PM tension reduction on single-cell motility behavior in 3D. Time-lapse movies showed that approximately 95% of control RNAi-treated cells displayed a rounded shape with a non-motile phenotype in a single-cell state when embedded in a collagen matrix (Fig. 2g, Supplementary Fig. 2g and Supplementary Movie 3). In contrast, knockdown of the MCA regulators all led to conversion to the migration phenotype in a mesenchymal fashion: approximately 50% of the cells displayed an elongated migration phenotype with dynamic cell-shape changes showing membrane protrusion and retraction (Fig. 2g, Supplementary Fig. 2g and Supplementary Movie 3). No amoeboid-like movement characterized by a rounded shape was observed in these knocked-down cells. In 3D, such an elongated phenotype was closely associated with lower levels of pERM and F-actin beneath the PM (Fig. 2h, i) as observed in metastatic cells. These data indicate that non-motile epithelial cells can spontaneously convert to a mesenchymal-like motility phenotype when PM tension is reduced.

**Correlation between reduced PM tension and malignant progression.** We noted that the elongated cell-shape changes induced by decreased PM tension were strikingly similar to those induced by EMT. Therefore, we assumed that the disruption of tensional homeostasis in the PM might be commonly associated with malignant transformation. To test this hypothesis, we established MCF10A cell lines stably overexpressing Snail or Slug, transcription factors that strongly induce EMT[41]. As expected, these

cells displayed EMT characteristics, including actin-based protrusion formation (Fig. 3a, arrowhead) and altered expression of E-cadherin and vimentin (Supplementary Fig. 3a). Interestingly, Snail expression resulted in a significant reduction in membrane-associated pERM and F-actin levels (Fig. 3a, b). Moreover, MCF10A cells overexpressing Snail or Slug exhibited lower PM tension than that in the parental cells (Fig. 3c). A similar result was obtained in MDCK II cells inducibly expressing a K-Ras-activating mutant (G12V) (Supplementary Fig. 3b), which is a known key driver of cancer progression and metastasis[42]. In 3D, EMT-induced elongated morphology was closely associated with altered pERM and actin staining patterns, as in 2D (Fig. 3d–f), suggesting a close correlation between changes in PM tension and the acquisition of an invasive phenotype.

To further clarify whether PM tension reduction is a common feature involved in cancer progression, we analyzed cancer genomes using pan-cancer data from The Cancer Genome Atlas (TCGA)[43]. Surprisingly, a comprehensive analysis of 14 major carcinomas across 6586 patients revealed that epithelial tumors frequently harbor putative heterozygous deletions of RHOA, SLK, STK10, and ERM (Fig. 3g and Supplementary Fig. 3c). Similar trends were observed in 961 cancer cell lines in the Cancer Cell Line Encyclopedia[44] (CCLE; Supplementary Fig. 3d). Moreover, a meta-analysis revealed significant associations between the increased expression of ERM kinases and increased patient survival in breast, lung, and gastric cancers (Fig. 3h). These data suggest that the disruption of homeostatic PM tension is a common mechanical characteristic of malignant cells and may be correlated with cancer progression.

**Increasing PM tension is sufficient to suppress 3D migration, tumor invasion, and metastasis.** We hypothesized that if PM tension reduction is key to acquiring migration and invasiveness, increasing MCA might be sufficient to suppress cancer cell dissemination. Therefore, we attempted to increase PM tension by directly manipulating MCA. Given that ERM proteins globally dissociate from the PM in aggressive cancer cells, we reasoned that membrane-targeted active ezrin (MA-ezrin) could rescue decreased PM tension. To test this, we engineered an MA-ezrin construct in which the conserved myristoylation sequence of Lyn was fused with ezrin, followed by the introduction of a phosphomimetic-activating mutation (T567E)[35], thereby maintaining its active state throughout the PM (Fig. 4a). Notably, its expression in MDA-MB-231 cells led to a significant increase in PM tension (Fig. 4a). MA-ezrin was globally localized to the PM, resulting in the suppression of prominent actin- and bleb-based membrane protrusions (Fig. 4b, c). These high-tension cells exhibited normal cell proliferation ability in vitro (Supplementary Fig. 4a), but a marked decrease in invasion and migration (Fig. 4d).

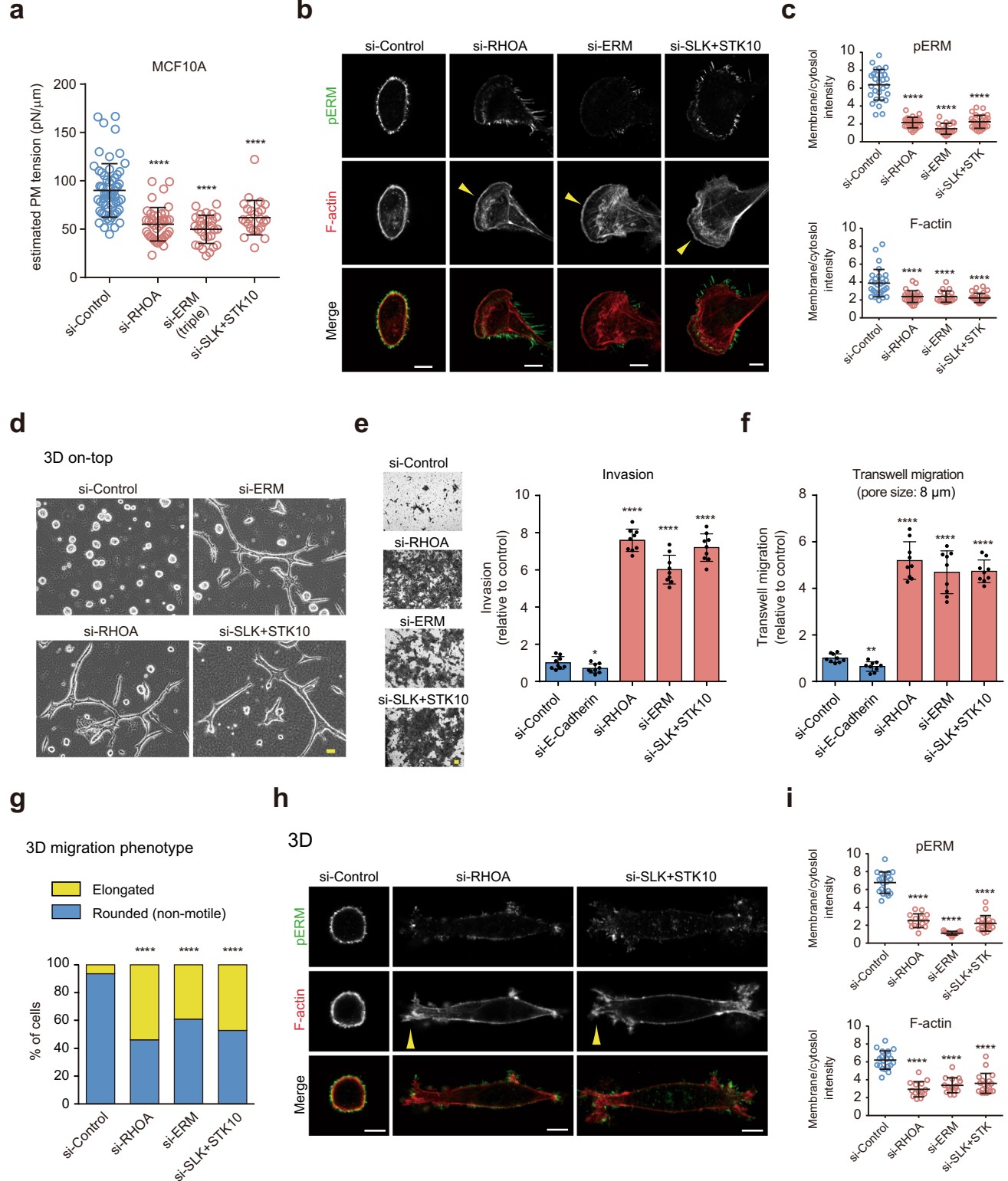

ERM activity has also been reported to be associated with cortical stiffness, at least in mitotic cell rounding[45]. We found that treatment with calyculin A, which is commonly used to increase cortical stiffness or tension by increasing myosin II activity, has no effect on the invasive ability (Supplementary Fig. 4b), reflecting the plasticity of cancer cell migration due to changes in contractility[16,17,46]. Furthermore, treating cells with methyl-β-cyclodextrin (MβCD), a cholesterol-removing

compound that increases PM tension[23,47] (Supplementary Fig. 4c) but decreases cortical stiffness[48], significantly reduced invasion (Supplementary Fig. 4b), ruling out potential effects of cortical stiffness on inhibiting cancer cell migration.

Next, we tested whether an increase in PM tension affected 3D migration. 3D reconstituted confocal images showed that while ezrin-expressing cells displayed different protrusion phenotypes, as in parental cells, MA-ezrin-expressing cells had rounded

**Fig. 2 Decreased PM tension transforms epithelial cells into a mesenchymal migratory phenotype in both 2D and 3D environments. a** Scatter plots comparing the estimated PM tension of MCF10A cells treated with the indicated RNAi. $n = 60$ (si-Control), $n = 40$ (si-RHOA), $n = 28$ (si-ERM), and $n = 25$ (si-SLK+STK10) cells pooled from three independent experiments. Mean ± SD. **b** Confocal images of MCF10A cells treated with the indicated RNAi, stained with anti-pERM antibodies and phalloidin. Yellow arrowheads indicate actin-based protrusions. Scale bars, 10 μm. **c** Quantification of (**b**). Membrane/cytoplasm intensity ratio of pERM and F-actin of $n = 29$ (si-Control), $n = 26$ (si-RHOA), $n = 19$ (si-ERM), and $n = 28$ (si-SLK+STK10) cells pooled from three independent experiments. Mean ± SD. **d** Phase-contrast images of MCF10A cells treated with the indicated RNAi grown in 3D on-top culture. Images are representative of three independent experiments with similar results. Scale bar, 20 μm. **e** Images (left) of siRNA-treated MCF10A cells that invaded through Matrigel and their quantified results (right). $n = 9$ fields from three independent experiments. Mean ± SD. *$P = 0.0471$. **f** Quantification of siRNA-treated MCF10A cells migrated through 8 μm pores. $n = 9$ fields from three independent experiments. Mean ± SD. **$P = 0.0011$. **g** Quantification of 3D migration phenotypes of $n = 155$ (si-Control), $n = 126$ (si-RHOA), $n = 128$ (si-ERM), and $n = 123$ (si-SLK + STK10) cells from three independent experiments. See also Supplementary Fig. 2g and Supplementary Movie 3. **h** Confocal images of the indicated RNAi-treated MCF10A cells stained with anti-pERM antibodies and phalloidin in 3D. Yellow arrowheads indicate actin-based protrusions. Scale bars, 10 μm. **i** Quantification of (**h**). Membrane/cytoplasm intensity ratio of pERM and F-actin of $n = 20$ (si-Control), $n = 14$ (si-RHOA), $n = 15$ (si-ERM), and $n = 18$ (si-SLK + STK10) cells pooled from three independent experiments. Mean ± SD. Significance tested using the two-tailed Mann–Whitney test (**a, c, i**), two-tailed Student's *t*-test (**e, f**), and chi-square test (**g**). ****$P < 0.0001$.

shapes with no protrusion formation (Fig. 4e), reminiscent of epithelial cell characteristics. Indeed, time-lapse movies showed that parental and ezrin-expressing MDA-MB-231 cells exhibited elongated mesenchymal motility (approximately 20%), rounded amoeboid motility, and phenotypic switching between the two modes (mixed phenotype; 50%) (Fig. 4f, g, Supplementary Fig. 4d and Supplementary Movie 4), as previously reported[19]. In contrast, the majority of MA-ezrin-expressing cells displayed rounded shapes with non-migratory behavior (80%) and markedly reduced migration velocity (over 4-fold slower) (Fig. 4f, g, Supplementary Fig. 4e and Supplementary Movie 4). These data demonstrate that increasing PM tension is sufficient to suppress both two different modes of 3D migration.

To further investigate whether maintaining high PM tension plays an active role in suppressing cancer cell dissemination, we employed an orthotopic mouse model of human breast tumor formation, local invasion, and spontaneous metastasis. When injected into the mammary fat pad, MA-ezrin-expressing MDA-MB-231 cells showed reduced tumorigenic ability and produced significantly smaller tumors (Fig. 4h and Supplementary Fig. 4f), although they proliferated at rates comparable to that of parental cells in vitro (Supplementary Fig. 4a). In addition, in contrast to control tumors, which had a prominent invasive behavior with numerous cancer cells invading into the surrounding adipose tissue, MA-ezrin-expressing tumors displayed a clear borderline between the region of tumor cells and the adjacent tissue, showing a significant reduction in invasiveness (Fig. 4i, j). Moreover, MA-ezrin cells exhibited a marked reduction in spontaneous (Fig. 4k) and experimental (tail vein injection) lung metastasis (Fig. 4l, m). Taken together, these data indicate that PM tension sustained by MCA acts as a potent tumor suppressor that inhibits tumorigenesis, invasion, and metastasis.

**Homeostatic PM tension suppresses cancer cell motility by counteracting BAR proteins.** Next, we examined the mechanisms by which PM tension inhibited cancer cell motility. We previously demonstrated that FBP17, a BAR domain-containing protein regulating membrane curvature, acts as a sensor of PM tension involved in actin-based directed migration[32]. Consistently, a mathematical model indicated that the polymerizing ability of BAR proteins is intrinsically dependent on membrane tension[49], suggesting a universal tension-sensing mechanism through BAR proteins. Thus, we performed an short interfering RNA (siRNA) screen for BAR proteins upon ERM knockdown to examine whether BAR proteins play a critical role in the low PM tension-induced invasive phenotype in 3D on-top culture (Fig. 5a and Supplementary Fig. 5a). We identified several BAR proteins, including MTSS1L (also known as ABBA) (Fig. 5a, blue bars). We also noted that knockdown of Toca proteins,

such as FBP17 and CIP4, modestly reduced the elongated invasive behavior induced by ERM depletion, suggesting their functional redundancy as previously reported[50]. In fact, their simultaneous depletion led to increased suppression of the invasive phenotype (FBP17 + CIP4 + Toca-1, triple KD) (Fig. 5a). Of the identified proteins, MTSS1L and Toca proteins were also required for invasion of MDA-MB-231 cells (Supplementary Fig. 5b) or MCF10A cells induced by the depletion of ERM proteins or RHOA (Supplementary Fig. 5c). Thus, in subsequent analyses, we focused on these proteins. MTSS1L and Toca proteins are implicated in lamellipodia formation or membrane ruffling through the activation of Arp2/3 complex-dependent actin nucleation[51,52], suggesting that these proteins may play a role in low-tension-mediated actin-based protrusions. Indeed, we found that knocking down these BAR proteins suppressed an elongated invasive phenotype driven by Snail overexpression (Fig. 5b). We further examined the role of BAR proteins in 3D cancer cell motility. Time-lapse movies showed that the depletion of MTSS1L or Toca proteins suppressed both mesenchymal and amoeboid motility; the majority of cells exhibited a non-motile phenotype with rounded shapes and threefold slower migration speeds (Fig. 5c, d, Supplementary Fig. 5d and Supplementary Movie 5). Consistently, the deletion of MTSS1L or Toca proteins resulted in not only the inhibition of actin-based protrusions, but also the formation of non-polarized membrane blebs (Fig. 5e and Supplementary Fig. 5e).

These data suggest that low-tension-mediated mechanosignaling by BAR proteins plays a pivotal role in cancer cell motility, whose mechanisms are normally suppressed by homeostatic PM tension in non-motile epithelial cells. Indeed, GFP-FBP17 mainly exhibited cytoplasmic distribution in MCF10A cells; however, the knockdown of ERM, their kinases, or Snail expression, resulted in the accumulation of GFP-FBP17 at the PM (Fig. 5f). In contrast, in MDA-MB-231 cells expressing ezrin, GFP-FBP17 spontaneously accumulated in both actin- and bleb-based membrane protrusions in 2D (Supplementary Fig. 5f, g) and 3D environments (Fig. 5g). However, no prominent accumulation of GFP-FBP17 was observed in high-tension MA-ezrin-expressing cells (Fig. 5g and Supplementary Fig. 5f, g). Similar results were obtained with GFP-MTSS1L (Supplementary Fig. 5g, h). Moreover, increasing PM tension by MBCD treatment was also sufficient to prevent the recruitment of FBP17 to the PM (Supplementary Fig. 5i, j). To further investigate whether PM tension directly controls the assembly of BAR proteins, we employed a cell-stretching device to increase tension by modulating the in-plane membrane tension[33]. Mechanical stretching of the PM led to rapid disassembly of FBP17 or MTSS1L, accompanied by the disappearance of the leading edges (Supplementary Fig. 5k, l). These data indicate that homeostatic

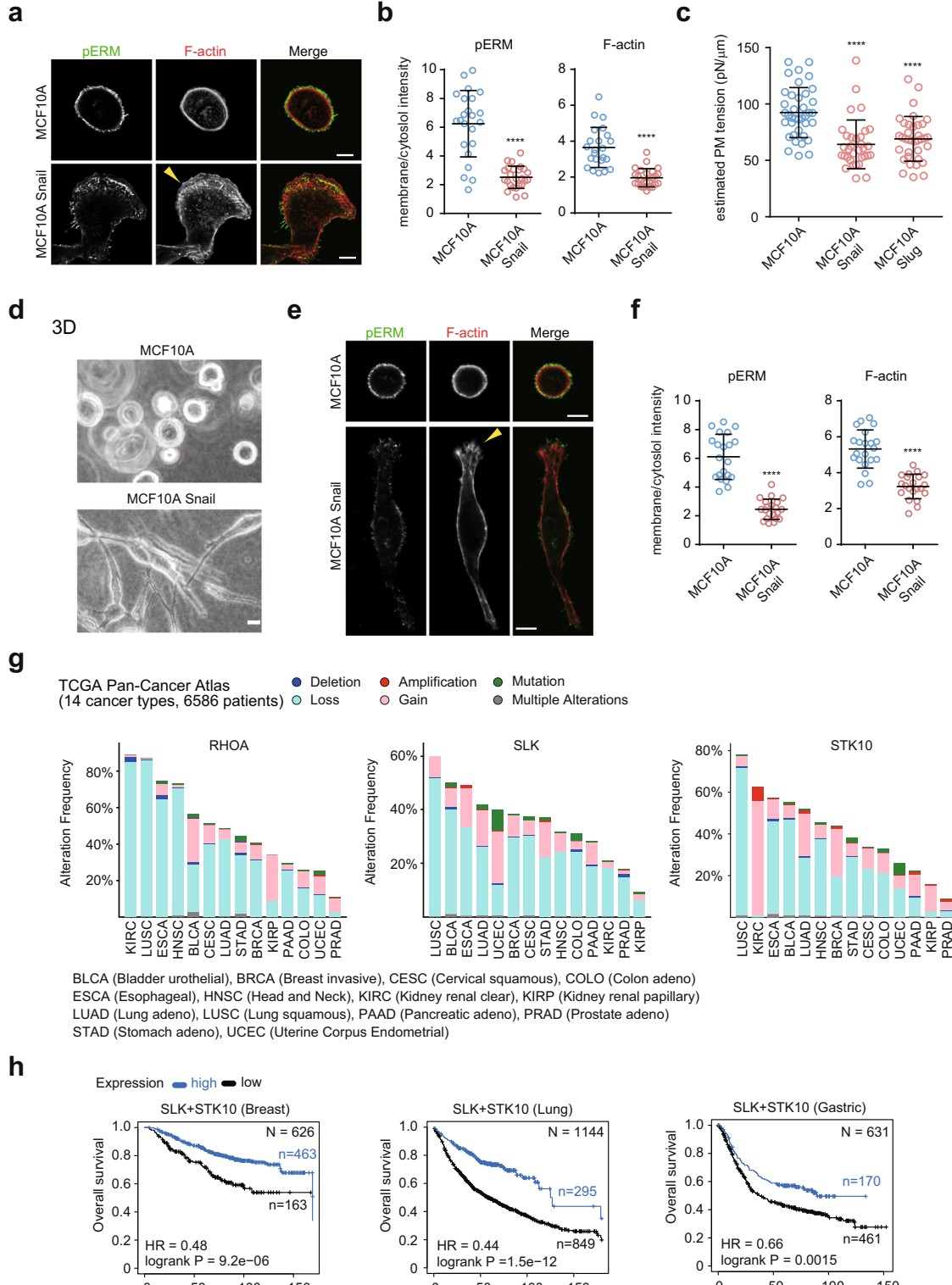

PM tension acts as a mechanical tumor suppressor that inhibits cancer cell dissemination by counteracting membrane-mediated mechanotransduction by BAR proteins.

## Discussion

It has long been considered that cell mechanics are inherently associated with invasion and metastasis. However, how such mechanical changes influence tumor dissemination ability at the cellular and molecular levels has remained elusive owing to a lack of understanding of the key physical parameters underlying the malignant phenotype. Here, we show that metastatic cells exhibit significantly lower PM tension than epithelial cells, and that this mechanical property is closely associated not only with membrane protrusion, but also with tumor invasion and metastasis. Our data further indicate that this decreased MCA-based tension is translated into Arp2/3 complex-dependent actin polymerization via self-assembly of BAR proteins, such as MTSS1L and Toca

**Fig. 3 Correlation between decreased PM tension and malignant progression. a** Confocal images of MCF10A or Snail-expressing cells stained with anti-pERM antibodies and phalloidin. The yellow arrowhead indicates actin-based protrusion. Scale bars, 10 μm. **b** Quantification of (**a**). Membrane/cytoplasm intensity ratio of pERM and F-actin of $n = 23$ (MCF10A) and $n = 25$ (Snail-expressing cells) cells pooled from three independent experiments. Mean ± SD. **c** Scatter plots comparing the estimated PM tension of the indicated cells. $n = 38$ (MCF10A), $n = 33$ (Snail-expressing cells), and $n = 36$ (Slug-expressing cells) cells pooled from three independent experiments. Mean ± SD. **d** Phase-contrast images of MCF10A cells or Snail-expressing cells in a 3D collagen matrix. Images are representative of three independent experiments with similar results. Scale bar, 20 μm. **e** Confocal images of MCF10A or Snail-expressing cells stained with anti-pERM antibodies and phalloidin in 3D. The yellow arrowhead indicates actin-based protrusion. Scale bars, 10 μm. **f** Quantification of (**e**). The membrane/cytoplasm intensity ratio of pERM and F-actin of $n = 21$ (MCF10A) and $n = 21$ (Snail-expressing cells) cells pooled from three independent experiments. Mean ± SD. **g** Genetic alterations of RHOA, SLK, and STK10 across 14 carcinoma types in The Cancer Genome Atlas (TCGA) data (6586 samples). **h** Kaplan–Meier plots showing the overall survival of breast, lung, and gastric cancer patients, which were stratified according to the mRNA expression of SLK+STK10. Significance tested using the two-tailed Mann–Whitney test (**b**, **c**, **f**) and two-tailed log-rank test (**h**). ****$P < 0.0001$.

proteins, promoting cancer cell migration and invasion (Fig. 6a, b). Recent studies have shown that a variety of BAR proteins play an active role in invasion and metastasis of various cancers[53–56]. An important aspect of our findings is that invasive behaviors, irrespective of the type of protrusions and motile modes, can be phenotypically normalized by simply increasing MCA-based PM tension. This is consistent with recent studies, showing that membrane-actin detachment is key to both actin- and bleb-based protrusions[57,58]. An in vivo study reported that the loss of moesin (the sole *Drosophila* ERM protein) alone is sufficient to induce invasion in *Drosophila*[59]. Furthermore, analysis of clinical samples showed that ERM proteins commonly exhibit cytoplasmic distribution in malignant tumors, including breast, lung, and head and neck squamous cell carcinoma[60–63]. Therefore, our study, together with these observations, supports a general role for MCA-based PM tension as a mechanical tumor suppressor.

A limitation of our study is that we cannot exclude the possibility that MA-ezrin expression could have effects other than PM tension, as ERM proteins are known to be involved in the regulation of a variety of signaling molecules[35]. A recent study reports the development of a synthetic molecular tool (iMC-linker) to manipulate MCA by simply connecting the cell membrane to the actin cortex[64]. Interestingly, two recent complementary studies using this tool and constitutively active ezrin show that stem cell spreading, correlated with cell differentiation, is inhibited by increased MCA-based tension[64,65]. Future studies using this tool will be needed to support the importance of ERM-mediated MCA in inhibiting cancer cell motility.

Our data indicate that ERM-mediated MCA is responsible for maintaining homeostatic PM tension in non-invasive cells. However, changes in the actin structure itself are also considered to affect PM tension[22,23]. Migrating cells are characterized by less stabilized F-actin and significantly enhanced actin filament turnover rates primarily mediated by cofilin[66], which contributes to dynamic protrusion formation by providing G-actin monomers[67]. Such accelerated depolymerization may also lead to changes in PM tension, thereby synergistically promoting cancer cell migration. This raises the question of how changes in PM tension lead to two different protrusion formations and subsequent migration modes. In epithelial cells, an overall reduction in MCA appeared to induce only slow mesenchymal-like motility. This indicates that a local increase in MCA, especially at the cell rear, may be required for fast migration modes[57], such as amoeboid migration[68]. In addition, experimental studies and mathematical considerations suggest that decreased PM tension and increased cortical tension favor bleb formation and thus bleb-based migration[26,31], in accordance with our tether force data and previous cortical tension measurements[69]. Therefore, a decrease in PM tension may be a prerequisite for both types of protrusion formation, and cortical tension is important for their switching and subsequent migration

modes[17,46,69], reflecting why maintaining high PM tension efficiently suppresses both migration modes. Future studies examining the mechanical relationship between these two forces in migratory behaviors, particularly in 3D environments, will advance our understanding of how cell mechanics control cancer cell migration.

Recent studies have shown that the disruption of cell–cell adhesion in epithelial cells does not necessarily lead to single-cell dissemination[70]. Our study suggests that homeostatic PM tension is a cell-autonomous characteristic of non-motile cells, which may partially explain the above phenomenon. In addition, cancer cells are known to exploit collective migration characterized by multicellular coordination through cell–cell adhesion for dissemination[16]. Such collective processes are mechanically mediated by the coordination between the cell adhesion forces and actomyosin contractility[71]. It will be interesting to investigate how PM tension integrates with these forces to control collective migration and whether manipulating PM tension could suppress this type of movement.

An unexpected finding was that homeostatic PM tension may also play an active role in hindering tumor formation and growth. Interestingly, it has been suggested that changes in cell surface mechanics appear to be correlated with cancer stemness[72,73]. Moreover, a recent study suggests a direct link between membrane protrusions and cancer progression[74]. Our data indicate that such membrane fluctuations driven by EMT or oncogenic transformation can be explained by decreased PM tension, suggesting that maintaining high PM tension may also function as an effective mechanism to suppress tumorigenicity. It will be interesting to further explore the link between PM tension and cancer progression, especially in the context of cancer stemness.

Our data showed that epithelial cells have a PM tension of ~100 pN/μm that is comparable with the membrane tension at which BAR domain self-assembly is affected, as determined by reconstitution experiments[75] and mathematical models[49], suggesting a threshold tension for BAR protein assembly. We propose that PM tension above a critical threshold inhibits the self-organization of BAR proteins, thereby suppressing branched actin assembly and subsequent actin-based migration. This inhibitory mechanism could also partially explain why increased PM tension suppresses tumor development, as some BAR proteins are reported to play an active role in tumorigenesis[76,77]. Our unexpected finding was that BAR proteins are also required for bleb-based movement. Our knockdown experiments suggested that BAR proteins are not required for the formation of membrane blebs, but are involved in their polarization. This may be relevant to a recent study that reported that local membrane invaginations driven by BAR proteins enable local membrane protrusions essential for directed bleb-based migration[78]. Importantly, because PM tension regulated by MCA should serve as a local parameter[21,28], local membrane undulations upon its detachment

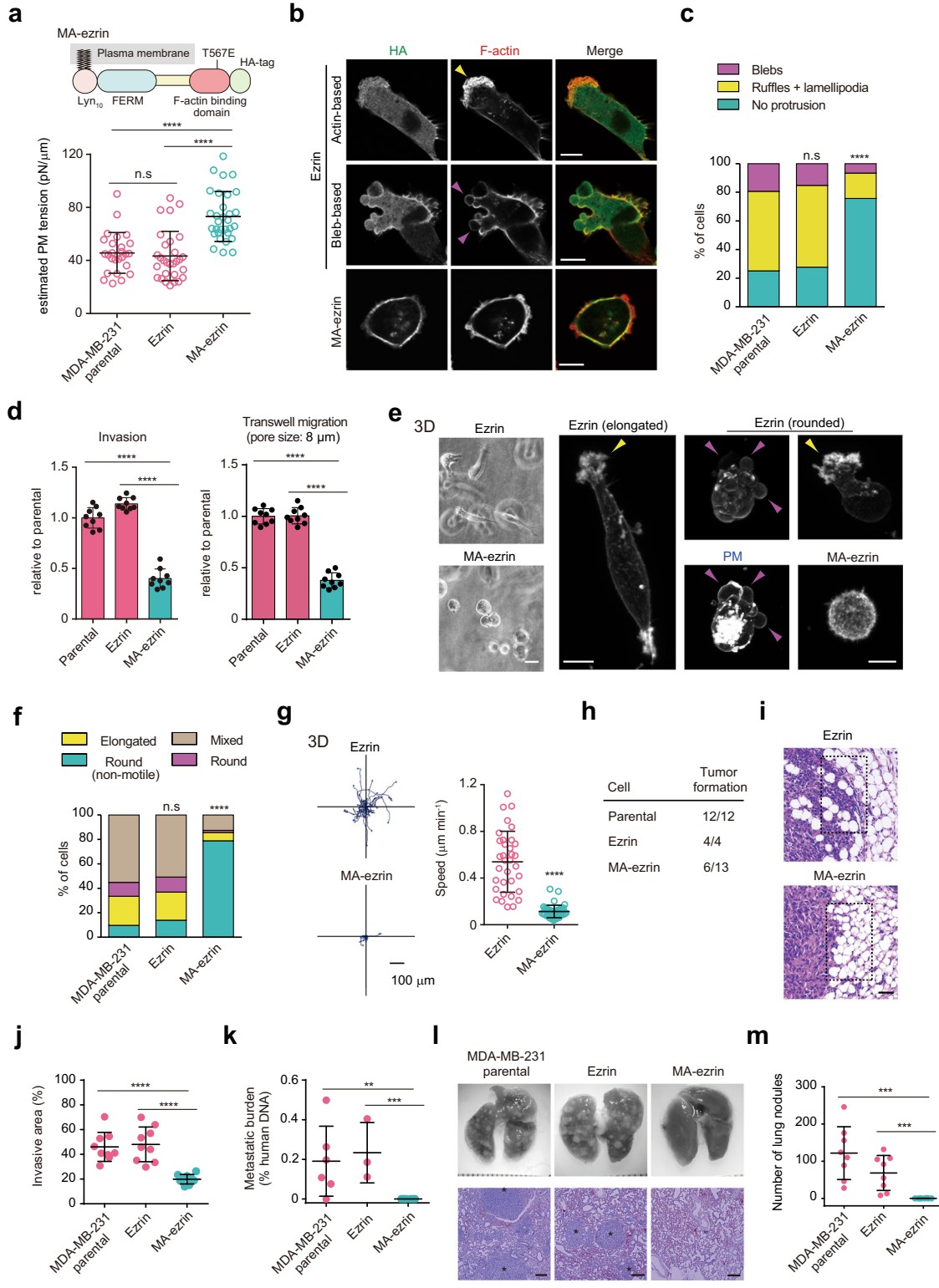

may recruit curvature-sensitive BAR proteins, driving both polarized actin- and bleb-based motility. Overall, these observations suggest that the low PM tension–BAR protein axis may serve as a general form of membrane-mediated mechanosignaling that drives cancer cell migration, re-emphasizing PM tension as a promising target for limiting tumor dissemination.

Abnormal changes in the cell membrane shape, including the formation of microvesicles/exosomes and micropinocytosis, are hallmarks of cancer. It is tempting to speculate that these functions could be directly controlled by PM tension. Our findings provide a foundation for future investigations into whether MCA manipulation can be exploited for therapeutic interventions aimed at normalizing cell membrane mechanics.

## Methods

**Cell culture**. Human non-tumorigenic mammary epithelial cells (MCF10A), human breast cancer cells (MCF7, AU565, MDA-MB-231, and Hs578T), and human pancreatic (PANC-1) cancer cells were purchased from the American Type

**Fig. 4 Increasing PM tension is sufficient for the suppression of 3D migration and metastasis. a** Upper, schematic outline of membrane-anchoring active ezrin. Lower, scatter plots comparing the estimated PM tension of the indicated cells. $n = 26$ (parental), $n = 29$ (ezrin), and $n = 31$ (MA-ezrin) cells pooled from three independent experiments. **b** Confocal images of the indicated cells stained with anti-HA antibodies and phalloidin. Scale bars, 10 μm. **c** Quantification of the protrusions of $n = 207$ (Parental), $n = 224$ (ezrin), and $n = 214$ (MA-ezrin) cells from three independent experiments. **d** Quantification of the migration or invasion rates of the indicated cells. $n = 9$ fields from three independent experiments. **e** Phase-contrast images (left) or maximum intensity projections (right; stained with phalloidin and WGA [blebbing cell]) of the indicated cells in 3D. Scale bars, 20 μm (left) and 10 μm (right). **f** Quantification of the 3D migration phenotypes of $n = 176$ (Parental), $n = 187$ (ezrin), and $n = 175$ (MA-ezrin) cells from three independent experiments. **g** Trajectories of cell centroids of the indicated cells tracked in (**f**) for 8 h. Right, the average speed of one cell over the course of 8 h. $n = 34$ (ezrin) and $n = 44$ (MA-ezrin) cells pooled from three independent experiments. **h** Tumor formation after injection of the indicated cells into the mammary fat pad. **i** Representative hematoxylin and eosin (H&E)-stained sections of the primary tumor and surrounding tissue of mice injected with the indicated cells. Scale bar, 50 μm. **j** Quantification of (**i**). Tumor invasive area in dashed boxes located at the tumor rim was quantified. $n = 9$ areas for three tumors per group. **k** Quantification of spontaneous lung metastasis by quantitative PCR. $n = 6$ mice (parental), $n = 3$ (ezrin), and $n = 6$ mice (MA-ezrin). \*\*$P = 0.0152$; \*\*\*$P = 0.0119$. **l** Whole images of the lungs and H&E staining of lung sections (bottom) after tail vein injection of the indicated cells. Scale bars, 250 μm. **m** Quantification of (**l**). $n = 8$ mice per group. \*\*\*$P = 0.0002$. In (**b**) and (**e**), yellow and magenta arrowheads indicate actin- and bleb-based protrusions, respectively. All data were expressed as mean ± SD. Significance tested using the two-tailed Mann–Whitney test (**a**, **g**, **j**, **k**, **m**), two-tailed Student's $t$-test (**d**), and chi-square test (**c**, **f**). n.s. not significant; \*\*\*\*$P < 0.0001$.

Culture Collection (ATCC). Normal rat liver (IAR-2) epithelial cells and human prostate (PC-3) cancer cells were obtained from the Japanese Collection of Research Bioresources (JCRB) Cell Bank. Normal canine kidney (MDCK II) epithelial cells were kindly provided by M. Murata (University of Tokyo). MDCK II cells carrying doxycycline-inducible RasV12 were previously described[79]. MCF10A cells were cultured in DMEM/F12 (Invitrogen) supplemented with 5% horse serum (Gibco), EGF (20 ng/ml; R&D Systems), insulin (10 μg/ml; Sigma-Aldrich), cholera toxin (80 μg/ml; Wako), and hydrocortisone (0.5 μg/ml; Sigma-Aldrich). AU565 cells were cultured in RPMI-1640 (Nacalai Tesque) supplemented with 10% FBS (Sigma-Aldrich). Other cell lines were cultured in DMEM (Nacalai Tesque) supplemented with 10% FBS (Sigma-Aldrich). All cell lines were cultured at 37 °C in 5% CO$_2$. All cell lines were regularly tested for mycoplasma contamination using a PCR-based mycoplasma detection kit (Venor GeM Classic; MB minerva biolabs).

For 3D culture, cells were incorporated into 3D collagen lattices (type-I bovine collagen; final concentration, 1.67 mg/ml; Advanced BioMatrix). For 3D on-top culture, cells were grown in a mix of Matrigel:collagen, as previously described (https://brugge.med.harvard.edu/Protocols). We used growth factor-reduced Matrigel (BD Biosciences).

**Tether force measurement with optical tweezers and the estimation of PM tension.** Tether force measurements were performed with an optical tweezers system (NanoTrackerTM 2, JPK Instruments) equipped with an infrared (IR) laser source (3 W, 1064 nm) on an Olympus IX-73 inverted microscope with a ×60 (numerical aperture = 1.2) water immersion objective (Olympus). Silica beads (1.5 μm diameter, Polysciences) were incubated with concanavalin A (Sigma-Aldrich) at 1 mg/ml for 1 h. Cells were plated onto glass-bottom dishes (WPI). Coated beads were added to the cell culture medium supplemented with 25 mM HEPES (pH 7.5), and experiments were performed at 37 °C. Tether force ($F$) can be calculated using Hooke's law: $F = k\Delta x$, where $k$ is the stiffness of the trap and $\Delta x$ is the displacement of the bead from the trap center. Trap stiffness ($k$, typically ~0.15 pN/nm) was calibrated for each experiment by a power spectrum analysis[80]. A single bead was captured by the optical trap, kept in contact with the PM for 500 ms, then pulled away to form a membrane tether (5 μm length) by moving a piezo stage under computer control at 1 μm/s. The displacement of the bead in the trap center ($\Delta x$) was determined by a quadrant photodiode detector with high precision (the resolution less than 1 nm). Data analysis was performed using the JPK data processing software. To compare PM tension between non-motile epithelial cells and malignant cells, we measured the tethering force on the lateral sides of cells. We noted that cells with low PM tension, like malignant cells, frequently formed double tethers, which exhibited a twofold increase in the tether force. This may be due to their low tension. Therefore, we carefully confirmed whether one tether was formed and excluded data from double tethers. We also noted that the membrane tether tended to extend longer in metastatic cells (>15 μm), as opposed to that in epithelial cells whose tethers were easy to break.

PM tension can be estimated with the following formula[21,33,81]: $T = F_{\text{tether}}^2 / 8B\pi^2$, where $T$ is the apparent plasma membrane tension (PM tension), $F_{\text{tether}}$ is the tether force measured by optical tweezers, and $B$ is the bending stiffness of the membrane. Given that $B$ is known to be relatively invariable between the cell types tested (from $1{-}3 \times 10^{-19}$ N m)[31], we used $B$ ($1.4 \times 10^{-19}$ N m)[81] to calculate PM tension.

**Materials.** Methyl-β-cyclodextrin (MβCD) (Sigma-Aldrich) and calyculin A (Cell Signaling Technology) were used at a final concentration of 4 and 0.5 nM, respectively. The following antibodies were used: anti-ERM (rabbit polyclonal, 1:1000 for immunoblotting; #3142, Cell Signaling Technology [CST]); anti-phospho-ERM (rabbit monoclonal, 1:100 for immunostaining; #3726, CST); anti-RHOA (mouse monoclonal, 1:1000 for immunoblotting; #sc-418, Santa Cruz

Biotechnology); anti-pS19MLC (Phospho-myosin light chain 2 [Ser19]) (mouse monoclonal, 1:1000 for immunoblotting; #3675, CST); anti-MLC (rabbit polyclonal, 1:1000 for immunoblotting; #3672, CST); anti-E-cadherin (rabbit monoclonal, 1:1000 for immunoblotting; #3915, CST); anti-vimentin (rabbit monoclonal, 1:1000 for immunoblotting; #5741, CST); anti-MTSS1L (rabbit polyclonal, 1:200 for immunoblotting; #NBP2-57037, Novus Biologicals); anti-FBP17 (FNBP1)[82] (rabbit polyclonal, 1:1000 for immunoblotting); anti-CIP4 (TRIP10) (mouse monoclonal, 1:1000 for immunoblotting; #612556, BD Transduction Laboratories); anti-HA-Tag (rabbit monoclonal [C29F4], 1:100 for immunostaining; #3724, CST), and anti-β-actin (rabbit polyclonal 1:2000 for immunoblotting; #PM053, MBL). Alexa-Fluor-488-conjugated secondary antibody (1:500 for immunostaining; rabbit, #A11034; mouse, #A11029) was obtained from Thermo Scientific.

Human Snail and Slug were subcloned into pMXs-IRES-Puro retroviral vector (Cell Biolabs, Inc.; modified by introducing an HA tag to the C-terminus). Lyn$_{10}$-ezrinT567E (MA-ezrin) was constructed by the fusion of the PM targeting signal (MGCIKSKRKD, a myristoylation motif derived from Lyn tyrosine kinase) to the N-terminus of human ezrin, followed by the generation of the T567E mutation using PCR primers and the sequence was confirmed. Ezrin and MA-ezrin constructs were subcloned into the pQCXIN-HA retroviral vector (Clontech; modified by introducing an HA tag to the C-terminus) with a neomycin-resistant gene. Human MTSS1L was subcloned into the pEGFP C-1 vector. Human GFP-FBP17 was previously described[32]. For retrovirus infection, cells were plated in a six-well plate and incubated with viruses in the presence of 4 μg/ml polybrene (Sigma-Aldrich). Infected cells were selected with G418 (0.8 mg/ml) or puromycin (1.5 μg/ml). Transgene expression was assessed by western analysis and confocal microscopy.

**siRNA and transfection.** For knockdown experiments, Dharmacon SMARTpool-ON-TARGETplus siRNAs (a mixture of four different siRNAs; Thermo Scientific) against human genes were used: EZR/ezrin (L-017370-00); RDX/radixin (L-011762-00); MSN/moesin (L-011732-00); RHOA (L-003860-00); SLK (L-003850-00); STK10 (L-004168-00); ARHGAP4 (L-003628-00); ARHGAP10/GARF2 (L-009382-01); ARHGAP17/RICH1 (L-008335-02); ARHGAP26/GRAF1 (L-008426-00); ARHGAP29 (L-008277-00); ARHGAP42/GRAF3 (L-026507-01); ARH-GAP44/RICH2 (L-009238-01); ARHAGP45/HMHA1 (L-023893-00); ARHGEF37 (L-032927-01); ARHGEF38 (L-020676-00); IRSp53/BAIAP2 (L-012206-02); BAIAP2L1/IRTKS (L-018664-02); DNMBP/Tuba (L-026304-01); FCHSD1 (L-015107-02); FCHSD2 (L-021240-01); FER (L-003129-00); FES (L-003130-00); FNBP1/FBP17 (L-026214-02); FNBP1L/Toca-1 (L-020718-01); GAS7 (L-011492-00); GIMP (L-021160-01); MTSS1/MIM (L-018506-00); MTSS1L/ABBA (L-022582-01); OPHN1/Oligophrenin 1 (L-009444-00); PACSIN1 (L-007735-00); PACSIN2 (L-019666-02); PACSIN3 (L-015343-00); SH3BP1 (L-009546-01); SRGAP1 (L-026974-00); SRGAP2 (L-021531-02); SRGAP3 (L-014175-00); and TRIP10/CIP4 (L-012685-00). ON-TARGETplus Non-Targeting siRNA Pool (D-001810-10) was used as a control siRNA. RNAs (25 nM) were transfected into cells with Lipofectamine RNAi MAX (Invitrogen). Analyses were performed 72 h after transfection. We confirmed the knockdown of key proteins (ERM proteins, RHOA, SLK, STK10, MTSS1L, FBP17, and CIP4) by western analysis. Plasmid transfections were performed using FuGENE HD (Roche) according to the manufacturer's protocol. Transfected cells were examined after 24 h.

**In vitro invasion and migration assays.** For invasion and migration assays, we used BioCoat Matrigel Invasion Chambers (Corning) and 8.0 μm PET transmigration inserts (Corning), respectively. In the invasion assay, $1 \times 10^5$ cells were suspended in serum-free medium and seeded on top of the chamber's membrane, in which medium containing serum was placed at the bottom. For MβCD or calyculin A treatment, drugs were added in both sides. Low-invasive epithelial

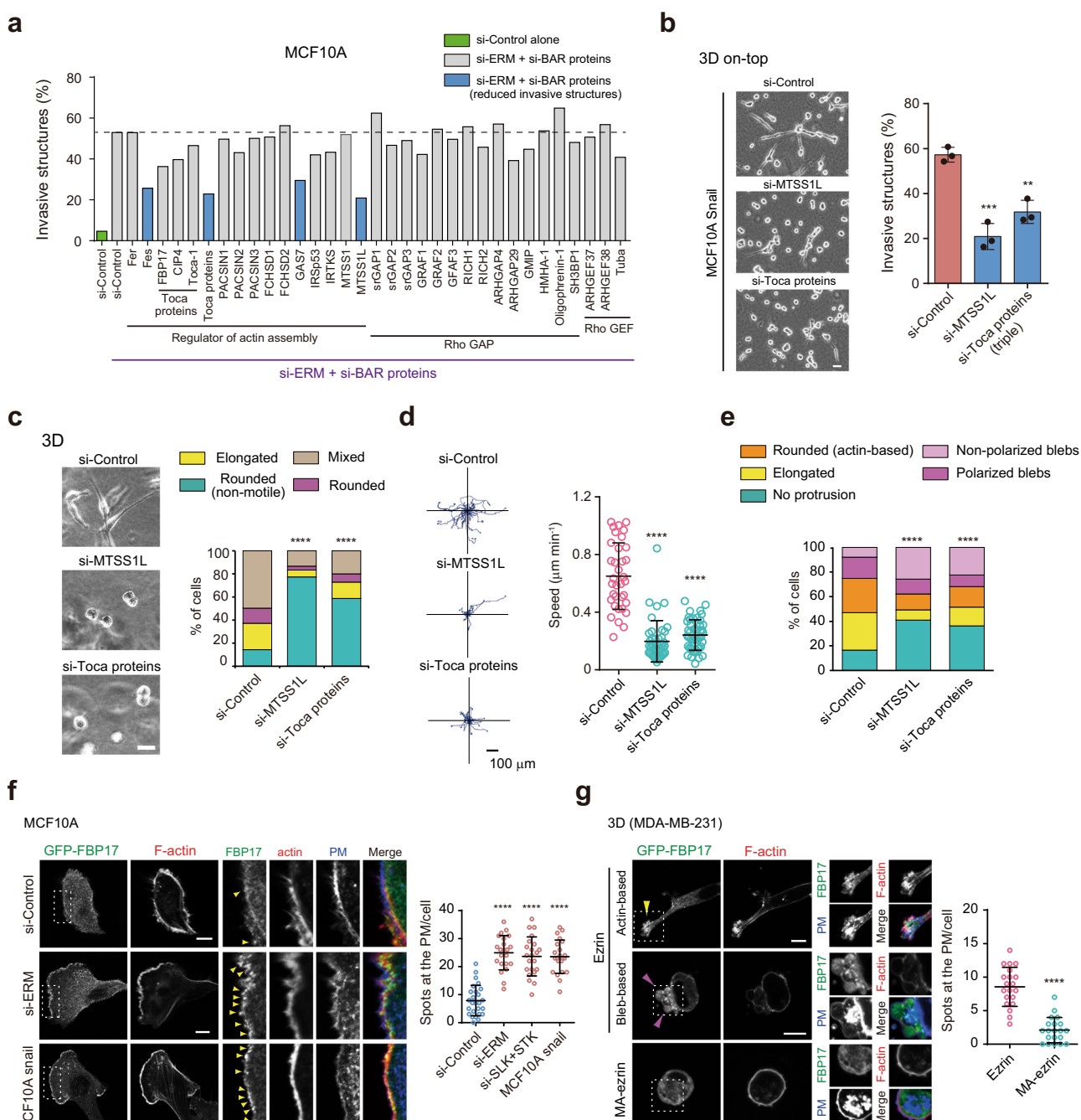

**Fig. 5 Homeostatic PM tension suppresses cancer cell migration by counteracting BAR proteins. a** Fraction of MCF10A spheroids with invasive structures grown in 3D on-top culture treated with the indicated RNAi. Control siRNA alone and siRNAs targeting BAR proteins that reduce invasive structures induced by ERM deletion are shown in green and blue, respectively. Data are mean of two independent experiments with at least 50 cells per experiment. **b** Left, representative images of the indicated cells in 3D on-top culture. Right, fraction of the invasive structures of the indicated cells. Data are mean ± SD of three independent experiments with at least 200 cells per experiment. Scale bar, 20 μm. **P = 0.002; ***P = 0.0007. **c** Phase-contrast images of MDA-MB-231 cells treated with the indicated RNAi in 3D. Scale bar, 20 μm. Right, quantification of 3D migration phenotypes of n = 153 (si-Control), n = 150 (si-MTSS1L), and n = 155 (si-Toca proteins) cells from three independent experiments. **d** Trajectories of cell centroids of the indicated cells tracked in **c** for 8 h. Right, the average speed of one cell over the course of 8 h. n = 35 (si-Control), n = 43 (si-MTSS1L), and n = 46 (si-Toca proteins) cells pooled from three independent experiments. Mean ± SD. **e** Quantification of protrusions of the indicated cells in 3D. n = 151 (si-Control), n = 132 (si-MTSS1L), and n = 138 (si-Toca proteins) from three independent experiments (see also Supplementary Fig. 5e). **f** Confocal images of the indicated cells expressing GFP-FBP17 stained with phalloidin and WGA. Yellow arrowheads indicate GFP-FBP17 spots at the PM. Scale bars, 10 μm. Right, quantification of GFP-FBP17 puncta of n = 26 (si-Control), n = 22 (si-ERM), n = 22 (si-SLK+STK10), and n = 22 (Snail-expressing cells) cells pooled from three independent experiments. **g** Confocal images of the indicated cells stained with phalloidin and WGA in 3D. Yellow and magenta arrowheads indicate FBP17 accumulation at actin- and bleb-based protrusions, respectively. Scale bars, 10 μm. Right, quantification of GFP-FBP17 puncta of n = 20 (ezrin) and n = 20 (MA-ezrin) cells pooled from three independent experiments. All data, except for **a**, were expressed as mean ± SD. Significance tested using the two-tailed Student's t-test (**b**, **f**, **g**), two-tailed Mann–Whitney test (**d**), and chi-square test (**c**, **e**). ****P < 0.0001.

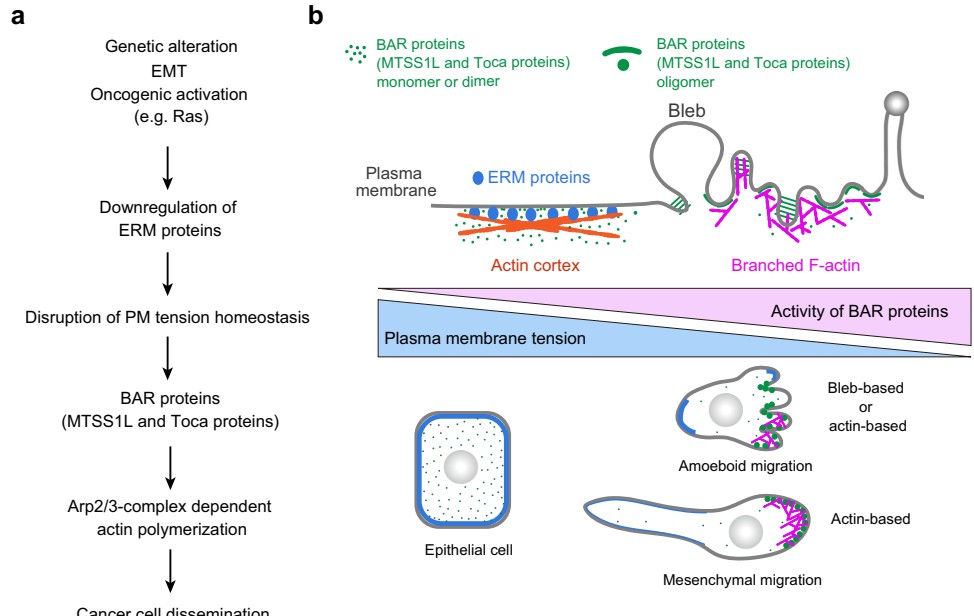

**Fig. 6 Proposed model describing how homeostatic PM tension acts as the mechanical suppressor of cancer cell dissemination. a** Proposed model to describe how cancer progression is linked to the disruption of homeostatic PM tension, leading to cancer cell dissemination via BAR proteins. **b** Homeostatic PM tension sustained by membrane-to-cortex attachment (MCA) can maintain a non-motile state by suppressing the assembly of BAR proteins, key regulators of both actin- and bleb-based protrusions.

(MCF10A, AU565, and MCF7) cells and MDA-MB-231 cells were incubated for 36 and 24 h, respectively, and then fixed in 4% formaldehyde. Invaded cells were imaged using a BZ-X700 microscope (Keyence) with ×10 magnification and counted. In the migration assay, $5 \times 10^4$ cells were seeded on top of the PET membrane, incubated for 24 h, and analyzed as described for the invasion assay. For the quantification of invasive structures grown in 3D on-top culture (Fig. 5a, b), the elongated mesenchymal-like and rounded morphologies were defined as invasive and non-invasive phenotypes, respectively, and quantified. No amoeboid-like movement characterized by a rounded shape was observed in ERM-knocked-down cells or Snail-expressing cells.

**Confocal microscopy, live-cell imaging, and image analysis**. For immunofluorescence analysis, cells were fixed with 4% formaldehyde in phosphate-buffered saline (PBS) for 10 min and permeabilized with 0.2% Triton X-100 for 10 min in PBS. Cells were blocked with 5% goat serum (Sigma-Aldrich) in PBS for 1 h and were incubated with primary antibodies for at least 3 h. Then, cells were incubated with secondary antibodies for 1 h. For membrane visualization, Alexa-Fluor™ 350-conjugated wheat germ agglutinin (WGA) (Thermo Scientific) was incubated with fixed cells for 30 min before permeabilization. For visualization of F-actin, Alexa-Fluor™ 568 Phalloidin (Thermo Scientific) was incubated with fixed cells for 30 min. Fluorescence images were captured using a confocal microscopy system (FluoView 1000-D; Olympus) equipped with 405-, 473-, and 568-nm diode lasers through an objective lens (×60 oil immersion objective, NA = 1.35). For 3D images, Z-stack images of consecutive optical planes spaced by 0.5 μm were acquired for the whole cell. Maximum intensity z-projections shown in Figs. 1d, 4e and Supplementary Fig. 5e were reconstructed using ImageJ and Imaris 8.0.2. All other confocal images were displayed as a single plane. In the 3D data shown in Figs. 1–3, the plane image near the middle of 3D stacks, where the membrane region can be clearly distinguishable, was selected as the representative image. For live imaging with phase-contrast microscopy, cells were mixed with a collagen matrix and plated in eight-well plates (IWAKI). The images were taken using the BZ-X700 microscope (Keyence) with ×20 magnification at 37 °C and 5% $CO_2$. Single cells were manually tracked using Manual Tracking Tool ImageJ software plugin. Migration plots and cell velocities were obtained with Chemotaxis and Migration Tool (Ibidi). To evaluate the cell morphodynamics in 3D, cells were classified as mesenchymal (elongated; aspect ratio >4) or amoeboid phenotypes (rounded with actin- or bleb-based protrusions; aspect ratio <4), as previously described[83]. The aspect ratio was determined as a ratio of the major cell axis length to minor cell axis length, which was calculated automatically by ImageJ. We observed that elongated cells typically have an aspect ratio of 5–8, whereas that of amoeboid cells is 2–3. To calculate the membrane/cytoplasm intensity ratio, the entire membrane region was segmented using the threshold images of the WGA channel and the membrane regions were manually selected using the brush selection tool with a size of 5 pixels (approximately 500 nm in width). The mean intensity of pERM or F-actin all along the membrane regions was then calculated and divided by their mean intensities throughout the cytosol (avoiding the

nucleus). To quantify 3D images, a single plane image (corresponding to the image near the middle of 3D stacks) with clearly distinguishable membrane regions by WGA staining, as shown in Fig. 1d, was used for quantitative analysis as in 2D. Additionally, two images that shifted up and down in the z-direction from the selected middle image by about 1.5 μm (elongated cells) to 4 μm (rounded cells) were quantified to account for the variation in fluorescence intensity due to the thickness of 3D stacks. The average membrane/cytoplasm intensity ratio of all three images was used as one sample. To quantify the accumulation of BAR proteins at the PM, the number of BAR protein spots that merged with the membrane region defined by the membrane marker was quantified.

**Western blotting**. Cell lysates were extracted using Laemmli buffer. The samples were electrophoresed in SDS-PAGE gels, transferred to a polyvinylidene difluoride membrane, blocked with 5% BSA or nonfat dry milk in TBS containing 0.1% Tween 20, incubated with primary antibodies, and then incubated secondary antibodies.

**Proliferation assay**. $2 \times 10^4$ cells were seeded in duplicate for each time-point in six-well plates and counted after 2 and 4 days using a Countess Automated cell counter (Thermo Fisher).

**Analysis of clinical datasets**. TCGA and CCLE datasets were analyzed in the 2020 version of cBioPortal (http://www.cbioportal.org/)[84]. Clinical datasets of cancer patients from the 2020 version of KMplotter (http://www.kmplot.com)[85] were analyzed using each probe (SLK: 206875_s_at and STK10: 228394_at) and the auto-selection best cutoff. The significance of survival differences between groups was assessed by the log-rank test.

**Animal studies**. For tumor formation, local invasion, and spontaneous metastasis assays, $1 \times 10^7$ cells were resuspended in PBS (0.1 ml) and injected into the mammary fat pad of 6-week-old female BALB/c nu/nu mice. Mice were maintained in a temperature (23 ± 1 °C, 55 ± 5% humidity) with a 12/12 h light/dark cycle. The mice were killed 7 weeks after injection, and the tumors and lungs were collected. The tumor volume was calculated according to the formula $V = 1/2 \ (A \times B^2)$, where $A$ and $B$ represent the largest and smallest dimensions of the tumor, respectively. Resected tumors were fixed in 4% paraformaldehyde, embedded in paraffin, and stained with hematoxylin and eosin (H&E). For invasiveness analysis, the borders of the tumor to the adipose tissue were manually defined, and 37,829 μm² quantification boxes were located. In each box, the tumor invasive areas were calculated using ImageJ. To quantify spontaneous metastasis, the ratio of human/mouse DNA was assessed using human-specific quantitative PCR (qPCR), as previously described[86]. Briefly, qPCR was performed using PowerTrack SYBR Green Master Mix (Thermo Fisher Scientific) and 100 ng of lung genomic DNA with human- and mouse-specific PTGER2 primer pairs. Primers used for qPCR are included in Supplementary Table 1. A standard curve was

generated using genomic DNA extracted from MDA-MB-231 cells and xenograft-naive mouse lung. qPCR was performed using the StepOnePlus Real-Time PCR System. For experimental lung metastasis assays, $2 \times 10^6$ cells were resuspended in 0.1 ml PBS and injected intravenously into 6-week-old female BALB/c nu/nu mice, and lung metastasis was assessed after 8 weeks. Lung was fixed and stained using H&E. All sections were examined under the BZ-X700 microscope or the BZ-8000 microscope (Keyence). All animal experiments were reviewed by the Institutional Ethics Committee and performed in compliance with the Guidelines for Laboratory Animal Research of the Tokyo University of Pharmacy and Life Sciences (Tokyo, Japan).

**Mechanical stretch of cells**. Cells were grown on silicon chambers (STB-CH-04, STREX) coated with fibronectin (0.05 mg/ml; Sigma-Aldrich) for 24 h. The chambers were set on the stretching device (STB-100, STREX), and stretched uniaxially for 5 min (20% stretch).

**Statistics**. Statistical analyses were carried out with GraphPad Prism 6 and Excel. D'Agostino Pearson omnibus and Kolmogorov–Smirnov (with the Dallal–Wilkinson–Lillie for $P$ value) tests were used to test datasets for Gaussian distribution. Statistical significance was determined using two-tailed Student's $t$-test or non-parametric Mann–Whitney $U$-test for two groups, and one-way ANOVA with Tukey's test for multiple comparisons. Phenotype distributions were compared using a chi-square test. The sample sizes, number of repeats for each experiment, and specific tests are stated in the figure legends.

**Reporting summary**. Further information on research design is available in the Nature Research Reporting Summary linked to this article.

## Data availability

TCGA and CCLE datasets are available from the cBioPortal[84] (http://www.cbioportal.org/). Clinical datasets of cancer patients are available from KMplot[85] (http://www.kmplot.com). All the other data are available within the article and its Supplementary Information. Source data are provided with this paper.

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

## Acknowledgements

We thank M. Murata for MDCK II cells and N. Shirai for technical assistance. This work was supported by AMED-PRIME from Japan Agency for Medical Research and Development (JP18gm5810007 to K.T.), Grant-in-Aid for Challenging Exploratory Research from Japan Society for the Promotion of Science (JSPS) KAKENHI (16K14725 to K.T.), Grants-in-Aid for Young Scientists (A) (17H05056 to R.S.), Grant-in-Aid for Scientific Research (B) (16H04785 to T.I.), Grant-in-Aid for Scientific Research on Innovative Areas (26114008 to T.I. and Y.F.), and resulted in US Provisional Patent Application No. 63/162,214.

## Author contributions

K.T. conceived the project and wrote the manuscript, and designed, performed, and analyzed experiments. R.S., S.A., and K.F. designed, performed, and analyzed animal experiments. L.A provided key reagents and assisted the cancer genome analysis. Y.N., K.S., Y.F., and T.I. provided key reagents and conceptual advice. All authors contributed with discussion and edited the manuscript.

## Competing interests

The authors declare no competing interests.
