## [Peer Review File · Nature Communications]

Homeostatic membrane tension constrains cancer cell dissemination by counteracting BAR protein assemblyREVIEWER COMMENTS

Reviewer #1 (Remarks to the Author):

Tsujita et al. submitted an article in Nature Communications entitled "Homeostatic membrane tension is an intrinsic mechanical suppressor of cancer cell dissemination". In this manuscript, the authors tested the novel hypothesis that the cell-intrinsic physical factors that maintain epithelial cell mechanics could function as tumor suppressors. They observed that epithelial cells maintain higher plasma membrane tension than their metastatic counterparts, which prevent invasion and metastasis. They demonstrate that in epithelial cells, high tension is maintained by the RHOA/ERM proteins pathway that connects the plasma membrane to the actin cell cortex (membrane-cortex adhesion, MCA). In non-invasive epithelial cells, downregulation of the RHOA/ERM pathway leads to a decrease in tension that triggers the membrane-curvature sensing/generating BAR family proteins to stimulate actin-polymerization powering mesenchymal and amoeboid migration. They overexpressed a plasma membrane-anchored active version of the ERM protein Ezrin (MA-ezrin) in invasive cells and this was sufficient to inhibit both mesenchymal and amoeboid 3D migration. In vivo, expression of MA-ezrin was sufficient to abrogate experimental metastasis. Collectively, these data led the authors to claim that restoring membrane tension in metastatic cells is sufficient to suppress invasion and metastasis.

The manuscript is well written, generally easy to follow, and technically solid. The authors robustly established that manipulating ERM activity influences cell migration in 2D, transwell migration, 3D migration and also experimental lung metastasis of a triple-negative breast cancer cell line. The authors attributed the anti-metastatic properties of the RHOA/ERM pathway to an increase in cell tension, which uncouples the membrane-binding of BAR family proteins that stimulates actin polymerization. There are however major concerns about the conclusions of this study.

Major Comments

1) To claim that restoring tension is sufficient to suppress invasion and metastasis, the authors assumed that the correlation between ERM proteins activity and high tension measured in 2D context was the same in 3D. To address this concern, the authors should think of a 3D compatible method to measure plasma membrane tension. I understand the challenge that it could represent. At a minimum, the author should demonstrate that increasing the plasma membrane tension using an alternative way than upregulating ERM proteins also suppress invasion and metastasis dependent of the BAR family proteins.

2) Another concern is about tether force measurement itself as a read-out of plasma membrane tension. I believe that optical tweezers technic allows measuring the tether force locally. It is mentioned in the method section that measurements were taken on the lateral side of the cell. However, it is very likely that the plasma membrane tension will be different at the rear, the side or the front of a migrating cell. I am not sure that one local measurement is a good read-out of the overall plasma membrane tension. That being said, that would be interesting if the authors could explore other techniques to measure or even visualize membrane tension for example using the newly-developed Fluorescent Membrane Tension Probe. Other approaches would of course be also acceptable to further support the tension measurements.

3) The in vivo assay to probe the contribution of MCA to metastasis is limiting. While the authors confirmed similar cell proliferation on plastic of control and MA-ezrin cells, whether this is also the case in vivo must be tested. The authors conducted an experimental metastasis assay that bypasses the local invasion and intravasation steps that are most relevant to this study. As such, a grafting experiment in mammary fat pads would allow for measuring both tumor growth and metastasis. The authors could also measure the tumor cell blood burden in these assays. This would provide a clear picture of altering membrane tension on cell invasion, intravasation, and colonization. Conversely, it would be highly informative if the authors could carry out an in vivo experiment with a non-metastatic cell model (MCF7?) and test whether depletion of ERMs allows local invasion or intravasation (tumor cell blood burden) or even metastasis to lungs.

4) While I appreciate the in-silico analyses performed by the authors, I wonder if some conclusions

are accurate. The differences in RHOA expression among subtypes remains rather small (3E). Since RHOA is an enzyme, a small decrease in protein levels could easily be compensated by increased activation by a GEF. It would be essential to explore this to make strong conclusions (this may be beyond the scope of this MS, but the concept should still be discussed). The same would be true for ERMs (vs. pERMs). The authors should also quantify the blots presented in Fig 3F-G from multiple experiments. Globally, while interesting, this data should not be over-interpreted and presented carefully.

5) siRNA screen to reveal BAR proteins important for decreased tension invasion. The screen was based on depleting the BAR proteins in the context where 3 ERMs are co-depleted. Would it be possible to test another condition to further validate (or diversify) the findings: for example, BAR could be depleted individually and cells could be treated for 1h with the ROCK inhibitor. Also, why did the authors individually deplete BAR proteins but also co-depleted all the TOCAs? This is biased – co-depletion of related members of BAR proteins should also be carried out, which may reveal additional hits? This should at least be explicitly discussed in the experimental design. Also, Fes and Gas7 are identified in the screen in Fig 5a, but when counter tested in MDA-231, (suppl 6b), they did not behave as expected. This should be discussed.

6) The authors should discuss a body of literature showing SLK or ERM proteins inactivation in *Drosophila* is linked to loss of epithelial phenotypes: PMID: 15371338; PMID: 12511959. This agrees with the data presented in this MS.

Additional specific comments

1) The authors used pERM immunostaining to compare activation of ERM proteins. The fluorescence intensity in figure 1b, 2c, 3b and extended figure 1e should be quantified and statistical analyses should be performed.

2) In general, the images presented are good quality but extremely small. Images could be zoomed and cropped further to display one or two cells so we can clearly appreciate the described phenotypes, especially if they are accompanied with quantification (1b, 2c, 3b, 4b). For example, in figure 2c, it is very difficult to distinguish between the different type of actin-based protrusions. Enlargements should be shown. In figure 5f, enlargement of the pointed area should be presented.

3) The authors explain in extended figure 1c how to calculate plasma membrane tension using the tethering force. Accordingly, that would be easier for the reader if the graphs in figure 2b, 3a and 4a would compare tension instead of tether force.

4) Description of the figure 3 is a little bit disorganized. Figure 3b and 3a should be switched to follow the description in the text. Also, maybe the MDCK +RasV12 data should be sent to supplemental or at least should be described altogether. In fact, the RASV12 is disconnected from the rest of the study. It is unclear from the rationale why they constitute a good model here. Since the model is not used again, and the paper is already data rich, I suggest removing this model for sake of simplicity of the paper.

5) In Figure 4b, cross sections of the Ezrin expressing cells are compared to basal level section of the MA-ezrin expressing cells. Please compare the same focus plane for each condition, especially if you are using these images for quantification of actin intensity.

6) In extended figure 6d-k and supplemental video 6-7-8, the authors show clear co-localization of GFP-FB17 or GFP-MTSS1L with F-actin markers. Even in MA-Ezrin expressing cell, GFP-FB17 puncta seem to align with actin stress fibers (extended figure 6f, Video 8). However, in the graph extended figure 6i-j GFP-FB17 or GFP-MTSS1L spots at the PM/cell were counted even though the related images show F-actin staining and not a membrane marker. Can the author precise which method they use to delineate the plasma membrane for these quantifications?

Minors comments:

-For all supplemental Videos, to facilitate the comparison between conditions, the author should be

present side-by-side videos instead of one after another.

-In figure 1a, the authors used a color code to differentiate the epithelial cells, non-invasive cancer cells versus metastatic cancer cells. They should keep that code among all figure as they did for extended figure1.

-p.13 line 4 "...phenotype driven by Snail overexpression (Fig. 5b not Fig.6b)"

-p.13 line 11 "...Toca proteins resulted in....the inhibition ofthe formation of non-polarized excess membrane blebs (Fig5e)". Could you please indicate on the figure what do you mean by "non-polarized excess membrane blebs"

-In the abstract, the authors state "...reduce PM tension correlates with EMT transition". This could be misleading to readers: they are referring to phenotypic appearance and not molecular EMT. Correct?

-p8 "Y27632 treatment immediately induced the motile phenotype". It would be more appropriate to say "...induce cell to adopt a mesenchymal-like morphology"?

-Figure 2h : typo in "single cell"

-Fig 1: PC3 and PANC-1 data could move to supplementary data.

-Fig 4D: keep the same order as previously in the MS to present migration and invasion data.

Reviewer #2 (Remarks to the Author):

This is and novel, timely and exciting study that links membrane-to-cortex attachment (MCA) to BAR domain protein binding/activity and malignancy. As such it clearly advances our knowledge of metastatic cell migration and membrane tension regulation. The authors use life cell imaging, tether pulling experiments, cell transformation and a metagenomics analysis to suggest that membrane tension is an intrinsic suppressor of metastasis. Two other papers in Biorxiv (Bisaria et al., 2019; Welf et al., 2019) also suggest that a reduction in ezrin activity is necessary for protrusion initiation and migration. This highlights the clear interest in the community for the role of MCA in cell migration and the timely relevance of this particular study. The manuscript is clearly written and nicely presented. Nevertheless, I have a few main concerns and a quite long list of minor ones before accepting it for publication.

Major concerns:

1) Throughout the manuscript there is a lack of quantifications and thus statistics. The display of a single biological replicate or images severely compromises their findings.

- Fig 1B: the authors should quantify the level of co-localization with F-actin of both Ezrin and pERM. Also, the authors should include a plasma membrane staining when they want to state that a protein localises at the membrane as F-actin can not be used as proxy. Last, the statement that the ERM staining in metastatic cells is cytoplasmic is not substantiated from the provided images.

- Extended Fig 1E: without an actin and membrane label it is hard to judge the blebs. Also, a quantification is needed.

- Fig 1D, 2H and 3C: the reconstructed 3D images are nice but don't provide any information on how reproducible this behaviour is. It is also not clear what type of image processing has been carried out (e.g. max projection?).

- Extended Fig 1F: it is very hard to see anything at all. The videos are a bit more helpful. Can the authors either find a better way to image this or provide time series images as in Extended Fig 4. Again a quantification is required.

- Fig 2G: can the authors add statistics?

- Fig 2H: the statement that "cortical actin" is decreased needs to be substantiated. To visualize F-actin they use Alexa Fluor 568 Phalloidin that does not specifically label cortical F-actin, but all

filamentous actin. I suggest they just name it F-actin.

- Fig 3B: can the authors explain how they define cortical F-actin in MCF10A cells? Do they mean F-actin at the cell surface? Quantifications of F-actin and pERM localisation are needed throughout this figure.

- Fig 4B: the authors conclude that MA-Ezrin expression: "resulting in the suppression of both actin- and bleb-based membrane protrusions (Fig. 4b)." But the quantification is of "cortical" F-actin. Can the authors either quantify the protrusion types or change the sentence? Furthermore, the authors do not describe how they quantified actin in the caption nor in the method section.

- Fig 4G: statistics are missing.

- Fig 5E: a quantification is missing. Also without a membrane and actin labels is hard to judge but I would say that the two depicted cells with si-MTSS1L and si-Toca proteins have small blebs.

- Fig 5H, Extended Fig 6D, G, H and K: a quantification is missing.

2) The images where the authors assess cell blebbing and ruffling are often not enough to distinguish those protrusions unequivocally. Actin and membrane markers should be used to assess what those cells are doing. This puts into question some of the conclusions in Fig 1A. The authors should provide representative images of such tether pulling experiments as it is hard for me to judge if those two cell behaviours can be easily distinguished with bright field. On the other hand, if it is not trivial to distinguish ruffling and blebbing cells, and there are no statistically significant changes in tether force between the cells with two protrusion types, maybe there is no need to distinguish between these in the plot?

3) The perturbations targeting RhoA signalling will have very severe cell mechanics effects aside from MCA (as myosin-2 phosphorylation and thus contractility are its main target), those should be acknowledged and also experimentally addressed.

- Y27 and si-RhoA will also clearly target cell contractility, could the authors address if the observed effects are not only due to a reduction in pERM but also a reduction in pMyo? Can they for example use blebbistatin?

- Fig 2C shows a different pERM localisation for si-RHOA and Y27, also suggesting that the presented signalling above ERM proteins is an oversimplification. Can the authors quantify such behaviour and discuss it further in the text?

4) The push of the authors to claim a general role of the RHOA-ERM axis in several cancer types has severe caveats. I suggest the authors tone down their conclusions on that section, the discussion (and maybe even the title) as the data presented does not really support the universality they claim.

- A lot of weight is given to the analysis of RhoA in human samples but the main role of this protein is not regulating ERM proteins but Myosin 2 and thus contractility (see my main concern 3).

- Extended Fig 3B: there are a very large number of tumours with no difference and also several with the opposite trend for RhoA expression, again challenging the universality of the RhoA-ERM axis.

- Fig 3H: could the authors plot only the ERM kinases? Does that also show a survival difference? As with RhoA, PIP5Ks have a very broad plethora of functions and thus the observed difference could be confounding. Plotting only ERM kinases could strengthen their claim of over the RhoA-ERM axis.

- Fig 3E: why do the authors show the data for Luminal and HER2 positive cells but don't discuss it? The difference between Normal and triple negative is significant but the magnitude is really small compared to the spread of the data.

- Extended Fig 3C: what are the percentages next to the gene names?

- The statement: "epithelial tumours frequently and exclusively harbour putative heterozygous deletions of RHOA and ERM, as opposed to metastasis-associated genes, such as SNAI1 (Snail) (Fig. 3d and Extended Data Fig. 3a)." needs to be better justified. It is not clear to me how such conclusion can be obtained from the presented data.

- How would the authors explain the result that MDA-MB-231 cells do not show decrease in RHOA levels? Please comment on it, as it is one of the main cell lines used in the paper.

General format comments:

- The order of the figure panels does not follow the text making it confusing sometimes. I suggest

the authors rearrange the figures so the text logic is followed in the figures.

- The frame rate of all movies is so high that it is hard to see anything. I would also suggest the authors add arrows to the relevant parts that reviewers/readers should be focusing on as in Extended Fig 4.
- The sample size in every biological replicates should be described in the figure legends.

Minor concerns:

- In the sentence: "The PM reversibly associates with the actin cortex via linker proteins, such as ezrin, radixin, and moesin (ERM) family proteins, whereby cell membrane deformability is intrinsically dependent on the degree of membrane-cortex adhesion (MCA)^{20, 21}." What do the authors mean with deformability?
 - Fig.1a: Why do the authors use the One-way ANOVA with Tukey-s multiple comparisons test if only single comparisons to MCF10A are indicated in the plot?
 - The role of cell adhesion in epithelia is under-explored and not really discussed in this manuscript. I agree that experimentally addressing it might be out of the scope of the current work but a paragraph in the discussion on how the balance of forces (membrane tension vs. adhesion) is key for migration is necessary as one of the main characteristics of epithelial cells is their cell-cell adhesion complexes. Also in Supplementary video 1 Hs578T cells seem to bleb when isolated but ruffle when adherent but it is hard for me to judge without actin and membrane labels (see my Major concern 3). Discussing how the balance of forces could affect protrusion could also be very fitting.
 - Last, in the text they mention: "E-cadherin depletion had no effect on invasive migration (Fig. 2e, f)." but I could not find the corresponding data.
 - The images on the left of Extended figure 1D are not described in the legend. They are so small that are hard to see and understand.
 - The schematic in Fig 2A is a massive oversimplification of what we know about ERM regulation. Either simplify further the schematic and not depict the regulation or add more information.
 - The authors write: "Indeed, we also found that knocking down these BAR proteins suppressed an elongated invasive phenotype driven by Snail overexpression (Fig. 6b)." I could not find the corresponding data.
 - The statement: "PM tension is regulated by MCA" is misleading as in-plane tension is regulated independently
 - The concept "restoring PM tension" is misleading. The authors do not restore PM tension, they increase tension of MDA-MB-231 above their basal level or decrease PM tension of MCF10a below their basal level.
 - MCF7 cells are labeled as a non-invasive cancer cell line, but they are actually metastatic. This contradicts the following statements in the main text: " We found that the tether force of MCF10A cells was largely comparable to that of non-invasive human breast cancer cells (AU565 and MCF7)".
 - The authors write: "The ability to maintain high PM tension appears to be a common characteristic (...)". Could the authors phrase it in a way to acknowledge that "high MT" is relative to other measurements in this paper, not in general. Many cells lines exhibit even higher values than reported in the paper. Using terms like "high" and "low" without context makes it imprecise (this comment is also applicable to "low" on pages 6 and 9).
 - Please specify what kind of image is shown (max intensity projection, mean intensity projection, single slice?) for Fig 1b, 2c, 3b, 4b, 5f and Extended Fig 1e, 6g, 6h, 6k.
 - Fig 5a,b: Please provide information on what is an "invasive structure" and how as such, was it quantified.
 - Extended figure 6b: Please provide the brief description of how the invasion rates were quantified.
- Also, as the levels of RhoA in MDA-MB-231 are comparable to MCF10A cells, could the authors perform the invasion assay's in the siRHOA together with si-BAR domain proteins? Does it mean that Toca proteins and MTSS1L alone increase invasiveness?
- What are the levels of pERM proteins in all the cell lines in the study?

Typos:

- Reference to Fig 2C: "A similar result was observed when ROCK activity was acutely reduced using its specific inhibitor Y27632 (Fig. 2b, right). As expected, RHOA knockdown or Y27632 treatment resulted in the cytoplasmic distribution of ezrin and decreased pERM signal (Fig. 1c)."

- Fig 2H: single cells.
- Fig 6b: Plasma membrane tension
- Extended Fig 2e, the legend "RhoA" should be "RhoA RNAi"
- Scale bars are missing in Fig 1c, Extended Fig 1b and 4b.
- On page 10, "We hypothesized that if PM tension reduction is key acquiring (...)", please mind the missing "to".

Reviewer #3 (Remarks to the Author):

Review for Tsujita et al " Homeostatic membrane tension is an intrinsic mechanical suppressor of cancer cell dissemination"

Summary

In this manuscript, Tsujita et al use a number of complementary methods to show that low membrane tension is associated with cell and tumor invasiveness and metastasis. In this sense, the authors describe this mechanical cellular property as being tumor suppressive.

The authors begin by measuring membrane tension (which is determined mostly by membrane to cortex attachment, MCA) using a membrane tether assay in several invasive and non-invasive cancer cell lines. They find that invasive cells, and indeed higher invasiveness in transwell migration assays, is associated with lower membrane tension. Low levels of membrane tension and invasiveness is also associated with reduced levels of F-actin and ERM proteins at the cell cortex and more elongated cell shapes/increased cell protrusions. The authors then show that reducing RhoA pathway activity and ERM activity result in lower membrane tension and increased protrusive and invasive behavior in cells cultured on 2D substrates, under 3D ECM networks or embedded in 3D ECM networks. Similarly, overexpression of the EMT drivers Snail and Slug and ectopic activation of Ras led to lower membrane tension, reduced ERM/F-actin at the cortex and increased protrusive activity. Meta-analysis of gene expression data from The Cancer Genome Atlas are used to show that epithelial tumors are associated with reduced Rho and ERM activity and that Rho/ERM pathways are associated with reduced survival. The authors also find that in most of the invasive cell lines tested, RhoA expression is low. The authors then show that expression of a membrane-targeted (MA-) Ezrin is sufficient to increase membrane tension and cortical F-actin levels. MA-Ezrin expression reduces protrusive activity, blocks cells migration/invasion in vitro and reduces metastasis in a tail vein metastasis assay in mice. The authors identify several BAR domain proteins whose knockdown reduces invasive phenotypes and which localize to protrusions.

This study offers a number of interesting observations, some of which have also been made in other recent publications. The major message of the paper, that mechanical properties themselves can be thought of as tumor suppressive, is attractive. Although the data presented in the manuscript indeed point in this direction, the physical origins of many of the phenotypes presented in the manuscript are unclear, which makes the overall model difficult to understand.

Major Points

1. In the introduction, the authors define membrane tension as arising largely from membrane to cortex attachment. In much of the paper, the authors focus on the role of ERM proteins, which link the two structures. However, they tend to ignore the structure of the actin cortex itself. In Fig. 1, for example, panel b shows that non-invasive cells (which also have high membrane tension) have low levels of cortical Ezrin. However, they also have considerably less cortical F-actin. It is unclear whether the difference in membrane tension is because Ezrin is not localized to the cortex, or because there is no actin cortex to which the membrane can attach. In the latter case, membrane tension would be expected to be low no matter what the state of ERM proteins. The authors seem to attribute much of their phenotypes to the state of ERM, but it is also likely that the organization of the actin cortex could play as much of a role. In line with this, a recent pre-print (Bisaria et al [2019] *BioRxiv*: <https://www.biorxiv.org/content/10.1101/705509v1.full>) highlights the role in

cortical actin in restricting protrusions by membrane fluctuations. The roles of cortical actin and ERM activity in membrane tension/protrusion formation should be reconciled in order to make sense of the overall model. It would be helpful to address this if relative ERM and cortical actin could be quantified when these images are presented.

2. In line with the above comments, RhoA siRNA and Y-27632 are mischaracterized as inhibitors of ERM activity. Rho activity has a number of downstream effects, perhaps most prominently on actomyosin itself, as it is involved in actin dynamics (via Formins, LIMK/Cofilin and Arp2/3 [due to Rac inhibition]) and myosin activity [via ROCK/Citron Kinase]. Based on previous studies (e.g. Tinevez et al [2010] PNAS), modification of Rho activity or Y-27632 treatment would also be expected to drastically change cortical tension. Previous studies have argued that low cortical tension, following treatment with Y-27632 and Blebbistatin (a myosin-2 inhibitor that would presumably not directly affect ERM activity), can also lead to increased protrusive activity (Bergert et al [2012] PNAS). As actin organization seems to change RhoA knockdown and Y-27632 treatment (Fig. 2c), it is unclear whether the primary defect is really ERM proteins or cortical actin/cortical tension. For a more upstream target of ERM, perhaps siRNA against SLK could be tested instead. In the manuscript, RhoA siRNA and Y-27632 should be better characterized to determine the true origin of reduced membrane tension and invasiveness.

3. The authors do show that directly abrogating ERM activity by triple siRNA knock-down can lead to lowered membrane tension and increased protrusion (Fig. 2). However, the complication here is that ERM knockdown has also been shown to reduce cortical tension, at least during mitosis (Kunda et al [2008] Curr Biol). In Kunda et al, cells with ERM knockdown fail to round during mitosis and remain flat and protrusive, a phenotype that is copied by knockdown of ROK/Citron and Myosin-2 light chain. Conditions of low cortical tension would therefore be expected to result in less round cell shapes, consistent with the results in this manuscript. Additionally, the MA-Ezrin data presented in Fig. 4 show an increase in cortical actin, and ERM phosphomimic mutants have been shown to increase cortical tension (Kunda et al [2008] Curr Biol). The loss of protrusions/invasiveness due to MA-Ezrin expression could therefore also be due rather to changes in cortical tension rather than MCA/membrane tension. If the authors would like to argue that membrane tension, and not cortical tension, is the primary physical mechanism, then cortical tension should also be measured. It is indeed possible that cortical tension and MCA/membrane tension are coupled and that these mechanical properties are part of the same mechanical signature linked with invasiveness. In this case, the message of the paper should reflect this and highlight the known role of cortical tension.

4. In the 3D collagen/Matrigel assays, the authors show a convincing link between RhoA/ERM loss and protrusive activity. It would be helpful to have some quantification of the migration to better interpret how differences in protrusions results in increased migration capacity. Again, these migration data should be compared with recent studies that have shown that inhibition of myosin-2 activity can lead to increased protrusions and increased single or collective migration (Liu et al [2010] Br J Pharmacol, Xu et al [2019] Nat Commun). These data should also at least be discussed in the context of results showing that the effects of increasing or decreasing cortical tension on cell migration depends on ECM network properties (e.g., Wang et al [2019] Nat Commun).

5. In the case of induced EMT by overexpression of Snail/Slug and expression of K-Ras G12V, it appears that F-actin organization is substantially changed, which again brings up the question of whether the primary change is the loss of actin cortex organization or MCA.

6. In Fig. 3a-3c, the authors focus on EMT as a trigger of low membrane tension/high invasiveness. For the EMT conditions (Snail/Slug expression), would it be possible to check for EMT by staining/Western blotting for classical EMT markers, as in Extended Fig. 2C? Also, in the control cells shown in Fig. 3b, are these phenotypes really epithelial? The cells also seem reasonably well-spread, and it is not clear if they are Apicobasally polarized.

7. In the model cartoon on Fig. 6b, the authors suggest a reversible conversion between static epithelial cells and motile cells. However, this does not really reflect the results in the manuscript. Many of the experiments show interconversion between more round and more elongated/protrusive cell shapes. However, a round shape is not enough to characterize a cell as "epithelial." In the Rho/ERM knock-downs, the authors show that cells do not lose their epithelial characteristics despite being more invasive (Extended Fig. 2c). Also, there is no evidence that the conditions that force cells to assume a round shape make them more epithelial.

Minor Points

1. In Fig. 1b, it would be helpful if the images from different cell lines are presented in the same order as in Fig. 1a.
2. In Extended Data Fig. 1d: Is the line a linear fit of the data? It does not seem to fit very well. The data look rather like 2 distinct populations than a linear continuum. The fit seems unnecessary and confusing.
3. In the results section, the authors claim that their results regarding the connection between loss of RhoA/ERM activity and invasiveness is contrary to common belief (“Surprisingly, despite that ERM proteins and RHOA are commonly thought to play an active role in cancer cell invasion^{36, 37}”). However, several more recent studies have also linked loss of RhoA to invasiveness, and the authors cite this in the Discussion section (“recent studies have indicated that RHOA plays a tumor suppressor role in many cancers by inhibiting invasion and metastasis⁶²⁻⁶⁶”). This fact should be acknowledged when motivating the data in the Results section to accurately reflect the state-of-the-art in the field, as the results are less surprising in light of these recent studies.
4. I find the following sentence slightly confusing/misleading: “...epithelial tumors frequently and exclusively harbor putative heterozygous deletions of RHOA and ERM, as opposed to metastasis-associated genes, such as SNAI1.” If I am understanding the plots correctly in Fig. 3d and Extended Fig. 3a, it seems like loss of RhoA is just a frequently found as gain of SNAI1.
5. Related to the following sentence: “...knockdown of ERM, RHOA, or EMT induction all resulted in the accumulation of GFP-FBP17 at the PM where it appeared to activate Arp2/3 complex-dependent actin nucleation for the directed migration.” – Although it is cited in the manuscript that MTSS1L and Toca proteins can activate Arp2/3, there is no evidence in Fig. 5 suggesting that is what is happening in this case. It would be better to remove the suggestion of Arp2/3 activation unless the authors can provide evidence of that.

We would like to thank all the reviewers for taking the time to carefully read our manuscript. We are confident that this constructive criticism and our responses have made this a much stronger paper. Please note that reviewer comments are in black and **our responses are in blue**. All changes in the manuscript text are **also highlighted in blue**. In addition, previous "Extended Figures" has been changed to "Supplementary Figures" according to the style of Nature Communications.

Reviewer #1 (Remarks to the Author):

Tsujita et al. submitted an article in Nature Communications entitled “Homeostatic membrane tension is an intrinsic mechanical suppressor of cancer cell dissemination”. In this manuscript, the authors tested the novel hypothesis that the cell-intrinsic physical factors that maintain epithelial cell mechanics could function as tumor suppressors. They observed that epithelial cells maintain higher plasma membrane tension than their metastatic counterparts, which prevent invasion and metastasis. They demonstrate that in epithelial cells, high tension is maintained by the RHOA/ERM proteins pathway that connects the plasma membrane to the actin cell cortex (membrane-cortex adhesion, MCA). In non-invasive epithelial cells, downregulation of the RHOA/ERM pathway leads to a decrease in tension that triggers the membrane-curvature sensing/generating BAR family proteins to stimulate actin-polymerization powering mesenchymal and amoeboid migration. They overexpressed a plasma membrane-anchored active version of the ERM protein Ezrin (MA-ezrin) in invasive cells and this was sufficient to inhibit both mesenchymal and amoeboid 3D migration. In vivo, expression of MA-ezrin was sufficient to abrogate experimental metastasis. Collectively, these data led the authors to claim that restoring membrane tension in metastatic cells is sufficient to suppress invasion and metastasis.

The manuscript is well written, generally easy to follow, and technically solid. The authors robustly established that manipulating ERM activity influences cell migration in 2D, transwell migration, 3D migration and also experimental lung metastasis of a triple-negative breast cancer cell line. The authors attributed the anti-metastatic properties of the RHOA/ERM pathway to an increase in cell tension, which uncouples the membrane-binding of BAR family proteins that stimulates actin polymerization. There are however major concerns about the conclusions of this study.

Major Comments

1) To claim that restoring tension is sufficient to suppress invasion and metastasis, the authors assumed that the correlation between ERM proteins activity and high tension measured in 2D context was the same in 3D. To address this concern, the authors should think of a 3D compatible method to measure plasma membrane tension. I

understand the challenge that it could represent. At a minimum, the author should demonstrate that increasing the plasma membrane tension using an alternative way than upregulating ERM proteins also suppress invasion and metastasis dependent of the BAR family proteins.

The reviewer has raised a good point. Probing PM tension in 3D is of great interest in the field; however, it remains challenging (Pontes B et al., *Semin Cell Dev Biol*, 2017 PMID: 28851599). Nevertheless, given the importance of MCA in PM tension control, we investigated the degree of MCA in 3D by quantitating ERM proteins and F-actin levels beneath the plasma membrane (PM). As shown in Fig. 1d, e, while malignant cells in 3D have lower levels of ERM and F-actin at the PM, epithelial cells exhibit overall higher MCA levels as they do in 2D. These data support our idea that even in 3D, PM tension in non-motile cells is higher than that in cancer cells.

In addition, as the reviewer suggested, we attempted to increase PM tension independently of manipulating the ERM activity. It has been reported that cholesterol depletion by methyl- β -cyclodextrin (M β CD) treatment increases PM tension, presumably independent of ERM activity (Pontes B et al., *Semin Cell Dev Biol*, 2017). We found that treating MDA-MB-231 cells with M β CD increased PM tension and significantly suppressed their invasion ability (Supplementary Fig. 4b and c). Furthermore, this treatment is sufficient to prevent the recruitment of BAR proteins to the PM (Supplementary Fig. 6i and j), consistent with our notion that homeostatic high PM tension suppresses cancer cell dissemination by counteracting mechanosensitive BAR proteins.

2) Another concern is about tether force measurement itself as a read-out of plasma membrane tension. I believe that optical tweezers technic allows measuring the tether force locally. It is mentioned in the method section that measurements were taken on the lateral side of the cell. However, it is very likely that the plasma membrane tension will be different at the rear, the side or the front of a migrating cell. I am not sure that one local measurement is a good read-out of the overall plasma membrane tension.

As the reviewer pointed out, PM tension appears to be locally altered in motile cells. We measured PM tension on the lateral side of malignant cells for a fair comparison between non-polarized epithelial cells (with no front and rear) and cancer cells. As PM tension correlates with MCA and epithelial cells exhibit an overall higher MCA level, we believe that optical tweezers measurements reflect the overall PM tension, at least in non-motile cells. Regarding migrating cells, it has been proposed that there is a direct link between decreased MCA and membrane protrusion (Welf et al., *Dev Cell*, 2020 PMID: 33308479). In addition, we found that the PM tension of cancer cells is low even on the lateral side, which may reflect why they tend to generate membrane protrusions

globally and randomly. Overall, these observations support the notion that the PM tension of epithelial cells is globally higher than that of cancer cells.

That being said, that would be interesting if the authors could explore other techniques to measure or even visualize membrane tension for example using the newly-developed Fluorescent Membrane Tension Probe. Other approaches would of course be also acceptable to further support the tension measurements.

We thank the reviewer for this suggestion. However, recently developed tension probes recognize in-plane membrane tension rather than MCA-dependent tension (Colom et al., *Nat Chem*, 2018 PMID: 30150727). To our knowledge, there is no probe that specifically detects MCA-based tension. ERM proteins are considered to be the most suitable molecules to reflect this type of tension currently, as described in recent studies (De Belly et al., *Cell Stem Cell*, 2021 PMID: 33217323). Again, we would like to emphasize that our ERM data support the tension measurements. This is further supported by our data showing that the localization and activity of BAR proteins are dependent on ERM-mediated PM tension. The quantitative analysis or visualization of local MCA-based tension is an important issue to be solved in the future.

3) The in vivo assay to probe the contribution of MCA to metastasis is limiting. While the authors confirmed similar cell proliferation on plastic of control and MA-ezrin cells, whether this is also the case in vivo must be tested. The authors conducted an experimental metastasis assay that bypasses the local invasion and intravasation steps that are most relevant to this study. As such, a grafting experiment in mammary fat pads would allow for measuring both tumor growth and metastasis. The authors could also measure the tumor cell blood burden in these assays. This would provide a clear picture of altering membrane tension on cell invasion, intravasation, and colonization. Conversely, it would be highly informative if the authors could carry out an in vivo experiment with a non-metastatic cell model (MCF7?) and test whether depletion of ERMs allows local invasion or intravasation (tumor cell blood burden) or even metastasis to lungs.

This is an important point. As the reviewer suggested, we evaluated tumor formation and spontaneous metastasis using an orthotopic mouse model. Unexpectedly, we found that MA-ezrin cells had a reduced tumorigenic ability and produced significantly smaller tumors (Fig. 4h and Supplementary Fig. 4f). Consistently, no lung metastasis was observed in mice injected with MA-ezrin cells (Fig. 4k). As tumor size is commonly correlated with metastasis, we also evaluated local invasion. As shown in Fig 4i and j, MA-ezrin cells displayed significant reduced invasion into the surrounding tissue, supporting the inhibitory role of PM tension in invasion *in vivo*.

Although the mechanisms by which PM tension suppresses tumor formation and growth are unclear, one possibility is that this can also be due to the tension-dependent inhibition of BAR proteins, as some BAR proteins have been reported to play an active role in tumor formation and growth (Oneyama et al., *Oncogene*, 2016 PMID: 25867068; Huang et al., *Gut*, 2018 PMID: 28647685). Another attractive possibility is that changes in PM tension could be associated with cancer stemness. Recent studies have suggested a direct link between cell deformability and cancer stemness (Zhang et al., *PNAS*, 2012 PMID: 23112172; Lv et al., *EMBO J*, 2021 PMID: 33274785). Our data showing that EMT induction, which is known to confer the properties of cancer stem cells leads to decreased PM tension might reflect this. These points have been included in the Discussion section (page 17, lines 386–394).

In addition, we agree with the reviewer that it is informative to examine whether the deletion of ERM proteins triggers local invasion and metastasis of non-invasive cells. Unfortunately, however, since the tumorigenicity of MA-ezrin cells was low and, thus, many mice were required for the experiments, we were forced to focus on MA-ezrin experiments as animal experiments were restricted because of COVID-19. Related to this point, as the reviewer helpfully pointed out below, an *in vivo* study reported that the loss of moesin alone, the sole *Drosophila* ERM protein, is sufficient to induce invasion in *Drosophila*. This study is now quoted in the revised Discussion section to support our conclusions (page 15, lines 345–347).

4) While I appreciate the in-silico analyses performed by the authors, I wonder if some conclusions are accurate. The differences in RHOA expression among subtypes remains rather small (3E). Since RHOA is an enzyme, a small decrease in protein levels could easily be compensated by increased activation by a GEF. It would be essential to explore this to make strong conclusions (this may be beyond the scope of this MS, but the concept should still be discussed). The same would be true for ERMs (vs. pERMs). The authors should also quantify the blots presented in Fig 3F-G from multiple experiments. Globally, while interesting, this data should not be over-interpreted and presented carefully.

We agree with this reviewer's concern. In fact, GEF expression, rather than RHOA expression, seems to correlate with the survival rate of cancer patients (previous Fig. 3h). Therefore, we removed the RHOA expression data. In addition, as the reviewer suggested below, we focused on SLK and STK10, specific ERM kinases, and found that the expression levels of SLK and STK10 are significantly correlated with the prognosis of cancer (including breast and lung cancer) patients (Fig. 3h). The expression level of ERM proteins does not seem to correlate with the survival rate of patients. Since RHOA also affects myosin activity, we replaced the GEF data with the ERM kinase data (Fig.

3h)

5) siRNA screen to reveal BAR proteins important for decreased tension invasion. The screen was based on depleting the BAR proteins in the context where 3 ERMs are co-depleted. Would it be possible to test another condition to further validate (or diversify) the findings: for example, BAR could be depleted individually and cells could be treated for 1h with the ROCK inhibitor. Also, why did the authors individually deplete BAR proteins but also co-depleted all the TOCAs? This is biased – co-depletion of related members of BAR proteins should also be carried out, which may reveal additional hits? This should at least be explicitly discussed in the experimental design. Also, Fes and Gas7 are identified in the screen in Fig 5a, but when counter tested in MDA-231, (suppl 6b), they did not behave as expected. This should be discussed.

As another condition to support our data, we validated this screen by co-depleting RHOA and BAR proteins (please see Supplementary Fig. 6c). The reason for the simultaneous knockdown of three Toca family proteins is that when these proteins were deleted individually, the invasive phenotype tended to decrease modestly. In fact, some BAR proteins have been reported to have functional redundancy, as reported for Toca proteins (Chan Wah Hak et al., *Nat Cell Bio*, 2018 PMID: 30061681). No such tendency was observed in other family proteins in this screen, and we have included this point in the text (page 13, lines 285–289). However, as the reviewer pointed out, combining the removal of other family proteins may reveal more hits. In fact, the function of Fes or Gas7 may be compensated for by other BAR proteins, perhaps depending on the cell type. We mentioned in our original submission that different BAR proteins are involved in the invasion and metastasis of various types of cancer (page 15, line 341).

6) The authors should discuss a body of literature showing SLK or ERM proteins inactivation in Drosophila is linked to loss of epithelial phenotypes: PMID:15371338; PMID:12511959. This agrees with the data presented in this MS.

We thank the reviewer for bringing these papers to our attention. We have included these points in the revised manuscript, as mentioned above.

Additional specific comments

1) The authors used pERM immunostaining to compare activation of ERM proteins. The fluorescence intensity in figure 1b, 2c, 3b and extended figure 1e should be quantified and statistical analyses should be performed.

We quantified the mean fluorescence intensity of pERM in these data and included their statistical data.

2) *In general, the images presented are good quality but extremely small. Images could be zoomed and cropped further to display one or two cells so we can clearly appreciate the described phenotypes, especially if they are accompanied with quantification (1b, 2c, 3b, 4b). For example, in figure 2c, it is very difficult to distinguish between the different type of actin-based protrusions. Enlargements should be shown. In figure 5f, enlargement of the pointed area should be presented.*

We have provided zoomed images of the cell, showing one cell in each panel (Figs. 1b, 1d, 1f, 2b, 3a, and 4b) to make it easier for readers to appreciate. We have also provided quantitative data corresponding to all these data. In Fig. 5f, enlarged images of the pointed area are displayed, and FBP17 spots are indicated by arrowheads.

3) *The authors explain in extended figure 1c how to calculate plasma membrane tension using the tethering force. Accordingly, that would be easier for the reader if the graphs in figure 2b, 3a and 4a would compare tension instead of tether force.*

We have displayed the estimated PM tension value instead of the tether force in Figs. 2a (previous Fig. 2b), 3c (previous Fig. 3a), and 4a, as suggested.

4) *Description of the figure 3 is a little bit disorganized. Figure 3b and 3a should be switched to follow the description in the text. Also, maybe the MDCK +RasV12 data should be sent to supplemental or at least should be described altogether. In fact, the RASV12 is disconnected from the rest of the study. It is unclear from the rationale why they constitute a good model here. Since the model is not used again, and the paper is already data rich, I suggest removing this model for sake of simplicity of the paper.*

We apologize for this confusion. We have switched these data. Please note that the previous Fig. 3a (PM tension data) corresponds to Fig. 3c, as the quantitative data in Fig. 3a has been added to Fig. 3b. In addition, we have moved the RasV12 data to Supplementary Fig. 3c and d. We agree that the RasV12 data do not follow the flow of the text. However, we believe that these data showing that oncogenic Ras expression leads to decreased PM tension further support our notion that reduced tension is closely associated with malignant progression.

5) *In Figure 4b, cross sections of the Ezrin expressing cells are compared to basal level section of the MA-ezrin expressing cells. Please compare the same focus plane for each condition, especially if you are using these images for quantification of actin intensity.*

We replaced the images with ones of the same focus as ezrin-expressing cells.

6) In extended figure 6d-k and supplemental video 6-7-8, the authors show clear co-localization of GFP-FB17 or GFP-MTSS1L with F-actin markers. Even in MA-Ezrin expressing cell, GFP-FB17 puncta seem to align with actin stress fibers (extended figure 6f, Video 8). However, in the graph extended figure 6i-j GFP-FB17 or GFP-MTSS1L spots at the PM/cell were counted even though the related images show F-actin staining and not a membrane marker. Can the author precise which method they use to delineate the plasma membrane for these quantifications?

We fully agree with the reviewer's concerns. To address this, we quantified the number of BAR protein spots that merge with the membrane region, defined by staining with wheat germ agglutinin (WGA), commonly used as a cell membrane marker. This information has been added to the Methods section (page 59, lines 1084–1086). The new data, including membrane staining, are shown in Supplementary Fig. 6f–h. The data in Fig. 5f and g have also been re-quantified accordingly, and new representative images are presented. In addition, we have removed the previous Extended Fig. 6d–f and the corresponding Supplemental Movies 6–8 because (1) it is difficult to perform time-lapse experiments with triple imaging including the PM under our microscopy, and (2) the dynamics of BAR proteins in actin- and bleb-based protrusions have already been shown (Tsujita et al., *Nat Cell Biol*, 2015; Chan Wah Hak et al., *Nat Cell Biol*, 2018; Goudarzi et al., *Dev Cell*, 2017). We believe that these quantitative data are sufficient to support our claim that high PM tension prevents the recruitment of BAR proteins to the PM.

Minors comments:

-For all supplemental Videos, to facilitate the comparison between conditions, the author should be present side-by-side videos instead of one after another.

We have changed the Video data accordingly.

-In figure 1a, the authors used a color code to differentiate the epithelial cells, non-invasive cancer cells versus metastatic cancer cells. They should keep that code among all figure as they did for extended figure 1.

Fixed.

-p.13 line 4 "...phenotype driven by Snail overexpression (Fig. 5b not Fig.6b)"

Corrected.

-p.13 line 11 "...Toca proteins resulted in....the inhibition ofthe formation of

non-polarized excess membrane blebs (Fig5e)”. Could you please indicate on the figure what do you mean by “non-polarized excess membrane blebs”

We have added typical representative images of protrusion phenotypes, including non-polarized blebs, to Supplementary Fig. 6e, showing how the protrusion phenotypes were judged. Quantitative data are shown in Fig. 5e. In addition, to prevent unnecessary confusion, the language of “excess” was removed from this sentence.

-In the abstract, the authors state “...reduce PM tension correlates with EMT transition”. This could be misleading to readers: they are referring to phenotypic appearance and not molecular EMT.

We apologize for this confusion. We paraphrased this sentence as follows: “The forced expression of epithelial–mesenchymal transition (EMT)-inducing transcription factors results in decreased PM tension.”

-p8 “Y27632 treatment immediately induced the motile phenotype”. It would be more appropriate to say “...induce cell to adopt a mesenchymal-like morphology”?

We thank the reviewer for pointing this out. However, we have deleted this sentence, as we have replaced the Y27632 data with the ERM kinase data.

-Figure 2h : typo in “single cell”

Removed.

-Fig 1: PC3 and PANC-1 data could move to supplementary data.

We agree with this point, as we primarily focused on breast cancer cells in our study. However, we believe that it is easier for readers to compare the PM tension of cancer cells when presented side by side with the data for these commonly used cancer cells. Therefore, we would like to keep their data in Fig. 1a.

-Fig 4D: keep the same order as previously in the MS to present migration and invasion data.

Fixed.

Reviewer #2 (Remarks to the Author):

This is and novel, timely and exciting study that links membrane-to-cortex attachment

(MCA) to BAR domain protein binding/activity and malignancy. As such it clearly advances our knowledge of metastatic cell migration and membrane tension regulation. The authors use live cell imaging, tether pulling experiments, cell transformation and a metagenomics analysis to suggest that membrane tension is an intrinsic suppressor of metastasis. Two other papers in Biorxiv (Bisaria et al., 2019; Welf et al., 2019) also suggest that a reduction in ezrin activity is necessary for protrusion initiation and migration. This highlights the clear interest in the community for the role of MCA in cell migration and the timely relevance of this particular study. The manuscript is clearly written and nicely presented. Nevertheless, I have a few main concerns and a quite long list of minor ones before accepting it for publication.

Major concerns:

1) Throughout the manuscript there is a lack of quantifications and thus statistics. The display of a single biological replicate or images severely compromises their findings. - Fig 1B: the authors should quantify the level of co-localization with F-actin of both Ezrin and pERM. Also, the authors should include a plasma membrane staining when they want to state that a protein localises at the membrane as F-actin can not be used as proxy. Last, the statement that the ERM staining in metastatic cells is cytoplasmic is not substantiated from the provided images.

We fully agree with the reviewer's concerns. As suggested, we quantified the level of co-localization of ERM and F-actin with the plasma membrane (PM). PM was stained with wheat germ agglutinin (WGA), commonly used as a PM marker. As shown in Supplementary Fig. 1e and f, we demonstrated that ERM proteins at the cell periphery were co-localized with the PM, where F-actin signals partially overlapped (membrane-proximal F-actin). Therefore, we quantified the mean intensity of ERM or F-actin in the entire membrane region defined by the membrane marker and throughout the cytoplasm and calculated the membrane/cytosol ratio. This description has been added to the Methods section (page 59, lines 1078–1084). For pERM staining, co-staining of pERM with the membrane or F-actin could not be performed because trichloroacetic acid (TCA) fixation compromises both phalloidin (Hayashi et al., *J Cell Sci*, 1999 PMID: 10085250) and WGA staining. Therefore, we quantified the mean intensity of pERM signals along the cell edge and moved the pERM data to the Supplementary Figs.

Last, the statement that the ERM staining in metastatic cells is cytoplasmic is not substantiated from the provided images.

We have deleted this sentence and rephrased it as follows; “MDA-MB-231 and Hs578T cells had globally lower levels of membrane-associated ERM and F-actin.” (page 6, line

123).

- *Extended Fig 1E: without an actin and membrane label it is hard to judge the blebs. Also, a quantification is needed.*

As described above, we co-stained PM and F-actin and defined actin-free membrane protrusions as blebs. Their quantitative data are shown in Fig. 1c. Please note that for the above reason, the quantitative data of ERM and F-actin, instead of pERM, are displayed.

- *Fig 1D, 2H and 3C: the reconstructed 3D images are nice but don't provide any information on how reproducible this behaviour is. It is also not clear what type of image processing has been carried out (e.g. max projection?).*

We agree with this reviewer's comment. In addition, in response to reviewer #1 (comment 1), instead of showing these representative data, we have provided quantitative data on ERM and actin levels in 3D (Fig. 1d and e), indicating that malignant cells have decreased MCA, even in different 3D migration modes. Representative confocal images of a single plane in the middle of the cell and the maximum projection images are shown in Fig. 1d and f, respectively. Accordingly, in Figs. 2h and 3e (previous Fig.3c), we replaced the previous representative data with the quantitative data.

- *Extended Fig 1F: it is very hard to see anything at all. The videos are a bit more helpful. Can the authors either find a better way to image this or provide time series images as in Extended Fig 4. Again a quantification is required.*

We agree that it was difficult to distinguish between the types of protrusions in such phase-contrast images. Therefore, as described above, cells embedded in 3D were co-stained with PM, ERM, and F-actin to quantify their phenotypes. These new representative images and quantitative data are shown in Fig. 1d and Supplementary Fig. 1h and i.

- *Fig 2G: can the authors add statistics?*

We have now added statistics to Fig. 2g.

- *Fig 2H: the statement that "cortical actin" is decreased needs to be substantiated. To visualize F-actin they use Alexa Fluor 568 Phalloidin that does not specifically label cortical F-actin, but all filamentous actin. I suggest they just name it F-actin.*

We thank the reviewer for this suggestion. We have removed the word “cortical actin” from the text. Instead, as mentioned above, we focused on membrane-proximal F-actin to better reflect MCA in the revised manuscript.

- Fig 3B: can the authors explain how they define cortical F-actin in MCF10A cells? Do they mean F-actin at the cell surface? Quantifications of F-actin and pERM localisation are needed throughout this figure.

Although we initially defined F-actin at the cell periphery as cortical F-actin, we have removed this term in the revised manuscript. As the co-staining of pERM and F-actin could not be performed for the above reason, we quantified the ERM and F-actin levels beneath the PM. As shown in Fig. 3a and b (corresponding to the previous Fig. 3b), overexpression of Snail resulted in lower levels of membrane-associated ERM and F-actin.

- Fig 4B: the authors conclude that MA-Ezrin expression: "resulting in the suppression of both actin- and bleb-based membrane protrusions (Fig. 4b)." But the quantification is of "cortical" F-actin. Can the authors either quantify the protrusion types or change the sentence? Furthermore, the authors do not describe how they quantified actin in the caption nor in the method section.

We apologize for this confusion. In this sentence, we wanted to indicate that the suppression of membrane protrusions is because of MA-ezrin expression. Therefore, we have provided the quantitative data of the protrusion types, showing that increasing MCA suppressed both types of protrusions (Fig. 4c). In addition, we removed the quantitative data of F-actin from Fig. 4b.

- Fig 4G: statistics are missing.

We have added statistics to Fig. 4f (corresponding to the previous Fig. 4g). Please confirm.

- Fig 5E: a quantification is missing. Also without a membrane and actin labels is hard to judge but I would say that the two depicted cells with si-MTSS1L and si-Toca proteins have small blebs.

We thank the reviewer for raising this concern. We added quantitative information to Fig. 5e. In addition, typical representative images of protrusion phenotypes, including non-polarized blebs (defined by staining with the cell membrane), are shown in Supplementary Fig. 6e, showing how protrusion phenotypes were judged.

- Fig 5H, Extended Fig 6D, G, H and K: a quantification is missing.

We have added quantitative data to these figures. Please note that we have removed previous Extended Fig. 6d–f and their corresponding movies in response to reviewer #1 (comment 6).

2) The images where the authors assess cell blebbing and ruffling are often not enough to distinguish those protrusions unequivocally. Actin and membrane markers should be used to assess what those cells are doing. This puts into question some of the conclusions in Fig 1A. The authors should provide representative images of such tether pulling experiments as it is hard for me to judge if those two cell behaviours can be easily distinguished with bright field. On the other hand, if it is not trivial to distinguish ruffling and blebbing cells, and there are no statistically significant changes in tether force between the cells with two protrusion types, maybe there is no need to distinguish between these in the plot?

We agree with this reviewer's concern that it is difficult to clearly distinguish between ruffling and blebbing in bright field images. However, as shown in a previous study (Chikina et al., *J Cell Biol*, 2019 PMID: 30541746), time-lapse movies can clearly distinguish between these two protruding structures (the ruffle formation of the wavy structure and the bleb formation that resembles an inflating balloon). We have provided representative movies of ruffling and blebbing cells under tether-pulling experiments (Supplementary Movie 1).

3) The perturbations targeting RhoA signalling will have very severe cell mechanics effects aside from MCA (as myosin-2 phosphorylation and thus contractility are its main target), those should be acknowledged and also experimentally addressed.

This is a very important point raised by this reviewer and the reviewer #3 and has been clarified below (please see also our response to the reviewer #3; comment 2).

- Y27 and si-RhoA will also clearly target cell contractility, could the authors address if the observed effects are not only due to a reduction in pERM but also a reduction in pMyo? Can they for example use blebbistatin?

We agree with this comment and have focused on ERM-specific kinases (SLK and STK10), rather than ROCK. We found that the co-depletion of SLK and STK10 in MCF10A cells led to decreased PM tension (Fig. 2a) and increased migration and invasion (Fig. 2e and f). Importantly, phosphorylated myosin light chain (pS19MLC), a marker for myosin II activation, was not altered in ERM- and SLK + STK10-depleted cells (Supplementary Fig. 2c), indicating that the observed difference in PM tension is

specifically due to changes in MCA rather than those in actomyosin contractility. Therefore, we replaced the Y27632 data with the ERM kinase data to better reflect the MCA.

- Fig 2C shows a different pERM localisation for si-RHOA and Y27, also suggesting that the presented signalling above ERM proteins is an oversimplification. Can the authors quantify such behaviour and discuss it further in the text?

As the reviewer pointed out, we noted that pERM exhibits rather cytoplasmic localization following Y27632 treatment. However, we did not quantify this difference, as we removed the data for Y27632. In SLK + STK10 knockdown cells, pERM localization appeared to be similar to that of RHOA-depleted cells (Supplementary Fig. 2d).

4) The push of the authors to claim a general role of the RHOA-ERM axis in several cancer types has severe caveats. I suggest the authors tone down their conclusions on that section, the discussion (and maybe even the title) as the data presented does not really support the universality they claim.

We fully agree with this comment and have addressed this by (1) toning down general language regarding the RHOA-ERM axis throughout the text and (2) removing the paragraph about RHOA in the Discussion section. We also removed the word "intrinsic" from the title and changed it to "**Homeostatic membrane tension acts as a mechanical tumor suppressor and constrains cancer cell dissemination**".

- A lot of weight is given to the analysis of RhoA in human samples but the main role of this protein is not regulating ERM proteins but Myosin 2 and thus contractility (see my main concern 3).

We agree and have removed the RHOA data from Fig. 3 (previous Figs. 3e–h). Instead, as the reviewer suggested below, we have added the ERM kinase data for cancer patients.

- Extended Fig 3B: there are a very large number of tumours with no difference and also several with the opposite trend for RhoA expression, again challenging the universality of the RhoA-ERM axis.

We have removed this data to tone down the RHOA-ERM axis as mentioned above.

- Fig 3H: could the authors plot only the ERM kinases? Does that also show a survival difference? As with RhoA, PIP5Ks have a very broad plethora of functions and thus the

observed difference could be confounding. Plotting only ERM kinases could strengthen their claim of over the RhoA-ERM axis.

We thank the reviewer for this suggestion. In the revised manuscript, only ERM kinases are plotted in Fig. 3h. Interestingly, we found that the expression levels of ERM-specific kinases (SLK + STK10) were positively correlated with the prognosis of breast, lung, and gastric cancer patients (Fig. 3h). These results support our idea that MCA-dependent tension acts as a mechanical tumor suppressor.

- Fig 3E: why do the authors show the data for Luminal and HER2 positive cells but don't discuss it? The difference between Normal and triple negative is significant but the magnitude is really small compared to the spread of the data.

We have removed this data according to the deletion of the RHOA data.

- Extended Fig 3C: what are the percentages next to the gene names?

We apologize for the lack of information. These percentages indicate the rate at which the gene of interest is altered across cancer cell types. However, we have changed these data to the same graph style used in Fig. 3g for easier comparison by readers (Supplementary Fig. 3f [previous Extended Fig. 3c]). Please also note that these graphs have been recreated based on the Cancer Cell Line Encyclopedia (CCLE) data updated in 2019 (Ghandi et al., *Nature*, 2019).

- The statement: "epithelial tumours frequently and exclusively harbour putative heterozygous deletions of RHOA and ERM, as opposed to metastasis-associated genes, such as SNAIL (Snail) (Fig. 3d and Extended Data Fig. 3a)." needs to be better justified. It is not clear to me how such conclusion can be obtained from the presented data.

Reviewer #3 made a similar point. Therefore, we have removed this sentence to avoid unnecessary confusion.

- How would the authors explain the result that MDA-MB-231 cells do not show decrease in RHOA levels? Please comment on it, as it is one of the main cell lines used in the paper.

One plausible interpretation is that MDA-MB-231 cells are capable of actively transitioning between mesenchymal- (low contractility) and amoeboid (high contractility)-motility modes. In this case, regulation of GEF or GAP activity, rather than RHOA levels, seems to be important for enabling rapid transitioning between motility modes. Reviewer #1 also made a similar point (please see our response to

comment 4). In addition, we removed the RHOA expression data in response to reviewer #1's comment.

General format comments:

- The order of the figure panels does not follow the text making it confusing sometimes. I suggest the authors rearrange the figures so the text logic is followed in the figures.

We apologize for these confusions. We have rearranged the figures so that the text flow follows the order of the figure panels.

- The frame rate of all movies is so high that it is hard to see anything. I would also suggest the authors add arrows to the relevant parts that reviewers/readers should be focusing on as in Extended Fig 4.

As the reviewer suggested, we lowered the frame rate of all movies (10 fps). We added time-lapse images of key movies to the corresponding Supplementary Figs and added arrows to the relevant parts, as shown in Supplementary Fig. 4d.

- The sample size in every biological replicates should be described in the figure legends.

In this revised manuscript, we added sample sizes in the figure legends.

Minor concerns:

- In the sentence: "The PM reversibly associates with the actin cortex via linker proteins, such as ezrin, radixin, and moesin (ERM) family proteins, whereby cell membrane deformability is intrinsically dependent on the degree of membrane-cortex adhesion (MCA)^{20, 21}." What do the authors mean with deformability?

We have replaced the word "cell membrane deformability" with "cell membrane mechanics"

-Fig.1a: Why do the authors use the One-way ANOVA with Tukey-s multiple comparisons test if only single comparisons to MCF10A are indicated in the plot?

This is because we also compared the tether force between the ruffling and blebbing cells. The graph shows that there was no significant difference between them.

-The role of cell adhesion in epithelia is under-explored and not really discussed in this manuscript. I agree that experimentally addressing it might be out of the scope of the current work but a paragraph in the discussion on how the balance of forces (membrane

tension vs. adhesion) is key for migration is necessary as one of the main characteristics of epithelial cells is their cell-cell adhesion complexes. Also in Supplementary video 1 Hs578T cells seem to bleb when isolated but ruffle when adherent but it is hard for me to judge without actin and membrane labels (see my Major concern 3). Discussing how the balance of forces could affect protrusion could also be very fitting.

We thank the reviewer for this suggestion. We did not discuss the role of cell adhesion in our original submission, as the inhibitory effect of PM tension on cell migration appears to be independent of cell adhesion, indicating that MCA is a cell-autonomous property. Our data indicated that homeostatic PM tension sustained by MCA was sufficient to suppress epithelial cell migration in a single-cell state and single-cancer cell migration. Nevertheless, we have added a paragraph on cell adhesion to the Discussion section to discuss how our findings may be related to previous studies on cell adhesion and motility (page 17, lines 376–385). In addition, based on the reviewer’s suggestion, the mechanical balance of forces, such as PM tension, adhesion force, and contractility, may be crucial to cell migration (particularly collective cell migration). This point is also mentioned in the Discussion section.

In previous Supplementary Movie 1, the transition between ruffle and bleb seemed to be dependent on cell adhesion; however, this was not statistically significant.

Last, in the text they mention: “E-cadherin depletion had no effect on invasive migration (Fig. 2e, f).” but I could not find the corresponding data.

We apologize for the confusion. In Fig. 2e and f, si-CDH1 (gene name of E-cadherin) corresponds to si-E-cadherin. We have replaced “CDH1” with “E-cadherin” in this figure.

- The images on the left of Extended figure 1D are not described in the legend. They are so small that are hard to see and understand.

We removed this migration data owing to space limitations in Supplementary Fig. 1 caused by the addition of new data. The clear difference in motility between normal and malignant cells is well established. Instead, as mentioned above, we present the quantified data of their protrusion phenotypes in 3D. We believe that these data would be more informative for readers to appreciate the link between PM tension and protrusion formation.

- The schematic in Fig 2A is a massive oversimplification of what we know about ERM regulation. Either simplify further the schematic and not depict the regulation or add more information.

We have added the pathway by which ERM proteins are activated from RHOA via ERM kinases (SLK and STK10), as shown in Supplementary Fig. 1d. We also added the pathway by which RHOA leads to myosin activation via ROCK, making it easier for readers to appreciate the specific contribution of ERM kinases to MCA regulation.

- *The authors write: “Indeed, we also found that knocking down these BAR proteins suppressed an elongated invasive phenotype driven by Snail overexpression (Fig. 6b).” I could not find the corresponding data.*

We apologize for this mistake. The corresponding Fig is Fig. 5b.

- *The statement: “PM tension is regulated by MCA” is misleading as in-plane tension is regulated independently*

We agree with this reviewer’s comment and have added this point and the reason for focusing on MCA as follows (page 6, lines 115–117); “PM tension is regulated by the in-plane tension of the lipid bilayer and MCA. Given that PM tension is largely dependent on MCA under static conditions (such as a non-motile state) (Sheetz MP, *Nat Rev Mol Cell Biol*, 2001, Gauthier et al., *Trends Cell Biol*, 2012), we focused on ERM proteins and F-actin beneath the PM.”

- *The concept “restoring PM tension” is misleading. The authors do not restore PM tension, they increase tension of MDA-MB-231 above their basal level or decrease PM tension of MCF10a below their basal level.*

We agree with this comment and have removed “restoring” and related words from the text.

- *MCF7 cells are labeled as a non-invasive cancer cell line, but they are actually metastatic. This contradicts the following statements in the main text: “ We found that the tether force of MCF10A cells was largely comparable to that of non-invasive human breast cancer cells (AU565 and MCF7)”.*

Although the MCF7 cell line has been reported to be metastatic, it is considered a low-to-non-metastatic cell in many studies and is used to study the mechanism of invasive and metastatic acquisition (Hazan et al., *J Cell Bio*, 2000, PMID: 10684258; Rhodes et al., *Cancer Res*, 2011, PMID: 21123450). Therefore, we believe that our data do not contradict those of previous studies. Nevertheless, to avoid unnecessary misleading, we changed the language of “non-invasive” to “low-invasive” in relation to MCF7 cells.

- The authors write: “The ability to maintain high PM tension appears to be a common characteristic (...)”. Could the authors phrase it in a way to acknowledge that “high MT” is relative to other measurements in this paper, not in general. Many cells lines exhibit even higher values than reported in the paper. Using terms like “high” and “low” without context makes it imprecise (this comment is also applicable to “low” on pages 6 and 9).

We agree with the reviewer’s comment and paraphrased the sentence as follows: “The ability to maintain higher PM tension than malignant cells appears to be a common characteristic of epithelial cells (...)”. We have also made similar corrections to the terms on pages 6 and 9.

- Please specify what kind of image is shown (max intensity projection, mean intensity projection, single slice?) for Fig 1b, 2c, 3b, 4b, 5f and Extended Fig 1e, 6g, 6h, 6k.

We apologize for the lack of information. The images in Figs. 1f and 4e and Supplementary Fig. 6e show the maximum intensity projections (this point was added to the legends), the rest were single slices. These points have been added to the Methods section (page 59, lines 1065–1067).

- Fig 5a,b: Please provide information on what is an “invasive structure” and how as such, was it quantified.

Based on previous studies and Fig. 1 data of epithelial cells, we defined the elongated mesenchymal-like morphology as the invasive structure and the rounded morphology with no protrusion as the non-invasive phenotype, respectively, and quantified them. In addition, we have mentioned in the text that ERM knockdown or Snail expression does not induce round amoeboid-like motility in epithelial cells. We have added these descriptions to the Methods section (page 58, lines 1045–1049).

- Extended figure 6b: Please provide the brief description of how the invasion rates were quantified. Also, as the levels of RhoA in MDA-MB-231 are comparable to MCF10A cells, could the authors perform the invasion assay’s in the siRHOA together with si-BAR domain proteins? Does it mean that Toca proteins and MTSS1L alone increase invasiveness?

We have added this brief description to the legend. As mentioned above, RHOA levels in MDA-MB-231 cells seem to be maintained to allow the rapid transition between motility modes. Our data indicate that the low PM tension-BAR protein axis is required for invasiveness.

- *What are the levels of pERM proteins in all the cell lines in the study?*

We confirmed the total pERM levels in all cells by western blotting. As shown in the Fig. A (right), total pERM levels are likely correlated with the expression level of ERM, and no considerable difference in total pERM levels between these cells was observed, indicating that the pool of pERM at the PM was significantly different in these cells. In addition, this may reflect the local activation of pERM, especially at the cell rear, in migrating cancer cells.

Fig. A Immunoblotting of ERM and pERM in the indicated cells.

Typos:

We thank the reviewer for pointing these errors. We have fixed the text as below.

- *Reference to Fig 2C: “A similar result was observed when ROCK activity was acutely reduced using its specific inhibitor Y27632 (Fig. 2b, right). As expected, RHOA knockdown or Y27632 treatment resulted in the cytoplasmic distribution of ezrin and decreased pERM signal (Fig. 1c).”*

We thank the reviewer for pointing this out. However, we have removed this sentence based on the deletion of the Y27632 data.

- *Fig 2H: single cells.*

Removed.

- *Fig 6b: Plasma membtane tension*

Corrected.

- *Extended Fig 2e, the legend “RhoA” should be “RhoA RNAi”*

Corrected.

- *Scale bars are missing in Fig 1c, Extended Fig 1b and 4b.*

We have included scale bars in the corresponding data.

- On page 10, "We hypothesized that if PM tension reduction is key acquiring (...)", please mind the missing "to".

Corrected.

Reviewer #3 (Remarks to the Author):

Review for Tsujita et al " Homeostatic membrane tension is an intrinsic mechanical suppressor of cancer cell dissemination"

Summary

In this manuscript, Tsujita et al use a number of complementary methods to show that low membrane tension is associated with cell and tumor invasiveness and metastasis. In this sense, the authors describe this mechanical cellular property as being tumor suppressive.

The authors begin by measuring membrane tension (which is determined mostly by membrane to cortex attachment, MCA) using a membrane tether assay in several invasive and non-invasive cancer cell lines. They find that invasive cells, and indeed higher invasiveness in transwell migration assays, is associated with lower membrane tension. Low levels of membrane tension and invasiveness is also associated with reduced levels of F-actin and ERM proteins at the cell cortex and more elongated cell shapes/increased cell protrusions. The authors then show that reducing RhoA pathway activity and ERM activity result in lower membrane tension and increased protrusive and invasive behavior in cells cultured on 2D substrates, under 3D ECM networks or embedded in 3D ECM networks. Similarly, overexpression of the EMT drivers Snail and Slug and ectopic activation of Ras led to lower membrane tension, reduced ERM/F-actin at the cortex and increased protrusive activity. Meta-analysis of gene expression data from The Cancer Genome Atlas are used to show that epithelial tumors are associated with reduced Rho and ERM activity and that Rho/ERM pathways are associated with reduced survival. The authors also find that in most of the invasive cell lines tested, RhoA expression is low. The authors then show that expression of a membrane-targeted (MA-) Ezrin is sufficient to increase membrane tension and cortical F-actin levels. MA-Ezrin expression reduces protrusive activity, blocks cells migration/invasion in vitro and reduces metastasis in a tail vein metastasis assay in mice. The authors identify several BAR domain proteins whose knockdown reduces invasive phenotypes and which localize to protrusions.

This study offers a number of interesting observations, some of which have also been made in other recent publications. The major message of the paper, that mechanical

properties themselves can be thought of as tumor suppressive, is attractive. Although the data presented in the manuscript indeed point in this direction, the physical origins of many of the phenotypes presented in the manuscript are unclear, which makes the overall model difficult to understand.

Major Points

1. In the introduction, the authors define membrane tension as arising largely from membrane to cortex attachment. In much of the paper, the authors focus on the role of ERM proteins, which link the two structures. However, they tend to ignore the structure of the actin cortex itself. In Fig. 1, for example, panel b shows that non-invasive cells (which also have high membrane tension) have low levels of cortical Ezrin. However, they also have considerably less cortical F-actin. It is unclear whether the difference in membrane tension is because Ezrin is not localized to the cortex, or because there is no actin cortex to which the membrane can attach. In the latter case, membrane tension would be expected to be low no matter what the state of ERM proteins. The authors seem to attribute much of their phenotypes to the state of ERM, but it is also likely that the organization of the actin cortex could play as much of a role. In line with this, a recent pre-print (Bisaria et al [2019] BioRxiv: <https://www.biorxiv.org/content/10.1101/705509v1.full>) highlights the role in cortical actin in restricting protrusions by membrane fluctuations. The roles of cortical actin and ERM activity in membrane tension/protrusion formation should be reconciled in order to make sense of the overall model. It would be helpful to address this if relative ERM and cortical actin could be quantified when these images are presented.

This is a very constructive suggestion for our study. As the reviewer pointed out, changes in cortical actin are known to be associated with changes in PM tension (Pontes B et al., *Semin Cell Dev Biol*, 2017 PMID: 28851599). Therefore, as helpfully suggested, we quantified the level of ERM and F-actin beneath the PM and found that, in invasive cells, decreased ERM levels are always correlated with decreased F-actin levels in both 2D and 3D (Fig. 1b–e). Knockdown of ERM proteins or their kinases gave a consistent phenotype (Fig. 2b and c). Notably, there was no decrease in F-actin signals alone, at least in the cancer cells examined. Thus, higher ERM activity appears to lead to higher cortical actin levels, consistent with the notion that ERM proteins play an essential role in the organization of the actin cortex (Fehon et al., *Nat Rev Mol Cell Biol*, 2010 PMID: 20308985). This indicates that the observed difference in PM tension can be primarily due to the dissociation of ERM proteins from the PM, which is in accordance with recent studies that MCA is a critical parameter for the regulation of PM tension (De Belly et al., *Cell Stem Cell*. 2021 PMID: 33217323; Bergert et al., *Cell Stem Cell* 2021 PMID: 33207217). However, in moving cells, it is possible that spatiotemporal changes in the actin cortex and its turnover can affect PM tension, probably independently of

ERM activity. Therefore, we have mentioned the possible involvement of cortical actin in PM tension regulation in the Discussion section (page 16, lines 351–359).

2. In line with the above comments, RhoA siRNA and Y-27632 are mischaracterized as inhibitors of ERM activity. Rho activity has a number of downstream effects, perhaps most prominently on actomyosin itself, as it involved in actin dynamics (via Formins, LIMK/Cofilin and Arp2/3 [due to Rac inhibition]) and myosin activity [via ROCK/Citron Kinase]). Based on previous studies (e.g. Tinevez et al [2010] PNAS), modification of Rho activity or Y-27632 treatment would also be expected to drastically change cortical tension. Previous studies have argued that low cortical tension, following treatment with Y-27632 and Blebbistatin (a myosin-2 inhibitor that would presumably not directly affect ERM activity), can also lead to increased protrusive activity (Bergert et al [2012] PNAS). As actin organization seems to change RhoA knockdown and Y-27632 treatment (Fig. 2c), it is unclear whether the primary defect is really ERM proteins or cortical actin/cortical tension. For a more upstream target of ERM, perhaps siRNA against SLK could be tested instead. In the manuscript, RhoA siRNA and Y-27632 should be better characterized to determine the true origin of reduced membrane tension and invasiveness.

We agree with this concern that RHOA knockdown and Y27632 treatment also affect myosin II activity and thus cortical tension. Therefore, following the reviewer's suggestion, we focused on the ERM-specific kinases SLK and STK10 rather than ROCK. We demonstrated that the co-depletion of SLK and STK10 led to decreased PM tension in MCF10A cells (Fig. 2a). Furthermore, this double knockdown resulted in a drastic increase in their invasive ability (Fig. 2e) and the accumulation of BAR proteins in the PM (Fig. 5f). Importantly, phosphorylated myosin light chain (pS19MLC), a marker for myosin II activation, was unaffected in these knocked-down cells (Supplementary Fig. 2c), supporting our conclusion that the observed phenotypes are dependent on PM tension. Therefore, we replaced the Y27632 data with the ERM kinase data to reflect the role of the MCA. We would like to thank the reviewer for this valuable suggestion.

3. The authors do show that directly abrogating ERM activity by triple siRNA knock-down can lead to lowered membrane tension and increased protrusion (Fig. 2). However, the complication here is that ERM knockdown has also been shown to reduce cortical tension, at least during mitosis (Kunda et al [2008] Curr Biol). In Kunda et al, cells with ERM knockdown fail to round during mitosis and remain flat and protrusive, a phenotype that is copied by knockdown of ROK/Citron and Myosin-2 light chain. Conditions of low cortical tension would therefore be expected to result in less round cell shapes, consistent with the results in this manuscript. Additionally, the MA-Ezrin data presented in Fig. 4 show an increase in cortical actin, and ERM phosphomimic

mutants have been shown to increase cortical tension (Kunda et al [2008] Curr Biol). The loss of protrusions/invasiveness due to MA-Ezrin expression could therefore also be due rather to changes in cortical tension rather than MCA/membrane tension. If the authors would like to argue that membrane tension, and not cortical tension, is the primary physical mechanism, then cortical tension should also be measured. It is indeed possible that cortical tension and MCA/membrane tension are coupled and that these mechanical properties are part of the same mechanical signature linked with invasiveness. In this case, the message of the paper should reflect this and highlight the known role of cortical tension.

This is an important point. Indeed, Kunda et al. reported that upregulation of ERM activity is associated with increased cortical stiffness, which is generally dependent on cortical tension (Salbreux et al., *Trends Cell Biol* 2012, PMID: 22871642). As described above, ERM activation is likely to lead to the enhancement of the actin cortex by increasing MCA, which could contribute to an increase in cortical tension. However, a recent study indicates that increased cortical actin amount does not necessarily lead to increased cortical tension and suggests that ERM-mediated MCA does not seem to affect this tension (Chugh et al., *Nat Cell Biol*, 2017 PMID: 28530659). Moreover, cortical tension is reported to be unchanged in ERM-deficient migrating cells (Diz-Munoz et al., *PLoS Biol*, 2010 PMID: 21151339). In addition, cortical tension is known to play a key role in the plasticity of cell migration (please note that Bergert et al., quoted by the reviewer above, argues that cortical tension plays a role in switching protrusion types rather than initiating them). Indeed, although low cortical tension favors mesenchymal migration, high-tension is required for fast migration modes, such as amoeboid migration (Liu et al., *Cell*, 2015 PMID: 25679760) and slingshot migration (Wang et al., *Nat Commun*, 2019), inconsistent with the idea that cortical tension acts as the inhibitor of cancer cell migration. Nevertheless, we performed some experiments to clarify whether PM tension or cortical tension plays a primary role in suppressing cancer cell motility, as described below.

First, we investigated whether an increase in PM tension without manipulating ERM activity suppressed invasive migration. It has been reported that cholesterol depletion by methyl- β -cyclodextrin (M β CD) treatment increases PM tension, presumably independently of ERM activity (Pontes B et al., *Semin Cell Dev Biol*, 2017). Interestingly, this treatment is reported to decrease cortical tension (Zanotelli MR et al., *Nat Commun*, 2019 PMID: 31519914). We established that this treatment led to a significant increase in PM tension in MDA-MB-231 cells (Supplementary Fig. 4c). Furthermore, this treatment not only suppressed their invasive ability (Supplementary Fig. 4b), but also caused the dissociation of BAR protein from the PM (Supplementary Fig. 6i and j), supporting the notion that PM tension is the dominant mechanical factor that suppresses membrane protrusions by counteracting BAR proteins.

Second, we investigated whether increased cortical tension inhibited invasive migration. As shown in Supplementary Fig. 4b, the treatment of MDA-MB-231 cells with calyculin A, which is commonly used to increase cortical tension by increasing myosin II activity, has no effect on their invasive ability, reflecting the plasticity of cancer cell migration due to changes in contractility. Importantly, although cortical tension of blebbing cells is significantly higher than that of actin-based protruding cells (Bergert et al., *PNAS* 2012), we found that there was no difference in PM tension between them (Fig. 1a). This is further supported by our data that BAR proteins accumulate in both types of protrusions (Fig. 5g and Supplementary Fig. 6f). Moreover, it should be noted that MA-ezrin expression was sufficient to suppress amoeboid migration, which presumably has higher cortical tension.

Therefore, we believe that our data indicate that PM tension plays a primary role in suppressing both types of protrusions, and thus cancer cell migration. In addition, given the importance of cortical tension in the plasticity of migration, a possible relationship between PM tension and cortical tension has been discussed in the Discussion section, as described below (comment 4).

4. In the 3D collagen/Matrigel assays, the authors show a convincing link between RhoA/ERM loss and protrusive activity. It would be helpful to have some quantification of the migration to better interpret how differences in protrusions results in increased migration capacity. Again, these migration data should be compared with recent studies that have shown that inhibition of myosin-2 activity can lead to increased protrusions and increased single or collective migration (Liu et al [2010] Br J Pharmacol, Xu et al [2019] Nat Commun). These data should also at least be discussed in the context of results showing that the effects of increasing or decreasing cortical tension on cell migration depends on ECM network properties (e.g., Wang et al [2019] Nat Commun).

The reviewer has raised a good point. An important question raised by our study is how decreased PM tension leads to two different types of protrusion formation and subsequent mesenchymal and amoeboid migration modes. We have quantitatively shown that, in epithelial cells, the overall reduction in MCA by knockdown of ERM or their kinases induces only actin-based protrusion and slow mesenchymal-like motility (Fig. 2g and h). This suggests that a local increase in MCA, especially at the cell rear, may be required for fast amoeboid migration. In addition, as mentioned above, differences in cortical tension are likely to be important for determining migration modes. Indeed, experimental study and mathematical models suggest that decreased PM tension and increased cortical tension favor bleb formation and bleb-based migration (Diz-Munoz et al., *PloS Biol*, 2010, Sens et al., *J Phys Condens Matter*, 2015 PMID: 26061624). Consistently, it is known that reduced myosin II activity favors actin-based

protrusion and migration (Bergert et al., *PNAS* 2012), as the reviewer pointed out. Again, our data and these findings suggest that reduced PM tension is a prerequisite for both types of protrusion formation and that cortical tension is important for their switching. Related to the work of Wang et al, it is known that extracellular matrix (ECM) stiffness and architecture have a profound effect on cell motility. It will be very important to investigate whether and how ECM stiffness or properties can affect MCA and thus PM tension. Moreover, elucidating how the balance of PM tension and cortical tension—presumably regulated by the microenvironment—controls cancer cell motility will be an important question for future studies. These points are mentioned in the Discussion section (page 16, lines 359–375).

5. In the case of induced EMT by overexpression of Snail/Slug and expression of K-Ras G12V, it appears that F-actin organization is substantially changed, which again brings up the question of whether the primary change is the loss of actin cortex organization or MCA.

In response to comment 1, our data indicate that a decrease in PM tension is primarily due to the loss of MCA.

6. In Fig. 3a-3c, the authors focus on EMT as a trigger of low membrane tension/high invasiveness. For the EMT conditions (Snail/Slug expression), would it be possible to check for EMT by staining/Western blotting for classical EMT markers, as in Extended Fig. 2C? Also, in the control cells shown in Fig. 3b, are these phenotypes really epithelial? The cells also seem reasonably well-spread, and it is not clear if they are Apicobasally polarized.

We confirmed EMT by western blotting for E-cadherin and vimentin, as shown in Supplementary Fig. 2c.

We thank the reviewer for raising this concern. As precisely pointed out by the reviewer, the control images in the previous Fig. 3b did not reflect the typical cell appearance. Therefore, we replaced the data with more typical images.

7. In the model cartoon on Fig. 6b, the authors suggest a reversible conversion between static epithelial cells and motile cells. However, this does not really reflect the results in the manuscript. Many of the experiments show interconversion between more round and more elongated/protrusive cell shapes. However, a round shape is not enough to characterize a cell as “epithelial.” In the Rho/ERM knock-downs, the authors show that cells do not lose their epithelial characteristics despite being more invasive (Extended Fig. 2c). Also, there is no evidence that the conditions that force cells to assume a round shape make them more epithelial.

We agree with the reviewer's comment and have removed the relevant parts.

Minor Points

1. In Fig. 1b, it would be helpful if the images from different cell lines are presented in the same order as in Fig. 1a.

Fixed.

2. In Extended Data Fig. 1d: Is the line a linear fit of the data? It does not seem to fit very well. The data look rather like 2 distinct populations than a linear continuum. The fit seems unnecessary and confusing.

We agree with this comment. However, we have removed these data due to space limitations in Supplementary Fig. 1. The clear difference in motility between normal and malignant cells is well established. Instead, we presented the quantified data of their protrusion phenotypes in 3D. We believe that these data would be more informative for readers to appreciate the link between PM tension and protrusion formation.

3. In the results section, the authors claim that their results regarding the connection between loss of RhoA/ERM activity and invasiveness is contrary to common belief (“Surprisingly, despite that ERM proteins and RHOA are commonly thought to play an active role in cancer cell invasion^{36, 37}”). However, several more recent studies have also linked loss of RhoA to invasiveness, and the authors cite this in the Discussion section (“recent studies have indicated that RHOA plays a tumor suppressor role in many cancers by inhibiting invasion and metastasis⁶²⁻⁶⁶”). This fact should be acknowledged when motivating the data in the Results section to accurately reflect the state-of-the-art in the field, as the results are less surprising in light of these recent studies.

We agree with this reviewer's comment and have removed this sentence from the Results section. These RHOA papers are quoted in this section accordingly (page 7, line 150). In addition, please note that the discussion of the RHOA part has been removed, as the RHOA conclusions were toned down in response to reviewer #2's suggestion.

4. I find the following sentence slightly confusing/misleading: “...epithelial tumors frequently and exclusively harbor putative heterozygous deletions of RHOA and ERM, as opposed to metastasis-associated genes, such as SNAIL.” If I am understanding the plots correctly in Fig. 3d and Extended Fig. 3a, it seems like loss of RhoA is just a frequently found as gain of SNAIL.

We apologize for this confusion. Reviewer #2 made a similar point. Therefore, we have removed this sentence to avoid unnecessary confusion.

5. Related to the following sentence: “...knockdown of ERM, RHOA, or EMT induction all resulted in the accumulation of GFP-FBP17 at the PM where it appeared to activate Arp2/3 complex-dependent actin nucleation for the directed migration.” – Although it is cited in the manuscript that MTSS1L and Toca proteins can activate Arp2/3, there is no evidence in Fig. 5 suggesting that is what is happening in this case. It would be better to remove the suggestion of Arp2/3 activation unless the authors can provide evidence of that.

We have removed this sentence from the text.

REVIEWER COMMENTS

Reviewer #1 (Remarks to the Author):

The authors have carefully addressed all of my comments with either new experiments, quantifications, figure reorganization or thoughtful answers. I believe this manuscript is significantly improved.

Jean-Francois Cote

Reviewer #2 (Remarks to the Author):

Tsujita et al have provided a much improved version of their manuscript entitled: "Homeostatic membrane tension acts as a mechanical tumor suppressor and constrains cancer cell dissemination". The story is clearer and the simplification of data in several instances makes it easier to follow, which I think grant publication in Nature Communications. Nevertheless I think there are several points that need to be clarified in the text and methods before publication. Last, I would suggest repeating one set of experiments to support their claims.

Major comment regarding text:

- In several instance the authors cite reviews instead of primary research to support their claims. For example in line 117 (this is particularly a concerning one as these reviews (refs. 22, 23) are not even supporting their claims). This happens again in lines 243 and 398, where they cite reviews (refs. 24 and 47) instead of showing the primary data paper.
- How the quantifications of ERM and pERM are done is really not clear in the methods section. More detail should be provided to not compromise their results. How do the authors determine the lateral side (as quantified in SF1f)? Do they measure both sides or only one? Why is only one plane quantified from 3D stacks? How is the cell segmentation done and how many pixels are considered "edge" (SF1g) or "membrane" (Fig 1c,e, 2c,i) by the authors?
- The authors should specify how the plane displayed in figures is selected from 3D stacks.
- The quantification of ERM in 3D is not sufficient to make the claim that membrane tension is changed (in support to the point raised by reviewer 1). I suggest the authors tone down their claims.
- The experiments with cyclodextrin are not a clean way to perturb membrane tension. The authors should consider toning down their claims even though these experiments are crucial to their rebuttal to reviewer 1. As written in the review the authors cite (ref 24): "Reducing plasma membrane cholesterol with methyl-beta cyclodextrin causes an increase in membrane tension in embryonic kidney cells, fibroblasts and cardiomyocytes. However, this increase is not only due to changes in membrane composition, but also correlates with cytoskeletal rearrangements. Upon methyl-beta cyclodextrin, fibroblasts showed de novo actin polymerization and Rho-GTPase dependent stress fiber formation whereas cardiomyocytes showed sarcomeric disorganization accompanied by contraction abnormalities".
- Line 133: "In both cases, membrane-proximal ERM and F-actin displayed overall decreased levels, although their local accumulation was observed to be the same as that in 2D (Fig. 1d-f)." How is the local accumulation assessed here? Can the author clarify what they mean?
- Line 170: E-cadherin depletion shows a significant decrease in invasion and migration (Fig 2e and f) so that sentence should be rewritten.
- Line 223: "We hypothesized that if PM tension reduction is key to acquiring migration and invasiveness, increasing cell membrane mechanics might be sufficient to suppress cancer cell dissemination." This sentence is very vague... Can the authors rewrite "increasing cell membrane mechanics" by a more precise term? Like increasing MCA?
- Line 251: "We found that changes in ERM levels at the PM were always correlated with changes in membrane-proximal F-actin levels; this indicated that MCA is the key determinant of PM tension, as recently described in other experimental settings". First, how does a correlation between ERM levels and membrane-proximal F-actin indicate that MCA is the key determinant of PM tension? The logic here is not clear and the reference not appropriate.
- Line 374: "It will thus be very important to investigate whether and how their properties can affect MCA and PM tension". Substrate stiffness has been shown to not affect MCA in mESC (Bergert, Lembo, 2021, Cell Stem Cell). The authors should rephrase that section of the discussion

and include this paper or remove the paragraph.

- In that same paper (Bergert, Lembo, 2021, Cell Stem Cell) a cleaner way to perturb MCA is described (iMC-linker). The authors should discuss the perturbations they use and their limitations in light of this new gold standard.

Suggested last experiments:

- The authors quantify ERM instead of pERM in Fig 1c,e, 2c,i and 3b,f. They suggest in their rebuttal letter than this is due to the fixation method required but there are antibodies in the market that would be compatible with concomitant phalloidin staining. I think these last experiments are key to supporting their claims.

Minor comment:

- Line 115: PM tension is regulated by the in-plane tension of the lipid bilayer and MCA. Technically it is not "regulated" but "has contributions from".

- Fig 4k and SF4f. The N for Ezrin is really low and specifically for 4f the spread of the data is very large. I would suggest considering removing the Ezrin data as it does not bring much to the paper and the N and spread of the data compromises the conclusion.

- Fig. 1d and f: Why display in one main figure both a single plan and the MIP? I would suggest the authors quantify the localisation in all planes and only display 1f.

- Typo in line 158. "the" is not needed.

- English in line 165 is confusing. "These" implies the control cells from the sentence above but I believe the authors mean the the low-tension cells.

- Fig 3D is not referred to in the main text.

- SF4c. Is Mock DMSO? if so can the authors write that?

- Fig 4e. What is the intracellular structure labelled in the PM channel? Is it labelled with WGA?

- Fig 4k. Why do here they show p-values instead of stars like in the rest of the paper?

- Line 264: Why is it unexpected? can the authors clarify? do they refer to the proliferation data?

- Is the data in SF5 needed? It does not really fit to Line 282.

- Fig 5a. Can they include a color legend?

- Reference 12 is not formatted appropriately.

Reviewer #3 (Remarks to the Author):

After reading the new version of the manuscript and the response to the reviewers' concerns, I found that the authors did a truly exceptional job in addressing all concerns. The authors performed a number of new experiments and quantifications that fit well with the previous data and help to support the overall conclusions of the paper. The authors make a very convincing claim that ERM-mediated MCA, and not just the organization of the actin cortex, is really the crucial parameter to the membrane tension phenotype they observe and attribute to invasiveness. I find the new discussion about PM tension being a prerequisite for all protrusion formation and cortical tension rather being involved in modulating protrusion type especially compelling, and it seems well supported by the data.

I am still, however, a little confused by the model. I understand that BAR protein accumulation/oligomerization is linked to protrusive activity and invasiveness, but it is not really clear in the text what is the causal link. I guess that it is probably attributed to activation of actin nucleation, as the authors previously showed that FBP17 leads to actin polymerization via N-WASP. The authors also provide references in the text showing that MTSSL1 and Toca proteins are also involved in Arp2/3-mediated actin polymerization. I think this could be incorporated into the model figure to clarify the causal link between BAR/Toca proteins and invasiveness. Also, the authors say in the introduction "Given that tense membranes resist cell membrane deformations, PM tension is assumed to disfavor the formation of any membrane protrusions." This suggests that the barrier to making protrusions is the mechanical bending stiffness of the PM. However, the results in the paper reveal a more complicated mechanism involving curvature-sensing proteins. I think it would be good if the authors could come back to this idea in the Discussion just to clarify this for the readers. Other than that, I have no further suggestions, and I congratulate the authors on a really fantastic revision of this manuscript.

We would like to thank the reviewers for their helpful and constructive comments. We
have addressed the additional points raised below (our responses in blue). Changes in
the manuscript text are also highlighted in blue.

*REVIEWER COMMENTS*

*Reviewer #1 (Remarks to the Author):*

*The authors have carefully addressed all of my comments with either new experiments,*
*quantifications, figure reorganization or thoughtful answers. I believe this manuscript is*
*significantly improved.*

*Jean-Francois Cote*

We thank the reviewer for your time and dedication in carefully reviewing our study.

Your valuable suggestions helped us much improve the manuscript.

*Reviewer #2 (Remarks to the Author):*

*Tsujita et al have provided a much improved version of their manuscript entitled:*
*“Homeostatic membrane tension acts as a mechanical tumor suppressor and constrains*
*cancer cell dissemination”. The story is clearer and the simplification of data in several*
*instances makes it easier to follow, which I think grant publication in Nature*
*Communications. Nevertheless I think there are several points that need to be clarified*
*in the text and methods before publication. Last, I would suggest repeating one set of*
*experiments to support their claims.*

We thank the reviewer for the careful reading of our manuscript and the many helpful
comments. In particular, we believe that new pERM data provide further support for our
model.

*Major comment regarding text:*

*- In several instance the authors cite reviews instead of primary research to support*
*their claims. For example in line 117 (this is particularly a concerning one as these*

*reviews (refs. 22, 23) are not even supporting their claims). This happens again in lines*
*243 and 398, where they cite reviews (refs. 24 and 47) instead of showing the primary*
*data paper.*

We agree with this reviewer's comment and have changed the references corresponding
to the relevant parts to the primary data papers as follows.

Line 117: refs. 25-27. In addition, we have removed the sentence "under static
conditions", as they do not mention this.

Line 238 (previous line 243): refs. 23 and 47.

Line 399 (previous line 398): please note that this (ref. 49) is the primary research
paper.

We also have replaced the review with the research papers (refs. 60–63: line 345). In
addition, we have added several important research papers that we had overlooked in
the previous manuscript (refs. 29, 36, 56, 68).

*- How the quantifications of ERM and pERM are done is really not clear in the methods*
*section. More detail should be provided to not compromise their results. How do the*
*authors determine the lateral side (as quantified in SF1f)? Do they measure both sides*
*or only one? Why is only one plane quantified from 3D stacks? How is the cell*
*segmentation done and how many pixels are considered "edge" (SF1g) or "membrane"*
*(Fig 1c,e, 2c,i) by the authors?*

Firstly, please note that we have changed all ERM data to pERM data according to the
reviewers' advice.

Line profiles for the PM, pERM and F-actin levels were measured across one lateral
side, as indicated by the white lines in SF1e. In the case of cells that do not have a clear
membrane extension or cell rear, such as MCF10A cells, the measurement points were
arbitrarily determined. Please note that pERM and F-actin are distributed globally in
these cells.

We quantified one plane image near the middle of 3D stacks where the membrane
region can be clearly defined by WGA staining, since, as the reviewer pointed out in the
first revision, membrane staining must be included when quantifying pERM and F-actin
levels in the PM. However, as the reviewer pointed out, we also had to account for any

variation in fluorescence intensity due to the thickness of 3D stacks. Therefore, two
images shifted up and down in the z-direction (within the range where the membrane
region could be identified) from the selected middle image were also quantified, and the
average of the total three quantitative data was used as one sample. These quantification
methods, image segmentation, and pixel sizes are described in the Methods Section
(page 66, lines 1072–1084).

- *The authors should specify how the plane displayed in figures is selected from 3D*
*stacks.*

As mentioned above, we selected the plane image with clearly distinguishable
membrane regions that corresponds to near the middle of 3D stacks. This is now
described in the Methods Section (page 66, line 1059).

- *The quantification of ERM in 3D is not sufficient to make the claim that membrane*
*tension is changed (in support to the point raised by reviewer 1). I suggest the authors*
*tone down their claims.*

We agree with the reviewer’s suggestion and have removed the relevant sentence
(previous lines 137–139) and rewritten the sentence as follows: “These results suggest
that the observed differences in PM tension between non-motile and invasive cells are
primarily due to ERM-mediated changes in MCA...” (page 8, line 134).

- *The experiments with cyclodextrin are not a clean way to perturb membrane tension.*
*The authors should consider toning down their claims even though these experiments*
*are crucial to their rebuttal to reviewer 1. As written in the review the authors cite (ref*
*24): “Reducing plasma membrane cholesterol with methyl-beta cyclodextrin causes an*
*increase in membrane tension in embry-onic kidney cells, fibroblasts and*
*cardiomyocytes. However, this increase is not only due to changes in membrane*
*composition, but also correlates with cytoskeletal rearrangements. Upon methyl-beta*
*cyclodextrin, fibroblasts showed de novo actin polymerization and Rho-GTPase*
*dependent stress fiber formation whereas cardiomyocytes showed sarcomeric*
*disorganization accompanied by contraction abnormalities”.*

As suggested by the reviewer, we have deleted the relevant sentence (previous lines

244–245) to tone down our claims based on M β CD data. Instead, we have added a text
as follows: “ruling out potential effects of cortical stiffness on inhibiting cancer cell
migration” (page 14, line 239).

- *Line 133: “In both cases, membrane-proximal ERM and F-actin displayed overall*
*decreased levels, although their local accumulation was observed to be the same as that*
*in 2D (Fig. 1d–f).” How is the local accumulation assessed here? Can the author*
*clarify what they mean?*

In this sentence, we wanted to point out that, as in 2D, the pERM signal is enriched in
the rear and blebbing membranes. However, since what we want to show by these data
is that pERM levels are overall higher in non-invasive cells than in invasive cells, we
have deleted this unnecessary sentence.

- *Line 170: E-cadherin depletion shows a significant decrease in invasion and*
*migration (Fig 2e and f) so that sentence should be rewritten.*

We have added a sentence as follows: “E-cadherin depletion resulted in a slight but
significant inhibition of invasion and migration” (page 10, line 166).

- *Line 223: “We hypothesized that if PM tension reduction is key to acquiring migration*
*and invasiveness, increasing cell membrane mechanics might be sufficient to suppress*
*cancer cell dissemination.” This sentence is very vague... Can the authors rewrite*
*“increasing cell membrane mechanics” by a more precise term? Like increasing MCA?*

We thank the reviewer for this suggestion and have replaced it with “increasing MCA”
(page 13, line 219).

- *Line 251: “We found that changes in ERM levels at the PM were always correlated*
*with changes in membrane-proximal F-actin levels; this indicated that MCA is the key*
*determinant of PM tension, as recently described in other experimental settings”. First,*
*how does a correlation between ERM levels and membrane-proximal F-actin indicate*
*that MCA is the key determinant of PM tension? The logic here is not clear and the*
*reference not appropriate.*

We agree with the reviewer’s comment that the logic of this sentence is not clear.

Therefore, we have rephrased it to the following sentence to account for the flow of our
discussion: “Our data indicate that ERM-mediated MCA is responsible for maintaining
homeostatic PM tension in non-invasive cells” (page 20, lines 356).

- *Line 374: “It will thus be very important to investigate whether and how their
properties can affect MCA and PM tension”. Substrate stiffness has been shown to not
affect MCA in mESC (Bergert, Lembo, 2021, Cell Stem Cell). The authors should
rephrase that section of the discussion and include this paper or remove the paragraph.*

We agree with this reviewer’s comment and have removed this paragraph.

- *In that same paper (Bergert, Lembo, 2021, Cell Stem Cell) a cleaner way to perturb
MCA is described (iMC-linker). The authors should discuss the perturbations they use
and their limitations in light of this new gold standard.*

We have mentioned in the Discussion Section about potential limitations of our study
using MA-ezrin and the need for more studies using this new tool (page 20, lines 347–
355). Thank you for this suggestion.

*Suggested last experiments:*

- *The authors quantify ERM instead of pERM in Fig 1c,e, 2c,i and 3b,f. They suggest in
their rebuttal letter than this is due to the fixation method required but there are
antibodies in the market that would be compatible with concomitant phalloidin staining.
I think these last experiments are key to supporting their claims.*

As the reviewer indicated, we confirmed that another anti-pERM antibody provided by
CST (#3726) accurately stains formaldehyde-fixed cells. Therefore, all representative
and quantitative data in Fig. 1 (b-e), SFig. 1 (e, f, and g), Fig. 2 (b, c, h, and i), and Fig.
3 (a, b, e, and f) have been changed to pERM (from ERM). Please note that previous
SFig. 3c and d have been removed, because canine MDCK cells were not stained with
this antibody. We believe that these new data provide further support for the importance
of ERM-mediated MCA in suppressing cancer cell motility.

*Minor comment:*

- *Line 115: PM tension is regulated by the in-plane tension of the lipid bilayer and MCA.
Technically it is not “regulated” but “has contributions from”.*

We have rewritten it as the reviewer suggested (page 7, line 115).

- Fig 4k and SF4f. The N for Ezrin is really low and specifically for 4f the spread of the
data is very large. I would suggest considering removing the Ezrin data as it does not
bring much to the paper and the N and spread of the data compromises the conclusion.

As the reviewer suggested, we have removed the Ezrin data from the Fig 4k and SF4f.

- Fig. 1d and f: Why display in one main figure both a single plan and the MIP? I would
suggest the authors quantify the localisation in all planes and only display 1f.

We believe that a single plane showing a distinct membrane region is necessary for
readers to appreciate these data accurately. MIP is suitable for displaying the entire 3D
image but does not allow for the extraction of the cell membrane region. In addition, as
described above, we quantified the intensity of pERM and F-actin in three planes.
Therefore, we would like to respectfully stand by our way of displaying both
single-plane and MIP images in Fig.1d in this revised manuscript.

- Typo in line 158. "the" is not needed.

We thank the reviewer for pointing this mistake. However, we have changed the
relevant part as shown in blue (page 9, line 154; previous line 158), as the ERM data
were removed.

- English in line 165 is confusing. "These" implies the control cells from the sentence
above but I believe the authors mean the the low-tension cells.

We apologize for the confusion. As pointed out by the reviewer, "the" is correct. We
have corrected this part (page 9, line 160).

- Fig 3D is not referred to in the main text.

We thank the reviewer for pointing this out. Fig.3D is now referred to in the text.

- SF4c. Is Mock DMSO? if so can the authors write that?

We apologize for the lack of information. Mock is water, as M β CD was dissolved in
water. This is now described in the Figure legend.

- *Fig 4e. What is the intracellular structure labelled in the PM channel? Is it labelled*
*with WGA?*

Yes, this was labelled with WGA, as indicated in the legend. We notice that in cancer
cells, WGA tends to stain the inner membrane structures of the cell, even before
permeabilization.

- *Fig 4k. Why do here they show p-values instead of stars like in the rest of the paper?*

This was because the p-value of Fig.4k is higher than the rest of the key data. We have
included a star in Fig 4k and the exact p-value in the corresponding legend.

- *Line 264: Why is it unexpected? can the authors clarify? do they refer to the*
*proliferation data?*

As pointed out by the reviewer, we referred to the in vitro proliferation data (SF4a).
Therefore, we have added the following sentence to this part (page 15, line 259;
previous line 264): “although they exhibited the proliferation rate comparable to that of
parental cells in vitro”.

- *Is the data in SF5 needed? It does not really fit to Line 282.*

We agree with the reviewer’s suggestion and have removed the previous Supplementary
Fig. 5.

- *Fig 5a. Can they include a color legend?*

We have added the description to the corresponding legend (page 42, line 729).

- *Reference 12 is not formatted appropriately.*

We thank the reviewer for pointing this out. However, we believe that this format is

correct, as this study was conducted by a collaborative network named “Physical
Sciences-Oncology Centers”.

*Reviewer #3 (Remarks to the Author):*

*After reading the new version of the manuscript and the response to the reviewers’*
*concerns, I found that the authors did a truly exceptional job in addressing all concerns.*
*The authors performed a number of new experiments and quantifications that fit well*
*with the previous data and help to support the overall conclusions of the paper. The*
*authors make a very convincing claim that ERM-mediated MCA, and not just the*
*organization of the actin cortex, is really the crucial parameter to the membrane tension*
*phenotype they observe and attribute to invasiveness. I find the new discussion about*
*PM tension being a prerequisite for all protrusion formation and cortical tension rather*
*being involved in modulating protrusion type especially compelling, and it seems well*
*supported by the data.*

*We thank the reviewer for the critical reading of our manuscript and rebuttal, and for*
*supporting the overall conclusions of our paper.*

*I am still, however, a little confused by the model. I understand that BAR protein*
*accumulation/oligomerization is linked to protrusive activity and invasiveness, but it is*
*not really clear in the text what is the causal link. I guess that it is probably attributed to*
*activation of actin nucleation, as the authors previously showed that FBP17 leads to*
*actin polymerization via N-WASP. The authors also provide references in the text*
*showing that MTSSL1 and Toca proteins are also involved in Arp2/3-mediated actin*
*polymerization. I think this could be incorporated into the model figure to clarify the*
*causal link between BAR/Toca proteins and invasiveness. Also, the authors say in the*
*introduction “Given that tense membranes resist cell membrane deformations, PM*
*tension is assumed to disfavor the formation of any membrane protrusions.” This*
*suggests that the barrier to making protrusions is the mechanical bending stiffness of*
*the PM. However, the results in the paper reveal a more complicated mechanism*
*involving curvature-sensing proteins. I think it would be good if the authors could come*
*back to this idea in the Discussion just to clarify this for the readers. Other than that, I*
*have no further suggestions, and I congratulate the authors on a really fantastic*
*revision of this manuscript.*

We thank the reviewer for these valuable comments. As suggested by the reviewer, we
have added a pathway in which a decrease in PM tension leads to invasiveness by
promoting Arp2/3-dependent actin polymerization via activation of MTSS1L and Toca
proteins (Fig. 6a). We have rewritten the Discussion Section to make it easier for the
readers to appreciate that the mechanical coupling between low PM tension and BAR
proteins, rather than just decreased tension, is important for membrane protrusions and
motility phenotypes (page 19, lines 330–335). In addition, we have changed the title as
follows so that the readers can understand at a glance that the tension-mediated
suppression of cancer cell motility is due to the inhibition of BAR proteins:
**“Homeostatic membrane tension constrains cancer cell dissemination by**
**counteracting BAR proteins”**.

REVIEWERS' COMMENTS

Reviewer #3 (Remarks to the Author):

Please see the attached document for my comments. Overall, the authors did a nice job responding to the reviewers' concerns, and there are just a few very minor changes I think could be made prior to publication.

Response to second rebuttal (Reviewer 3):

As I have been asked to comment on the responses to both my concerns and the
concerns of Reviewer 2, I have written my comments below in a point-by-point manner
with ** at the beginning of my comments and in red. Overall, the authors did a nice job
responding to the reviewers' concerns, and there are just a few very minor changes I
think could be made prior to publication.

We would like to thank the reviewers for their helpful and constructive comments. We
have addressed the additional points raised below (our responses in blue). Changes in
the manuscript text are also highlighted in blue.

*REVIEWER COMMENTS*

*Reviewer #1 (Remarks to the Author):*

*The authors have carefully addressed all of my comments with either new experiments,*
*quantifications, figure reorganization or thoughtful answers. I believe this manuscript is*
*significantly improved.*

*Jean-Francois Cote*

We thank the reviewer for your time and dedication in carefully reviewing our study.
Your valuable suggestions helped us much improve the manuscript.

*Reviewer #2 (Remarks to the Author):*

*Tsujita et al have provided a much improved version of their manuscript entitled:*
*“Homeostatic membrane tension acts as a mechanical tumor suppressor and constrains*
*cancer cell dissemination”. The story is clearer and the simplification of data in several*
*instances makes it easier to follow, which I think grant publication in Nature*
*Communications. Nevertheless I think there are several points that need to be clarified*
*in the text and methods before publication. Last, I would suggest repeating one set of*
*experiments to support their claims.*

We thank the reviewer for the careful reading of our manuscript and the many helpful
comments. In particular, we believe that new pERM data provide further support for our
model.

*Major comment regarding text:*

- *In several instance the authors cite reviews instead of primary research to support*
*their claims. For example in line 117 (this is particularly a concerning one as these*
*reviews (refs. 22, 23) are not even supporting their claims). This happens again in lines*
*243 and 398, where they cite reviews (refs. 24 and 47) instead of showing the primary*
*data paper.*

We agree with this reviewer's comment and have changed the references corresponding
to the relevant parts to the primary data papers as follows.

Line 117: refs. 25-27. In addition, we have removed the sentence "under static
conditions", as they do not mention this.

Line 238 (previous line 243): refs. 23 and 47.

Line 399 (previous line 398): please note that this (ref. 49) is the primary research
paper.

We also have replaced the review with the research papers (refs. 60–63: line 345). In
addition, we have added several important research papers that we had overlooked in
the previous manuscript (refs. 29, 36, 56, 68).

****This seems better now with the primary literature.**

- *How the quantifications of ERM and pERM are done is really not clear in the methods*
*section. More detail should be provided to not compromise their results. How do the*
*authors determine the lateral side (as quantified in SF1f)? Do they measure both sides*
*or only one? Why is only one plane quantified from 3D stacks? How is the cell*
*segmentation done and how many pixels are considered "edge" (SF1g) or "membrane"*
*(Fig 1c,e, 2c,i) by the authors?*

Firstly, please note that we have changed all ERM data to pERM data according to the
reviewers' advice.

Line profiles for the PM, pERM and F-actin levels were measured across one lateral

side, as indicated by the white lines in SF1e. In the case of cells that do not have a clear
membrane extension or cell rear, such as MCF10A cells, the measurement points were
arbitrarily determined. Please note that pERM and F-actin are distributed globally in
these cells.

We quantified one plane image near the middle of 3D stacks where the membrane
region can be clearly defined by WGA staining, since, as the reviewer pointed out in the
first revision, membrane staining must be included when quantifying pERM and F-actin
levels in the PM. However, as the reviewer pointed out, we also had to account for any
variation in fluorescence intensity due to the thickness of 3D stacks. Therefore, two
images shifted up and down in the z-direction (within the range where the membrane
region could be identified) from the selected middle image were also quantified, and the
average of the total three quantitative data was used as one sample. These quantification
methods, image segmentation, and pixel sizes are described in the Methods Section
(page 66, lines 1072–1084).

****This seems clear in the figure. In my opinion doing this on an equatorial slice is the
correct way to measure, especially for the linescans, as projections or measurements on
non-equatorial slice would lead to artifacts due to the cell curvature.**

*- The authors should specify how the plane displayed in figures is selected from 3D
stacks.*

As mentioned above, we selected the plane image with clearly distinguishable
membrane regions that corresponds to near the middle of 3D stacks. This is now
described in the Methods Section (page 66, line 1059).

****This seems fine. Normally it is very easy to select an equatorial slice by eye from a
3D stack.**

*- The quantification of ERM in 3D is not sufficient to make the claim that membrane
tension is changed (in support to the point raised by reviewer 1). I suggest the authors
tone down their claims.*

We agree with the reviewer’s suggestion and have removed the relevant sentence
(previous lines 137–139) and rewritten the sentence as follows: “These results suggest
that the observed differences in PM tension between non-motile and invasive cells are
primarily due to ERM-mediated changes in MCA...” (page 8, line 134).

****This change seems appropriate.**

- *The experiments with cyclodextrin are not a clean way to perturb membrane tension.*
*The authors should consider toning down their claims even though these experiments*
*are crucial to their rebuttal to reviewer 1. As written in the review the authors cite (ref*
*24): “Reducing plasma membrane cholesterol with methyl-beta cyclodextrin causes an*
*increase in membrane tension in embry-onic kidney cells, fibroblasts and*
*cardiomyocytes. However, this increase is not only due to changes in membrane*
*composition, but also correlates with cytoskeletal rearrangements. Upon methyl-beta*
*cyclodextrin, fibroblasts showed de novo actin polymerization and Rho-GTPase*
*dependent stress fiber formation whereas cardiomyocytes showed sarcomeric*
*disorganization accompanied by contraction abnormalities”.*

As suggested by the reviewer, we have deleted the relevant sentence (previous lines
244–245) to tone down our claims based on MβCD data. Instead, we have added a text
as follows: “ruling out potential effects of cortical stiffness on inhibiting cancer cell
migration” (page 14, line 239).

****This seems like an appropriate change.**

- *Line 133: “In both cases, membrane-proximal ERM and F-actin displayed overall*
*decreased levels, although their local accumulation was observed to be the same as that*
*in 2D (Fig. 1d–f).” How is the local accumulation assessed here? Can the author*
*clarify what they mean?*

In this sentence, we wanted to point out that, as in 2D, the pERM signal is enriched in
the rear and blebbing membranes. However, since what we want to show by these data
is that pERM levels are overall higher in non-invasive cells than in invasive cells, we
have deleted this unnecessary sentence.

****Seems appropriate.**

- *Line 170: E-cadherin depletion shows a significant decrease in invasion and*
*migration (Fig 2e and f) so that sentence should be rewritten.*

We have added a sentence as follows: “E-cadherin depletion resulted in a slight but
significant inhibition of invasion and migration” (page 10, line 166).

****This new sentence accurately reflects the data.**

- *Line 223: “We hypothesized that if PM tension reduction is key to acquiring migration*
*and invasiveness, increasing cell membrane mechanics might be sufficient to suppress*
*cancer cell dissemination.” This sentence is very vague... Can the authors rewrite*
*“increasing cell membrane mechanics” by a more precise term? Like increasing MCA?*

We thank the reviewer for this suggestion and have replaced it with “increasing MCA”
(page 13, line 219).

****Good.**

- *Line 251: “We found that changes in ERM levels at the PM were always correlated*
*with changes in membrane-proximal F-actin levels; this indicated that MCA is the key*
*determinant of PM tension, as recently described in other experimental settings”. First,*
*how does a correlation between ERM levels and membrane-proximal F-actin indicate*
*that MCA is the key determinant of PM tension? The logic here is not clear and the*
*reference not appropriate.*

We agree with the reviewer’s comment that the logic of this sentence is not clear.
Therefore, we have rephrased it to the following sentence to account for the flow of our
discussion: “Our data indicate that ERM-mediated MCA is responsible for maintaining
homeostatic PM tension in non-invasive cells” (page 20, lines 356).

****Yes, this is better phrasing.**

- *Line 374: “It will thus be very important to investigate whether and how their*
*properties can affect MCA and PM tension”. Substrate stiffness has been shown to not*
*affect MCA in mESC (Bergert, Lembo, 2021, Cell Stem Cell). The authors should*

*rephrase that section of the discussion and include this paper or remove the paragraph.*

We agree with this reviewer's comment and have removed this paragraph.

**Seems appropriate.**

*- In that same paper (Bergert, Lembo, 2021, Cell Stem Cell) a cleaner way to perturb*
*MCA is described (iMC-linker). The authors should discuss the perturbations they use*
*and their limitations in light of this new gold standard.*

We have mentioned in the Discussion Section about potential limitations of our study
using MA-ezrin and the need for more studies using this new tool (page 20, lines 347–
355). Thank you for this suggestion.

****The new paragraph in the discussion does a good job of stating the limitations and**
**highlighting this new molecular tool to study MCA.**

*Suggested last experiments:*

*- The authors quantify ERM instead of pERM in Fig 1c,e, 2c,i and 3b,f. They suggest in*
*their rebuttal letter than this is due to the fixation method required but there are*
*antibodies in the market that would be compatible with concomitant phalloidin staining.*
*I think these last experiments are key to supporting their claims.*

As the reviewer indicated, we confirmed that another anti-pERM antibody provided by
CST (#3726) accurately stains formaldehyde-fixed cells. Therefore, all representative
and quantitative data in Fig. 1 (b-e), SFig. 1 (e, f, and g), Fig. 2 (b, c, h, and i), and Fig.
3 (a, b, e, and f) have been changed to pERM (from ERM). Please note that previous
SFig. 3c and d have been removed, because canine MDCK cells were not stained with
this antibody. We believe that these new data provide further support for the importance
of ERM-mediated MCA in suppressing cancer cell motility.

****The use of the pERM data indeed strengthens the conclusions, and the results are**
**consistent with the overall model.**

*Minor comment:*

*- Line 115: PM tension is regulated by the in-plane tension of the lipid bilayer and MCA.*

*Technically it is not “regulated” but “has contributions from”.*

*We have rewritten it as the reviewer suggested (page 7, line 115).*

****I agree this is a more precise way of phrasing this.**

*- Fig 4k and SF4f. The N for Ezrin is really low and specifically for 4f the spread of the*
*data is very large. I would suggest considering removing the Ezrin data as it does not*
*bring much to the paper and the N and spread of the data compromises the conclusion.*

*As the reviewer suggested, we have removed the Ezrin data from the Fig 4k and SF4f.*

****With all due respect to Reviewer 2, I do not agree with this assessment and would**
**leave this data in the paper. Considering these are mouse experiments, N=3-4 is not so**
**unusual, and the authors are using appropriate non-parametric rank tests**
**(Mann-Whitney) to analyze these results. Also, the main conclusions from these panels**
**are the results of the MA-Ezrin, which is clear. The authors do not make any definitive**
**statements based on the parental vs. wt Ezrin results, so I would not say that any**
**conclusions are compromised. I think wt Ezrin is an appropriate control here and should**
**be included if the data exists. Also, it would appear strange to leave wt Ezrin out of Fig.**
**4k, since wt Ezrin is included everywhere else in this figure. Even if the data is more**
**spread and it is harder to conclude, I would not omit this data, but I guess this is up to**
**the authors and editors to decide.**

*- Fig. 1d and f: Why display in one main figure both a single plane and the MIP? I would*
*suggest the authors quantify the localisation in all planes and only display 1f.*

*We believe that a single plane showing a distinct membrane region is necessary for*
*readers to appreciate these data accurately. MIP is suitable for displaying the entire 3D*
*image but does not allow for the extraction of the cell membrane region. In addition, as*
*described above, we quantified the intensity of pERM and F-actin in three planes.*
*Therefore, we would like to respectfully stand by our way of displaying both*
*single-plane and MIP images in Fig.1d in this revised manuscript.*

****I agree with the authors that it is useful for readers to have both the single plane and**
**MIP in order for readers to better appreciate the data used to quantify and to get an**

impression of the localization in 3D. I think the label “3D” on the top left is a little
confusing, though. Maybe this could be changed to “2D” or “single slice”. As
mentioned above, I also agree with the authors that quantification of the equatorial plane
(or the 3 nearest-equatorial planes) provides the most accurate measurements. I do not
think quantification in all planes, as Reviewer 2 suggests, is necessary here.

- *Typo in line 158. “the” is not needed.*

We thank the reviewer for pointing this mistake. However, we have changed the
relevant part as shown in blue (page 9, line 154; previous line 158), as the ERM data
were removed.

****The new wording looks fine.**

- *English in line 165 is confusing. “These” implies the control cells from the sentence
above but I believe the authors mean the the low-tension cells.*

We apologize for the confusion. As pointed out by the reviewer, “the” is correct. We
have corrected this part (page 9, line 160).

****Yes, this sentence is clearer without the pronoun.**

- *Fig 3D is not referred to in the main text.*

We thank the reviewer for pointing this out. Fig.3D is now referred to in the text.

****Fine.**

- *SF4c. Is Mock DMSO? if so can the authors write that?*

We apologize for the lack of information. Mock is water, as MβCD was dissolved in
water. This is now described in the Figure legend.

****Fine, as it is specified now in the legend.**

- *Fig 4e. What is the intracellular structure labelled in the PM channel? Is it labelled*

*with WGA?*

Yes, this was labelled with WGA, as indicated in the legend. We notice that in cancer
cells, WGA tends to stain the inner membrane structures of the cell, even before
permeabilization.

****True that WGA is written in the legend, but it could be clearer that WGA is used for
plasma membrane labeling. Maybe something like: “Plasma Membrane (PM) was
labeled with WGA. Same goes for Fig. 1d.” This would be helpful for readers who are
not familiar with WGA as a PM stain.**

*- Fig 4k. Why do here they show p-values instead of stars like in the rest of the paper?*

This was because the p-value of Fig.4k is higher than the rest of the key data. We have
included a star in Fig 4k and the exact p-value in the corresponding legend.

****This is fine, as the authors prefer.**

*- Line 264: Why is it unexpected? can the authors clarify? do they refer to the
proliferation data?*

As pointed out by the reviewer, we referred to the in vitro proliferation data (SF4a).
Therefore, we have added the following sentence to this part (page 15, line 259;
previous line 264): “although they exhibited the proliferation rate comparable to that of
parental cells in vitro”.

****This clarifies what the authors mean, but I think the wording could be improved,
maybe instead of “...exhibited the proliferation rates comparable...”, use
“...proliferated at rates comparable...”**

*- Is the data in SF5 needed? It does not really fit to Line 282.*

We agree with the reviewer’s suggestion and have removed the previous Supplementary
Fig. 5.

****Fine.**

- Fig 5a. Can they include a color legend?

We have added the description to the corresponding legend (page 42, line 729).

****The addition to the legend helps, but I think a color legend would make this much**
**easier for the reader to understand and should be easy to insert.**

- Reference 12 is not formatted appropriately.

We thank the reviewer for pointing this out. However, we believe that this format is
correct, as this study was conducted by a collaborative network named “Physical
Sciences-Oncology Centers”.

****Fine.**

*Reviewer #3 (Remarks to the Author):*

*After reading the new version of the manuscript and the response to the reviewers’*
*concerns, I found that the authors did a truly exceptional job in addressing all concerns.*
*The authors performed a number of new experiments and quantifications that fit well*
*with the previous data and help to support the overall conclusions of the paper. The*
*authors make a very convincing claim that ERM-mediated MCA, and not just the*
*organization of the actin cortex, is really the crucial parameter to the membrane tension*
*phenotype they observe and attribute to invasiveness. I find the new discussion about*
*PM tension being a prerequisite for all protrusion formation and cortical tension rather*
*being involved in modulating protrusion type especially compelling, and it seems well*
*supported by the data.*

We thank the reviewer for the critical reading of our manuscript and rebuttal, and for
supporting the overall conclusions of our paper.

*I am still, however, a little confused by the model. I understand that BAR protein*
*accumulation/oligomerization is linked to protrusive activity and invasiveness, but it is*
*not really clear in the text what is the causal link. I guess that it is probably attributed to*
*activation of actin nucleation, as the authors previously showed that FBP17 leads to*

*actin polymerization via N-WASP. The authors also provide references in the text*
*showing that MTSSL1 and Toca proteins are also involved in Arp2/3-mediated actin*
*polymerization. I think this could be incorporated into the model figure to clarify the*
*causal link between BAR/Toca proteins and invasiveness. Also, the authors say in the*
*introduction “Given that tense membranes resist cell membrane deformations, PM*
*tension is assumed to disfavor the formation of any membrane protrusions.” This*
*suggests that the barrier to making protrusions is the mechanical bending stiffness of*
*the PM. However, the results in the paper reveal a more complicated mechanism*
*involving curvature-sensing proteins. I think it would be good if the authors could come*
*back to this idea in the Discussion just to clarify this for the readers. Other than that, I*
*have no further suggestions, and I congratulate the authors on a really fantastic*
*revision of this manuscript.*

*We thank the reviewer for these valuable comments. As suggested by the reviewer, we*
*have added a pathway in which a decrease in PM tension leads to invasiveness by*
*promoting Arp2/3-dependent actin polymerization via activation of MTSSL1 and Toca*
*proteins (Fig. 6a). We have rewritten the Discussion Section to make it easier for the*
*readers to appreciate that the mechanical coupling between low PM tension and BAR*
*proteins, rather than just decreased tension, is important for membrane protrusions and*
*motility phenotypes (page 19, lines 330–335). In addition, we have changed the title as*
*follows so that the readers can understand at a glance that the tension-mediated*
*suppression of cancer cell motility is due to the inhibition of BAR proteins:*
***“Homeostatic membrane tension constrains cancer cell dissemination by***
***counteracting BAR proteins”.***

*****The additions to Figure 6 and the Discussion are very helpful and provide readers***
***with a more complete picture of the mechanism. I think the idea of a “mechanical tumor***
***suppressor” is very compelling and an out-of-the-box way of thinking about cancer***
***progression, so I think it adds something to leave that phrasing in the title. I find***
***“counteracting BAR proteins” a little unclear. Maybe “counteracting BAR protein***
***assembly” or “preventing BAR protein assembly” would be more precise. Probably***
***both tumor suppressor idea and BAR proteins could appear in the title. This being said,***
***the phrasing of the title is something the authors should decide themselves, and it does***
***not affect my overall positive opinion about this manuscript.***

As I have been asked to comment on the responses to both my concerns and the
concerns of Reviewer 2, I have written my comments below in a point-by-point manner
with ** at the beginning of my comments and in red. Overall, the authors did a nice job
responding to the reviewers' concerns, and there are just a few very minor changes I
think could be made prior to publication.

We would like to thank the reviewer for careful evaluation of our manuscript and for
helpful and constructive comments and suggestions. Please note the point-by-point
responses below, which are given in **green**.

*REVIEWER COMMENTS*

*Reviewer #1 (Remarks to the Author):*

*The authors have carefully addressed all of my comments with either new experiments,*
*quantifications, figure reorganization or thoughtful answers. I believe this manuscript is*
*significantly improved.*

*Jean-Francois Cote*

We thank the reviewer for your time and dedication in carefully reviewing our study.
Your valuable suggestions helped us much improve the manuscript.

*Reviewer #2 (Remarks to the Author):*

*Tsujita et al have provided a much improved version of their manuscript entitled:*
*“Homeostatic membrane tension acts as a mechanical tumor suppressor and constrains*
*cancer cell dissemination”.* *The story is clearer and the simplification of data in several*
*instances makes it easier to follow, which I think grant publication in Nature*
*Communications. Nevertheless I think there are several points that need to be clarified*
*in the text and methods before publication. Last, I would suggest repeating one set of*
*experiments to support their claims.*

We thank the reviewer for the careful reading of our manuscript and the many helpful

comments. In particular, we believe that new pERM data provide further support for our
model.

*Major comment regarding text:*

- *In several instance the authors cite reviews instead of primary research to support*
*their claims. For example in line 117 (this is particularly a concerning one as these*
*reviews (refs. 22, 23) are not even supporting their claims). This happens again in lines*
*243 and 398, where they cite reviews (refs. 24 and 47) instead of showing the primary*
*data paper.*

We agree with this reviewer's comment and have changed the references corresponding
to the relevant parts to the primary data papers as follows.

Line 117: refs. 25-27. In addition, we have removed the sentence "under static
conditions", as they do not mention this.

Line 238 (previous line 243): refs. 23 and 47.

Line 399 (previous line 398): please note that this (ref. 49) is the primary research
paper.

We also have replaced the review with the research papers (refs. 60–63: line 345). In
addition, we have added several important research papers that we had overlooked in
the previous manuscript (refs. 29, 36, 56, 68).

****This seems better now with the primary literature.**

**We thank the reviewer for this positive comment.**

- *How the quantifications of ERM and pERM are done is really not clear in the methods*
*section. More detail should be provided to not compromise their results. How do the*
*authors determine the lateral side (as quantified in SF1f)? Do they measure both sides*
*or only one? Why is only one plane quantified from 3D stacks? How is the cell*
*segmentation done and how many pixels are considered "edge" (SF1g) or "membrane"*
*(Fig 1c,e, 2c,i) by the authors?*

Firstly, please note that we have changed all ERM data to pERM data according to the
reviewers' advice.

Line profiles for the PM, pERM and F-actin levels were measured across one lateral
side, as indicated by the white lines in SF1e. In the case of cells that do not have a clear
membrane extension or cell rear, such as MCF10A cells, the measurement points were
arbitrarily determined. Please note that pERM and F-actin are distributed globally in
these cells.

We quantified one plane image near the middle of 3D stacks where the membrane
region can be clearly defined by WGA staining, since, as the reviewer pointed out in the
first revision, membrane staining must be included when quantifying pERM and F-actin
levels in the PM. However, as the reviewer pointed out, we also had to account for any
variation in fluorescence intensity due to the thickness of 3D stacks. Therefore, two
images shifted up and down in the z-direction (within the range where the membrane
region could be identified) from the selected middle image were also quantified, and the
average of the total three quantitative data was used as one sample. These quantification
methods, image segmentation, and pixel sizes are described in the Methods Section
(page 66, lines 1072–1084).

****This seems clear in the figure. In my opinion doing this on an equatorial slice is the
correct way to measure, especially for the linescans, as projections or measurements on
non-equatorial slice would lead to artifacts due to the cell curvature.**

**We thank the reviewer for the positive evaluation of our quantification methods.**

*- The authors should specify how the plane displayed in figures is selected from 3D
stacks.*

As mentioned above, we selected the plane image with clearly distinguishable
membrane regions that corresponds to near the middle of 3D stacks. This is now
described in the Methods Section (page 66, line 1059).

****This seems fine. Normally it is very easy to select an equatorial slice by eye from a
3D stack.**

We thank the reviewer for this positive comment.

- *The quantification of ERM in 3D is not sufficient to make the claim that membrane*
*tension is changed (in support to the point raised by reviewer 1). I suggest the authors*
*tone down their claims.*

We agree with the reviewer's suggestion and have removed the relevant sentence
(previous lines 137–139) and rewritten the sentence as follows: "These results suggest
that the observed differences in PM tension between non-motile and invasive cells are
primarily due to ERM-mediated changes in MCA...." (page 8, line 134).

****This change seems appropriate.**

We thank the reviewer for the comment that this change is appropriate.

- *The experiments with cyclodextrin are not a clean way to perturb membrane tension.*
*The authors should consider toning down their claims even though these experiments*
*are crucial to their rebuttal to reviewer 1. As written in the review the authors cite (ref*
*24): "Reducing plasma membrane cholesterol with methyl-beta cyclodextrin causes an*
*increase in membrane tension in embry-onic kidney cells, fibroblasts and*
*cardiomyocytes. However, this increase is not only due to changes in membrane*
*composition, but also correlates with cytoskeletal rearrangements. Upon methyl-beta*
*cyclodextrin, fibroblasts showed de novo actin polymerization and Rho-GTPase*
*dependent stress fiber formation whereas cardiomyocytes showed sarcomeric*
*disorganization accompanied by contraction abnormalities".*

As suggested by the reviewer, we have deleted the relevant sentence (previous lines
244–245) to tone down our claims based on M β CD data. Instead, we have added a text
as follows: "ruling out potential effects of cortical stiffness on inhibiting cancer cell
migration" (page 14, line 239).

****This seems like an appropriate change.**

We thank the reviewer for this positive comment.

- *Line 133: “In both cases, membrane-proximal ERM and F-actin displayed overall*
*decreased levels, although their local accumulation was observed to be the same as that*
*in 2D (Fig. 1d–f).” How is the local accumulation assessed here? Can the author*
*clarify what they mean?*

In this sentence, we wanted to point out that, as in 2D, the pERM signal is enriched in
the rear and blebbing membranes. However, since what we want to show by these data
is that pERM levels are overall higher in non-invasive cells than in invasive cells, we
have deleted this unnecessary sentence.

****Seems appropriate.**

We thank for reviewer’s comment.

- *Line 170: E-cadherin depletion shows a significant decrease in invasion and*
*migration (Fig 2e and f) so that sentence should be rewritten.*

We have added a sentence as follows: “E-cadherin depletion resulted in a slight but
significant inhibition of invasion and migration” (page 10, line 166).

****This new sentence accurately reflects the data.**

We thank the reviewer for this positive comment.

- *Line 223: “We hypothesized that if PM tension reduction is key to acquiring migration*
*and invasiveness, increasing cell membrane mechanics might be sufficient to suppress*
*cancer cell dissemination.” This sentence is very vague... Can the authors rewrite*
*“increasing cell membrane mechanics” by a more precise term? Like increasing MCA?*

We thank the reviewer for this suggestion and have replaced it with “increasing MCA”
(page 13, line 219).

****Good.**

Thank you.

- Line 251: “We found that changes in ERM levels at the PM were always correlated
with changes in membrane-proximal F-actin levels; this indicated that MCA is the key
determinant of PM tension, as recently described in other experimental settings”. First,
how does a correlation between ERM levels and membrane-proximal F-actin indicate
that MCA is the key determinant of PM tension? The logic here is not clear and the
reference not appropriate.

We agree with the reviewer’s comment that the logic of this sentence is not clear.
Therefore, we have rephrased it to the following sentence to account for the flow of our
discussion: “Our data indicate that ERM-mediated MCA is responsible for maintaining
homeostatic PM tension in non-invasive cells” (page 20, lines 356).

****Yes, this is better phrasing.**

**We thank for reviewer’s comment.**

- Line 374: “It will thus be very important to investigate whether and how their
properties can affect MCA and PM tension”. Substrate stiffness has been shown to not
affect MCA in mESC (Bergert, Lembo, 2021, Cell Stem Cell). The authors should
rephrase that section of the discussion and include this paper or remove the paragraph.

**We agree with this reviewer’s comment and have removed this paragraph.**

**Seems appropriate.**

**We thank for reviewer’s comment.**

- In that same paper (Bergert, Lembo, 2021, Cell Stem Cell) a cleaner way to perturb
MCA is described (iMC-linker). The authors should discuss the perturbations they use
and their limitations in light of this new gold standard.

**We have mentioned in the Discussion Section about potential limitations of our study
using MA-ezrin and the need for more studies using this new tool (page 20, lines 347–
355). Thank you for this suggestion.**

****The new paragraph in the discussion does a good job of stating the limitations and**
**highlighting this new molecular tool to study MCA.**

**We thank the reviewer for this positive comment.**

*Suggested last experiments:*

*- The authors quantify ERM instead of pERM in Fig 1c,e, 2c,i and 3b,f. They suggest in*
*their rebuttal letter than this is due to the fixation method required but there are*
*antibodies in the market that would be compatible with concomitant phalloidin staining.*
*I think these last experiments are key to supporting their claims.*

*As the reviewer indicated, we confirmed that another anti-pERM antibody provided by*
*CST (#3726) accurately stains formaldehyde-fixed cells. Therefore, all representative*
*and quantitative data in Fig. 1 (b-e), SFig. 1 (e, f, and g), Fig. 2 (b, c, h, and i), and Fig.*
*3 (a, b, e, and f) have been changed to pERM (from ERM). Please note that previous*
*SFig. 3c and d have been removed, because canine MDCK cells were not stained with*
*this antibody. We believe that these new data provide further support for the importance*
*of ERM-mediated MCA in suppressing cancer cell motility.*

****The use of the pERM data indeed strengthens the conclusions, and the results are**
**consistent with the overall model.**

**We thank the reviewer for evaluating the new pERM data and supporting our overall**
**model.**

*Minor comment:*

*- Line 115: PM tension is regulated by the in-plane tension of the lipid bilayer and MCA.*
*Technically it is not “regulated” but “has contributions from”.*

*We have rewritten it as the reviewer suggested (page 7, line 115).*

****I agree this is a more precise way of phrasing this.**

**We thank for reviewer’s comment.**

*- Fig 4k and SF4f. The N for Ezrin is really low and specifically for 4f the spread of the*

*data is very large. I would suggest considering removing the Ezrin data as it does not*
*bring much to the paper and the N and spread of the data compromises the conclusion.*

*As the reviewer suggested, we have removed the Ezrin data from the Fig 4k and SF4f.*

***With all due respect to Reviewer 2, I do not agree with this assessment and would*
*leave this data in the paper. Considering these are mouse experiments, N=3-4 is not so*
*unusual, and the authors are using appropriate non-parametric rank tests*
*(Mann-Whitney) to analyze these results. Also, the main conclusions from these panels*
*are the results of the MA-Ezrin, which is clear. The authors do not make any definitive*
*statements based on the parental vs. wt Ezrin results, so I would not say that any*
*conclusions are compromised. I think wt Ezrin is an appropriate control here and should*
*be included if the data exists. Also, it would appear strange to leave wt Ezrin out of*
*Fig.4k, since wt Ezrin is included everywhere else in this figure. Even if the data is*
*more spread and it is harder to conclude, I would not omit this data, but I guess this is*
*up to the authors and editors to decide.*

*We thank the reviewer for constructive comments. We fully agree with the reviewer's*
*opinion and have brought back the ezrin data to Fig.4k and Fig. S4f.*

*- Fig. 1d and f: Why display in one main figure both a single plane and the MIP? I would*
*suggest the authors quantify the localisation in all planes and only display 1f.*

*We believe that a single plane showing a distinct membrane region is necessary for*
*readers to appreciate these data accurately. MIP is suitable for displaying the entire 3D*
*image but does not allow for the extraction of the cell membrane region. In addition, as*
*described above, we quantified the intensity of pERM and F-actin in three planes.*
*Therefore, we would like to respectfully stand by our way of displaying both*
*single-plane and MIP images in Fig.1d in this revised manuscript.*

***I agree with the authors that it is useful for readers to have both the single plane and*
*MIP in order for readers to better appreciate the data used to quantify and to get an*
*impression of the localization in 3D. I think the label "3D" on the top left is a little*
*confusing, though. Maybe this could be changed to "2D" or "single slice". As*
*mentioned above, I also agree with the authors that quantification of the equatorial plane*
*(or the 3 nearest-equatorial planes) provides the most accurate measurements. I do not*

think quantification in all planes, as Reviewer 2 suggests, is necessary here.

We are pleased that the reviewer agrees that showing both the single plane and MIP
helps readers to appreciate our data accurately. We agree with the reviewer's comment
that the label "3D" is confusing. Therefore, we have changed "3D" to "single slice" on
the top left in Fig. 1d, as the reviewer suggested. We also thank the reviewer for your
positive evaluation of our measurements.

- *Typo in line 158. "the" is not needed.*

We thank the reviewer for pointing this mistake. However, we have changed the
relevant part as shown in blue (page 9, line 154; previous line 158), as the ERM data
were removed.

****The new wording looks fine.**

We thank for reviewer's comment.

- *English in line 165 is confusing. "These" implies the control cells from the sentence*
*above but I believe the authors mean the the low-tension cells.*

We apologize for the confusion. As pointed out by the reviewer, "the" is correct. We
have corrected this part (page 9, line 160).

****Yes, this sentence is clearer without the pronoun.**

We thank for reviewer's comment.

- *Fig 3D is not referred to in the main text.*

We thank the reviewer for pointing this out. Fig.3D is now referred to in the text.

****Fine.**

Thank you.

- *SF4c. Is Mock DMSO? if so can the authors write that?*

We apologize for the lack of information. Mock is water, as M β CD was dissolved in
water. This is now described in the Figure legend.

****Fine, as it is specified now in the legend.**

We thank for reviewer's comment.

- *Fig 4e. What is the intracellular structure labelled in the PM channel? Is it labelled*
*with WGA?*

Yes, this was labelled with WGA, as indicated in the legend. We notice that in cancer
cells, WGA tends to stain the inner membrane structures of the cell, even before
permeabilization.

****True that WGA is written in the legend, but it could be clearer that WGA is used for**
**plasma membrane labeling. Maybe something like: "Plasma Membrane (PM) was**
**labeled with WGA. Same goes for Fig. 1d." This would be helpful for readers who are**
**not familiar with WGA as a PM stain.**

We thank the reviewer for this suggestion. We have added this sentence "Plasma
membrane (PM) was labeled with WGA" in the legend corresponding to Fig. 1d (page
49, line 916).

- *Fig 4k. Why do here they show p-values instead of stars like in the rest of the paper?*

This was because the p-value of Fig.4k is higher than the rest of the key data. We have
included a star in Fig 4k and the exact p-value in the corresponding legend.

****This is fine, as the authors prefer.**

We thank for reviewer's comment.

- *Line 264: Why is it unexpected? can the authors clarify? do they refer to the*
*proliferation data?*

As pointed out by the reviewer, we referred to the in vitro proliferation data (SF4a).
Therefore, we have added the following sentence to this part (page 15, line 259;
previous line 264): “although they exhibited the proliferation rate comparable to that of
parental cells in vitro”.

****This clarifies what the authors mean, but I think the wording could be improved,
maybe instead of “...exhibited the proliferation rates comparable...”, use
“...proliferated at rates comparable...”**

We thank the reviewer for this suggestion. We have changed “...exhibited the
proliferation rates comparable...”, to “...proliferated at rates comparable...”in the
sentence, as the reviewer suggested (page 15, line 260).

*- Is the data in SF5 needed? It does not really fit to Line 282.*

We agree with the reviewer’s suggestion and have removed the previous Supplementary
Fig. 5.

****Fine.**

Thank you.

*- Fig 5a. Can they include a color legend?*

We have added the description to the corresponding legend (page 42, line 729).

****The addition to the legend helps, but I think a color legend would make this much
easier for the reader to understand and should be easy to insert.**

We thank the reviewer for this suggestion. The color legend has been inserted in Fig. 5a.

*- Reference 12 is not formatted appropriately.*

We thank the reviewer for pointing this out. However, we believe that this format is
correct, as this study was conducted by a collaborative network named “Physical

Sciences-Oncology Centers”.

****Fine.**

Thank you.

*Reviewer #3 (Remarks to the Author):*

*After reading the new version of the manuscript and the response to the reviewers’*
*concerns, I found that the authors did a truly exceptional job in addressing all concerns.*
*The authors performed a number of new experiments and quantifications that fit well*
*with the previous data and help to support the overall conclusions of the paper. The*
*authors make a very convincing claim that ERM-mediated MCA, and not just the*
*organization of the actin cortex, is really the crucial parameter to the membrane tension*
*phenotype they observe and attribute to invasiveness. I find the new discussion about*
*PM tension being a prerequisite for all protrusion formation and cortical tension rather*
*being involved in modulating protrusion type especially compelling, and it seems well*
*supported by the data.*

We thank the reviewer for the critical reading of our manuscript and rebuttal, and for
supporting the overall conclusions of our paper.

*I am still, however, a little confused by the model. I understand that BAR protein*
*accumulation/oligomerization is linked to protrusive activity and invasiveness, but it is*
*not really clear in the text what is the causal link. I guess that it is probably attributed to*
*activation of actin nucleation, as the authors previously showed that FBP17 leads to*
*actin polymerization via N-WASP. The authors also provide references in the text*
*showing that MTSSL1 and Toca proteins are also involved in Arp2/3-mediated actin*
*polymerization. I think this could be incorporated into the model figure to clarify the*
*causal link between BAR/Toca proteins and invasiveness. Also, the authors say in the*
*introduction “Given that tense membranes resist cell membrane deformations, PM*
*tension is assumed to disfavor the formation of any membrane protrusions.” This*
*suggests that the barrier to making protrusions is the mechanical bending stiffness of*
*the PM. However, the results in the paper reveal a more complicated mechanism*
*involving curvature-sensing proteins. I think it would be good if the authors could come*
*back to this idea in the Discussion just to clarify this for the readers. Other than that, I*

*have no further suggestions, and I congratulate the authors on a really fantastic*
*revision of this manuscript.*

We thank the reviewer for these valuable comments. As suggested by the reviewer, we
have added a pathway in which a decrease in PM tension leads to invasiveness by
promoting Arp2/3-dependent actin polymerization via activation of MTSS1L and Toca
proteins (Fig. 6a). We have rewritten the Discussion Section to make it easier for the
readers to appreciate that the mechanical coupling between low PM tension and BAR
proteins, rather than just decreased tension, is important for membrane protrusions and
motility phenotypes (page 19, lines 330–335). In addition, we have changed the title as
follows so that the readers can understand at a glance that the tension-mediated
suppression of cancer cell motility is due to the inhibition of BAR proteins:
**“Homeostatic membrane tension constrains cancer cell dissemination by**
**counteracting BAR proteins”**.

****The additions to Figure 6 and the Discussion are very helpful and provide readers**
**with a more complete picture of the mechanism. I think the idea of a “mechanical tumor**
**suppressor” is very compelling and an out-of-the-box way of thinking about cancer**
**progression, so I think it adds something to leave that phrasing in the title. I find**
**“counteracting BAR proteins” a little unclear. Maybe “counteracting BAR protein**
**assembly” or “preventing BAR protein assembly” would be more precise. Probably**
**both tumor suppressor idea and BAR proteins could appear in the title. This being said,**
**the phrasing of the title is something the authors should decide themselves, and it does**
**not affect my overall positive opinion about this manuscript.**

We are pleased that the reviewer agrees that new Fig. 6 and our discussion are helpful
for readers to understand our proposed model. We thank the reviewer for your feedback
on the title. Based on the reviewer’s opinion, we have changed the title as follows:
**“Homeostatic membrane tension acts as a tumor suppressor and constrains cancer**
**cell dissemination by counteracting BAR protein assembly.**
